# Feature Adaptation for Sparse Linear Regression

**Jonathan A. Kelner**[*]
MIT

**Frederic Koehler**[†]
Stanford

**Raghu Meka**[‡]
UCLA

**Dhruv Rohatgi**[§]
MIT

## Abstract

Sparse linear regression is a central problem in high-dimensional statistics. We study the correlated random design setting, where the covariates are drawn from a multivariate Gaussian $N(0, \Sigma)$, and we seek an estimator with small excess risk.

If the true signal is $t$-sparse, information-theoretically, it is possible to achieve strong recovery guarantees with only $O(t \log n)$ samples. However, computationally efficient algorithms have sample complexity linear in (some variant of) the *condition number* of $\Sigma$. Classical algorithms such as the Lasso can require significantly more samples than necessary even if there is only a single sparse approximate dependency among the covariates.

We provide a polynomial-time algorithm that, given $\Sigma$, automatically adapts the Lasso to tolerate a small number of approximate dependencies. In particular, we achieve near-optimal sample complexity for constant sparsity and if $\Sigma$ has few "outlier" eigenvalues. Our algorithm fits into a broader framework of *feature adaptation* for sparse linear regression with ill-conditioned covariates. With this framework, we additionally provide the first polynomial-factor improvement over brute-force search for constant sparsity $t$ and arbitrary covariance $\Sigma$.

## 1 Introduction

Sparse linear regression is a fundamental problem in high-dimensional statistics. In a natural random design formulation of this problem, we are given $m$ independent and identically distributed samples $(X_i, y_i)_{i=1}^m$ where each sample's covariates are drawn from an $n$-dimensional Gaussian random vector $X_i \sim N(0, \Sigma)$, and each response is $y_i = \langle X_i, v^* \rangle + \xi_i$ for independent noise $\xi_i \sim N(0, \sigma^2)$ and a $t$-sparse ground truth regressor $v^* \in \mathbb{R}^n$, where $t$ is much smaller than $n$. The goal[5] is to output a vector $\hat{v} \in \mathbb{R}^n$ for which the *excess risk*

$$\mathbb{E}(\langle X_0, \hat{v} \rangle - y_0)^2 - \sigma^2 = (\hat{v} - v^*)^\top \Sigma (\hat{v} - v^*) =: \|\hat{v} - v^*\|_\Sigma^2$$

is as small as possible, where $(X_0, y_0)$ is an independent sample from the same model.

Without the sparsity assumption, the number of samples needed to achieve small excess risk (say, $O(\sigma^2)$) is linear in the dimension; with $O(n)$ samples, simple and computationally efficient algorithms such as ordinary least squares achieve the statistically optimal excess risk $O\left(\frac{\sigma^2 n}{m}\right)$. Sparsity

---

[*]kelner@mit.edu. This work was supported in part by NSF Large CCF-1565235, NSF Medium CCF-1955217, and NSF TRIPODS 1740751.

[†]fkoehler@stanford.edu. This work was supported in part by NSF award CCF-1704417, NSF award IIS-1908774, and N. Anari's Sloan Research Fellowship

[‡]raghum@cs.ucla.edu. This work was supported in part by NSF CAREER Award CCF-1553605 and NSF Small CCF-2007682

[§]drohatgi@mit.edu. This work was supported by a U.S. DoD NDSEG Fellowship.

[5]More generally, from a learning theory perspective, we could consider an arbitrary improper learner outputting a function $\hat{f}(X_0)$, rather than specifically learning a linear function $\langle X_0, \hat{v} \rangle$. At least when $\Sigma$ is known, there is no advantage as we can always project $\hat{f}$ onto the space of linear functions.

allows for a significant statistical improvement: ignoring computational efficiency, it is well known that there is an estimator $\hat{v}$ with excess risk $O(\frac{\sigma^2 t \log n}{m})$ as long as $m = \Omega(t \log n)$ (see e.g. [13, 33]; Theorem 4.1 in [23]).

The catch is that computing this estimator involves a brute-force search over $\binom{n}{t}$ possibilities (i.e., the possible supports for $v^*$). At first glance, this combinatorial search may seem unavoidable if we wish to take advantage of sparsity. Indeed, similar problems are notoriously difficult: the only non-trivial algorithms for e.g., learning $t$-sparse parities with noise still require $n^{\Omega(t)}$ time [29, 37]. However, it is a celebrated fact that for sparse linear regression, computationally efficient methods such as Lasso and Orthogonal Matching Pursuit can avoid this combinatorial search and still achieve very strong theoretical guarantees under conditions such as the Restricted Isometry Property (see e.g. [7, 10, 5, 4, 3, 1]). In the random design setting we consider, the Lasso is known to achieve optimal statistical rates (up to constants) when the covariance matrix $\Sigma$ is *well-conditioned* [32, 46].

What about when $\Sigma$ is ill-conditioned? In contrast with the statistically optimal estimator, Lasso and its cousins provably *require* sample complexity scaling with (some variant of) the condition number of $\Sigma$ (see e.g. Theorem 14 in [38] or Theorem 6.5 in [23]). And with a few exceptions (e.g., in some settings with special graphical structure [23]) there has been little progress on designing new efficient algorithms for sparse linear regression with ill-conditioned $\Sigma$ (see Section 4 for further discussion). For a general covariance $\Sigma$, no algorithm is even known that can achieve sample complexity $f(t) \cdot n^{1-\epsilon}$ (for an arbitrary function $f$) without brute-force search.

A computationally efficient algorithm that approaches the optimal statistical rate for *arbitrary* $\Sigma$ might be too much to hope for. While no computational lower bounds are known, even in restricted computational models such as the Statistical Query model,[6] the related *worst-case* problem of finding a $t$-sparse solution to a system of linear equations requires $n^{\Omega(t)}$ time under standard complexity assumptions [15]. So it is plausible, though not certain, that some assumptions on $\Sigma$ are necessary. In this work – inspired by a long tradition (in random matrix theory, statistics, graph theory, and other areas) of studying matrices with a spectrum that is split between a large "bulk" and a small number of outlier "spike" eigenvalues [28, 39, 43] – we identify a broad generalization of the standard well-conditionedness assumption, under which brute-force search can still be avoided.

## 1.1 Beyond well-conditioned $\Sigma$

Say that $\Sigma$ has eigenvalues $\lambda_1 \leq \cdots \leq \lambda_n$, and that the sparsity $t$ is a constant.[7] Then standard bounds for Lasso require sample complexity $(\lambda_n/\lambda_1) \cdot O(\log n)$. But if the covariates contain even a single approximate linear dependency, then $\lambda_n/\lambda_1$ may be arbitrarily large. Moreover, if the dependency is sparse (e.g. two covariates are highly correlated), then there is a natural choice of $v^*$ for which Lasso provably fails (see Theorem 6.5 of [23]). Indeed, this phenomenon is not just a limitation of the analysis; Lasso fails empirically as well, even for very small $t$ (see Figure 2 in Appendix H for a simple example with $t = 3$).

Such dependencies arise in applications ranging from finance (e.g., where some pairs of stocks or ETFs may be highly correlated, and an investor may be interested in the differences) to genomic data (where functionally related genes may have highly correlated expression patterns). Two-sparse dependencies can be directly identified by looking at the covariance matrix; see Section 4 for some discussion of previous research in this direction. But as $t$ increases, naive methods for identifying $t$-sparse dependencies quickly become computationally intractable. With domain knowledge, it may be possible to manually identify and correct such dependencies, but this process would also be time-consuming. Thus, we ask the following question: instead of assuming that $\lambda_n/\lambda_1$ is bounded, suppose that there are constants $d_\ell$ and $d_h$ so that $\lambda_{n-d_h}/\lambda_{d_\ell+1}$ is bounded, i.e. the spectrum of $\Sigma$ has only $d_\ell$ outliers at the low end, and only $d_h$ outliers at the high end. Can we still design an algorithm that achieves sample complexity $O(\log n)$ without resorting to brute-force search?

**Main result.** We give a positive answer: an algorithm for sparse linear regression that is both computationally and statistically efficient for covariance matrices with a small number of "outlier"

---

[6]There are lower bounds for a family of regression estimators with coordinate-separable regularization [44] and a family of "preconditioned-Lasso" estimators [23, 24].

[7]Note that for moderate-sized datasets (e.g. $n = 1000$), brute-force search is infeasible even for $t$ as small as four or five.

eigenvalues. In particular, this means we can handle a few approximate dependencies among the covariates (quantified by the number of eigenvalues below a threshold). In comparison, Lasso and other classical algorithms cannot tolerate even a single sparse approximate dependency. Our main algorithmic result is the following:

**Theorem 1.1.** *Let $n, t, d_\ell, d_h, L \in \mathbb{N}$ and $\sigma, \delta > 0$. Let $\Sigma \in \mathbb{R}^{n \times n}$ be a positive semi-definite matrix with (non-negative) eigenvalues $\lambda_1 \leq \cdots \leq \lambda_n$. Let $v^* \in \mathbb{R}^n$ be any $t$-sparse vector. Let $(X_i, y_i)_{i=1}^m$ be independent with $X_i \sim N(0, \Sigma)$ and $y_i = \langle X_i, v^* \rangle + \xi_i$, where $\xi_i \sim N(0, \sigma^2)$.*

*Let $n_{eff} := t(\lambda_{n-d_h}/\lambda_{d_\ell+1}) \log(nL/\delta) + t^{O(t)} d_l + d_h$. Given $\Sigma$, $t$, $d_\ell$, $\delta$, and $(X_i, y_i)_{i=1}^m$, there is an estimator $\hat{v} \in \mathbb{R}^n$ that has excess risk*

$$\|\hat{v} - v^*\|_\Sigma^2 \leq O\left(\frac{\sigma^2 n_{eff} L}{m}\right) + 2^{-L} \cdot \|v^*\|_\Sigma^2$$

*with probability at least $1 - \delta$, so long as $m \geq \Omega(n_{eff}L)$. Moreover, $\hat{v}$ can be computed in time $\text{poly}(n)$.*

Specifically, taking $L \sim \log(m \|v^*\|_\Sigma^2 / \sigma^2)$, the time complexity is dominated by $L$ eigendecompositions and $L$ calls to a Lasso program, for overall runtime $\tilde{O}(n^3)$ (see Algorithm 2). This is substantially faster than the brute-force method (which takes $O(n^t)$ time) even for small values of $t$.

The excess risk decays at rate $\tilde{O}(\sigma^2 n_{\text{eff}}/m)$ (hiding the logarithmic factor), which is near the statistically optimal rate of $\tilde{O}(\sigma^2 t/m)$ so long as $n_{\text{eff}}$ is small, i.e. $t$ is small and only a few eigenvalues lie outside a constant-factor range. In our analysis, we prove that the standard Lasso estimator can already tolerate a few *large* eigenvalues — the main algorithmic innovation is needed to tolerate a few *small* eigenvalues, which turns out to be much trickier. Notice that when $d_\ell = d_h = 0$ we recover standard Lasso guarantees up to the factor of $L$; thus, Theorem 1.1 morally represents a generalization of classical results.

We also show how to achieve a different trade-off between time and samples, eliminating the dependence on $d_\ell$ in sample complexity at the cost of larger runtime:

**Theorem 1.2.** *In the setting of Theorem 1.1, let $n'_{eff} := t(\lambda_{n-d_h}/\lambda_{d_\ell+1}) \log(nL/\delta) + t^2 \log(t) + d_h$. Given $\Sigma$, $t$, $d_\ell$, $\delta$, and $(X_i, y_i)_{i=1}^m$, there is an estimator $\hat{v} \in \mathbb{R}^n$ that has excess risk*

$$\|\hat{v} - v^*\|_\Sigma^2 \leq O\left(\frac{\sigma^2 n'_{eff} L}{m}\right) + 2^{-L} \cdot \|v^*\|_\Sigma^2$$

*with probability at least $1 - \delta$, so long as $m \geq \Omega(n'_{eff}L)$. Moreover, $\hat{v}$ can be computed in time $\text{poly}(n, m, d_\ell^t, t^{t^2})$.*

**Discussion & limitations.** We discuss two limitations of the above results. First, both results incur exponential dependence on the sparsity $t$ (in the sample complexity for Theorem 1.1, and the runtime for Theorem 1.2), which may be suboptimal. For Theorem 1.1, we remark that in practice the algorithm may not suffer this dependence (see e.g. Figure 1), and it is possible that the analysis can be tightened. For Theorem 1.2, we emphasize that the runtime is still fundamentally different than brute-force search: in particular, it's *fixed-parameter tractable* in $t$ and $d_\ell$.

Second, both results require that $\Sigma$ is known. Thus, they are only applicable in settings where we either have a priori knowledge, or can estimate $\Sigma$ accurately because a large amount of unlabelled data is available. At a high level, this limitation is due to the need to compute the eigendecomposition of $\Sigma$, which cannot be approximated from the empirical covariance of a small number of samples.

For simplicity, we have stated our results in terms of Gaussian covariates and noise, but this is not a fundamental limitation. We expect it is possible to prove similar results in the sub-Gaussian case at the cost of making the proof longer — for instance, by building upon the techniques from [25] and related works.

**Pseudocode & simulation.** See Algorithm 1 for complete pseudocode of `AdaptedBP()`, a simplification of the method for the noiseless setting $\sigma = 0$. In Figure 1 we show that `AdaptedBP()` significantly outperforms standard Basis Pursuit (i.e. Lasso for noiseless data [7]) on a simple example with $n = 1000$ variables, $d_\ell = 10$ sparse approximate dependencies, and a ground truth

**Algorithm 1:** Adapted BP for sparse linear regression with few outlier eigenvalues

---

**Procedure** FindHeavyCoordinates($\{v_1, \ldots, v_k\}, \alpha$)

    `/* GRAM-SCHMIDT computes an orthonormalization of` $v_1, \ldots, v_k$     `*/`

    $a_1, \ldots, a_k \leftarrow$ GRAM-SCHMIDT($\{v_1, \ldots, v_k\}$)

    **return** $\{i \in [n] : \sum_{j=1}^{k} ((a_j)_i)^2 \geq \alpha^2\}$

**Procedure** IterativePeeling($\Sigma, d, t$)

    Compute eigendecomposition $\Sigma = \sum_{i=1}^{n} \lambda_i u_i u_i^\top$

    $P \leftarrow \sum_{i=d+1}^{n} u_i u_i^\top$

    $K_t \leftarrow \{i \in [n] : P_{ii} < 1 - 1/(9t^2)\}$

    **for** $j = t$ *to* $1$ **do**

        $\mathcal{I}_P(K_j) \leftarrow$ FindHeavyCoordinates($\{P_i : i \in K_j\}, 1/(6t)$)

        $K_{j-1} \leftarrow K_j \cup \mathcal{I}_P(K_j)$

    **return** $K_0$

**Procedure** AdaptedBP($\Sigma$, $d$, $t$, $(X_i, y_i)_{i=1}^{m}$)

    $S \leftarrow$ IterativePeeling($\Sigma, d, t$)

    **return** $\hat{v} \in \text{argmin}_{v \in \mathbb{R}^n : \mathbb{X}v = y} \sum_{i \notin S} |v_i|$

---

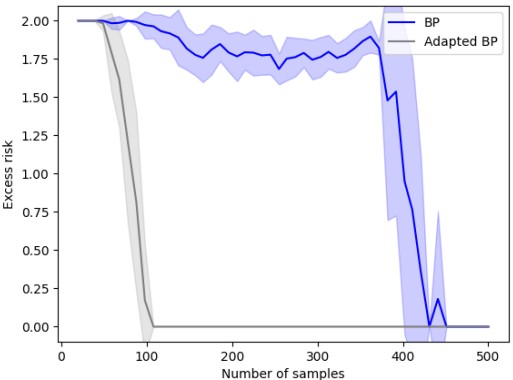

Figure 1: Basis Pursuit (BP) versus Adapted BP in a simple synthetic example with $n = 1000$ covariates. The $x$-axis is the number of samples. The $y$-axis is the out-of-sample prediction error (averaged over 10 independent runs, and error bars indicate the standard deviation).

regressor with sparsity $t = 13$. The covariates $X_{1:1000}$ are all independent $N(0, 1)$ except for 10 disjoint triplets $\{(X_i, X_{i+1}, X_{i+2}) : i = 1, 4, \ldots, 28\}$, each of which has joint distribution

$$X_i := Z_i; \quad X_{i+1} = Z_i + 0.4 Z_{i+1}; \quad X_{i+2} = Z_{i+1} + 0.4 Z_{i+2}$$

where $Z_i, Z_{i+1}, Z_{i+2} \sim N(0, 1)$ are independent. The (noiseless) responses are $y = 6.25(X_1 - X_2) + 2.5 X_3 + \frac{1}{\sqrt{10}} \sum_{i=991}^{1000} X_i$. See Appendix I for implementation details.

## 1.2 Organization

In Section 2 we give an overview of the proofs of Theorem 1.1 and 1.2 (the complete proofs and full algorithm pseudocode are given in Appendix C). In Section 3 we discuss our other results obtained via feature adaptation. Section 4 covers related work.

## 2 Proof techniques

We obtain Theorems 1.1 and 1.2 as outcomes of a flexible algorithmic approach for tackling sparse linear regression with ill-conditioned covariates: *feature adaptation*. As a pre-processing step, adapt

or augment the covariates with additional features (i.e. well-chosen linear combinations of the covariates). Then, to predict the responses, apply $\ell_1$-regularized regression (Lasso) over the new set of features rather than the original covariates. In other words, we algorithmically change the *dictionary* (set of features) used in the Lasso regression. See Section 4 for a comparison to past approaches.

We start by explaining the goals of feature adaptation for general $\Sigma$, and then show how we achieve those desiderata when $\Sigma$ has few outlier eigenvalues. More precisely, the main technical difficulty is in dealing with the small eigenvalues, so in this proof overview we focus on the case where the only outliers are small eigenvalues. Complete proofs of Theorems 1.1 and 1.2 are in Appendix C.

## 2.1 What makes a good dictionary: the view from weak learning

Obviously, the feature adaptation approach generalizes Lasso. Surprisingly, even though the sample complexity of the standard Lasso estimator is thoroughly understood, the basic question of whether for *every* covariate distribution (i.e. every $\Sigma$) there *exists* a good dictionary remains wide-open. To crystallize the power of feature adaptation, we introduce the following notion of a "good" dictionary. We suggest considering the simplified setting of $\alpha$-*weak learning*, where the goal is just to find some $\hat{v}$ so that the predictions $\langle X, \hat{v} \rangle$ are $\alpha$-correlated with the ground truth $\langle X, v^* \rangle$ when $X \sim N(0, \Sigma)$. Moreover, we focus first on the existential question (rather than the algorithmic question of finding the dictionary). We will return to the setting of Theorems 1.1 and 1.2 later. For now, in the weak learning setting, a good dictionary (when the covariate distribution is $N(0, \Sigma)$) is one that satisfies the following covering property, but is not too large:

**Definition 2.1.** Let $\Sigma \in \mathbb{R}^{n \times n}$ be a positive semi-definite matrix and let $t, \alpha > 0$. A set $\{D_1, \ldots, D_N\} \subseteq \mathbb{R}^n$ is a $(t, \alpha)$-dictionary for $\Sigma$ if for every $t$-sparse $v \in \mathbb{R}^n$, there is some $i \in [N]$ with
$$|\langle v, D_i \rangle_\Sigma| \geq \alpha \|v\|_\Sigma \|D_i\|_\Sigma ,$$
where we define $\langle x, y \rangle_\Sigma := x^\top \Sigma y$ and $\|x\|_\Sigma^2 := x^\top \Sigma x$ for any $x, y \in \mathbb{R}^n$. Let $\mathcal{N}_{t,\alpha}(\Sigma)$ be the size of the smallest $(t, \alpha)$-dictionary.

The relevance of the covering number $\mathcal{N}_{t,\alpha}(\Sigma)$ is quite simple: given a $(t, \alpha)$-dictionary $\mathcal{D}$ for $\Sigma$, and given samples $(X_i, y_i)_{i=1}^m$, the weak learning algorithm can simply output the vector $\hat{v} \in \mathcal{D}$ that maximizes the empirical correlation between the predictions $\langle X_i, \hat{v} \rangle$ and the responses $y_i$. So long as there are enough samples for empirical correlations to concentrate, Definition 2.1 guarantees success. Formally, allowing for preprocessing time to compute the dictionary, $O(\alpha)$-weak learning is possible in time $\mathcal{N}_{t,\alpha}(\Sigma) \cdot \text{poly}(n)$, with $O(\alpha^{-2} \log \mathcal{N}_{t,\alpha}(\Sigma))$ samples (Proposition A.5).

Hypothetically, bounding $\mathcal{N}_{t,\alpha}(\Sigma)$ may not be *necessary* to develop an efficient sparse linear regression algorithm. However, all assumptions on $\Sigma$ that are currently known to enable efficient sparse linear regression also immediately imply bounds on $\mathcal{N}_{t,\alpha}$ (see Appendix G). For example, when $\Sigma$ is well-conditioned, the standard basis is a good dictionary of size $n$ (Fact A.4).

In contrast, the only known bounds for arbitrary $\Sigma$ (until the present work) are $\mathcal{N}_{t,1/\sqrt{t}}(\Sigma) \leq t \cdot \binom{n}{t}$ (the brute-force dictionary, which includes a $\Sigma$-orthonormal basis for every set of $t$ covariates) and $\mathcal{N}_{t,1/\sqrt{n}}(\Sigma) \leq n$ (a $\Sigma$-orthonormal basis for all $n$ covariates, which doesn't take advantage of sparsity and corresponds to algorithms such as Ordinary Least Squares). Thus, the following basic question – when can we improve upon these trivial bounds – seems central to understanding when brute-force search can be avoided in sparse linear regression:

**Question 2.2.** *How large is $\mathcal{N}_{t,\alpha}(\Sigma)$ for an arbitrary positive semi-definite $\Sigma \in \mathbb{R}^{n \times n}$? Are there natural families of ill-conditioned $\Sigma$ (and functions $f, g$) for which $\mathcal{N}_{t,1/f(t)}(\Sigma) \leq g(t) \cdot \text{poly}(n)$?*

## 2.2 Constructing a good dictionary when $\Sigma$ has few small eigenvalues

We now address Question 2.2 in the setting where $\Sigma$ has a small number of eigenvalues that are much smaller than $\lambda_n$. In this setting, the standard basis may not be a good dictionary. For example, if two covariates are highly correlated, their difference may not be correlated with any of them. Nonetheless, we can prove the following covering number bound:

**Theorem 2.3.** *Let $n, t, d \in \mathbb{N}$. Let $\Sigma \in \mathbb{R}^{n \times n}$ be a positive semi-definite matrix with eigenvalues $\lambda_1 \leq \cdots \leq \lambda_n$. Then $\mathcal{N}_{t,\alpha}(\Sigma) \leq t(7t)^{2t^2+t} d^t + n$, where $\alpha = \frac{1}{7\sqrt{t}} \sqrt{\lambda_{d+1}/\lambda_n}$.*

In particular, when $t = O(1)$ and $\Sigma$ is well-conditioned except for $O(1)$ outliers $\lambda_1, \ldots, \lambda_d$, we get a linear-size dictionary just as in the case where $\Sigma$ is well-conditioned. In fact, the desired $(t, \alpha)$-dictionary can be constructed efficiently. Our key lemma shows that when $\Sigma$ has few small eigenvalues, there is a small subset of covariates that "causes" all of the sparse approximate dependencies – in the sense that the $\ell_2$ norm of any sparse vector, *excluding* the mass on the subset, can be upper bounded in terms of the $\Sigma$-norm of the vector. Moreover, there is an efficient algorithm that finds a superset of these covariates. Formally, we prove the following:

**Lemma 2.4.** *Let $n, t, d \in \mathbb{N}$. Let $\Sigma \in \mathbb{R}^{n \times n}$ be a positive semi-definite matrix with eigenvalues $\lambda_1 \leq \cdots \leq \lambda_n$. Given $\Sigma$, $d$, and $t$, there is a polynomial-time algorithm* `IterativePeeling()` *producing a set $S \subseteq [n]$ with the following guarantees:*

*(a) For every $t$-sparse $v \in \mathbb{R}^n$, it holds that $\left\| v_{[n] \setminus S} \right\|_2 \leq 3\lambda_{d+1}^{-1/2} \left\| v \right\|_\Sigma$.*

*(b) $|S| \leq (7t)^{2t+1} d$.*

Once this set $S$ has been found, the dictionary is simply the standard basis $\{e_1, \ldots, e_n\}$, together with a $\Sigma$-orthonormal basis for every set of $t$ covariates in $S$. By guarantee (a), we can prove that every $t$-sparse vector correlates with some element of this dictionary under the $\Sigma$-inner product. By guarantee (b), the dictionary is much smaller than the brute-force dictionary that contains a basis for all $\binom{n}{t}$ sets of $t$ covariates. Together, this gives an algorithmic proof for Theorem 2.3.

**Intuition for** `IterativePeeling()`**.** We compute the set $S$ via a new iterative method which leverages knowledge of the small eigenspaces of $\Sigma$. See Algorithm 1 for the pseudocode. To compute $S$, the algorithm `IterativePeeling()` first computes the orthogonal projection matrix $P$ that projects onto the subspace spanned by the top $n - d$ eigenvectors of $\Sigma$. Starting with the set of coordinates that correlate with $\ker(P)$, the procedure then iteratively grows $S$ in such a way that at each step, a new participant of each approximate sparse dependency is discovered, but $S$ does not become too much larger.

The intuition is as follows: as a preliminary attempt, we could identify all $O(d)$ coordinates that correlate (with respect to the standard inner product) with the lowest $d$ eigenspaces of $\Sigma$. If e.g. the covariates have a sparse dependency
$$X_1 + X_2 = 0,$$
then $\ker \Sigma$ contains the vector $e_1 + e_2$, so the coordinates $\{e_1, e_2\}$ will be correctly discovered. Unfortunately, if $\Sigma$ contains a more complex sparse dependency such as
$$\epsilon^{-1}(X_1 - X_2) - X_3 - X_4 = 0$$
where $\epsilon > 0$ is very small, then this heuristic will discover $\{e_1, e_2\}$ but miss $\{e_3, e_4\}$. For this example, the solution is to notice that $e_3$ and $e_4$ *do* correlate with the subspace spanned by $\ker(\Sigma) \cup \{e_1, e_2\}$ (which contains $e_3 + e_4$). In general, if $S$ is the set of coordinates discovered thus far, then by finding basis vectors that correlate with an appropriate subspace (of dimension at most $|S|$), we can efficiently augment $S$ with at least one new coordinate from each $t$-sparse approximate dependency, without making $S$ bigger by more than a factor of $O(t)$. Iterating this augmentation $t$ times therefore provably identifies all problematic coordinates.

To formalize this intuition, the following lemma will be needed to bound how much $S$ grows at each iteration; it shows that the number of coordinates that correlate with a low-dimensional subspace is not too large (proof deferred to Appendix B):

**Lemma 2.5.** *Let $V \subseteq \mathbb{R}^n$ be a subspace with $d := \dim V$. For some $\alpha > 0$ define*
$$S = \left\{ i \in [n] : \sup_{x \in V \setminus \{0\}} \frac{x_i}{\|x\|_2} \geq \alpha \right\}.$$
*Then $|S| \leq d/\alpha^2$. Moreover, given a set of vectors that span $V$, we can compute $S$ in time $\mathrm{poly}(n)$.*

We also define the set of vectors $v$ that have unusually large norm outside a set $S$, compared to $\sqrt{v^\top P v}$, which is the distance from $v$ to the subspace spanned by the bottom $d$ eigenvectors of $\Sigma$:

**Definition 2.6.** For any matrix $P \in \mathbb{R}^{n \times n}$ and subset $S \subseteq [n]$, define $\mathcal{W}_{P,S} := \{v \in \mathbb{R}^n : \|v_{S^c}\|_2 > 3\sqrt{v^\top P v}\}$.

We then formalize the guarantee of each iteration of `IterativePeeling()` as follows:

**Lemma 2.7.** *Let $n, t \in \mathbb{N}$ and let $P : n \times n$ be an orthogonal projection matrix. Suppose $\tau \geq 1$ and $K \subseteq [n]$ satisfy*

(a) *$P_{ii} \geq 1 - 1/(9t^2)$ for all $i \notin K$,*

(b) *$|\operatorname{supp}(v) \setminus K| \leq \tau$ for every $v \in B_0(t) \cap \mathcal{W}_{P,K}$.*

*Then there exists a set $\mathcal{I}_P(K)$ with $|\mathcal{I}_P(K)| \leq 36t^2|K|$ such that*
$$|\operatorname{supp}(v) \setminus (\mathcal{I}_P(K) \cup K)| \leq \tau - 1$$
*for all $v \in B_0(t) \cap \mathcal{W}_{P,K}$. Moreover, given $P$, $K$, and $t$, we can compute $\mathcal{I}_P(K)$ in time $\operatorname{poly}(n)$.*

**Proof sketch.** We define the set
$$\mathcal{I}_P(K) := \left\{ a \in [n] \setminus K : \sup_{x \in \operatorname{span}\{Pe_i : i \in K\} \setminus \{0\}} \frac{|x_a|}{\|x\|_2} \geq 1/(6t) \right\}.$$

It is clear from Lemma B.2 (applied with parameters $V := \operatorname{span}\{Pe_i : i \in K\}$ and $\alpha := 1/(6t)$) that $|\mathcal{I}_P(K)| \leq 36t^2|K|$, and that $\mathcal{I}_P(K)$ can be computed in time $\operatorname{poly}(n)$. It remains to show that $|\mathcal{G}_P(v) \setminus (\mathcal{I}_P(K) \cup K)| \leq \tau - 1$ for all $v \in B_0(t)$.

Consider any $v \in B_0(t) \cap \mathcal{W}_{P,K}$. Then $\|v_{K^c}\|_2 > 3\|Pv\|_2$. It's sufficient to show that $\mathcal{I}_P(K)$ contains some $j \in \operatorname{supp}(v) \setminus K$, i.e. that there is some $j \in \operatorname{supp}(v) \setminus K$ such that $e_j$ correlates with $\operatorname{span}\{P_i : i \in K\}$. We accomplish this by showing that $v_{K^c}$ correlates with $Pv_K = \sum_{i \in K} v_i P_i$.

At a high level, the reason for this is that $v_{K^c}$ is close to $Pv_{K^c}$ (since $P_i \approx e_i$ for $i \in K^c$), and $Pv = Pv_K + Pv_{K^c}$ is much smaller than $Pv_{K^c} \approx v_{K^c}$, so $Pv_K$ and $Pv_{K^c}$ must be highly correlated. See Appendix B for the full proof. ∎

We can now complete the proof of Lemma 2.4 by repeatedly invoking Lemma B.4.

**Proof of Lemma 2.4.** Let $\Sigma = \sum_{i=1}^n \lambda_i u_i u_i^\top$ be the eigendecomposition of $\Sigma$, and let $P := \sum_{i=d+1}^n u_i u_i^\top$ be the projection onto the top $n - d$ eigenspaces of $\Sigma$. Set $K_t = \{i \in [n] : P_{ii} < 1 - 1/(9t^2)\}$. Because $\operatorname{tr}(P) = n - d$ and $P_{ii} \leq 1$ for all $i \in [n]$, it must be that $|K_t| \leq 9t^2 d$. Also, for any $v \in B_0(t) \cap \mathcal{W}_{P,K_t}$ we have trivially by $t$-sparsity that $|\operatorname{supp}(v) \setminus K_t| \leq t$.

Define $K_{t-1}$ to be $K_t \cup \mathcal{I}_P(K_t)$ where $\mathcal{I}_P(K_t)$ is as defined in Lemma B.4; we have the guarantees that $|K_{t-1}| \leq (1 + 36t^2)|K_t|$ and $|\mathcal{G}_P(v) \setminus K_t| \leq t - 1$ for all $v \in B_0(t) \cap \mathcal{W}_{P,K_t}$. Since $K_{t-1} \supseteq K_t$, it holds that $\mathcal{W}_{P,K_{t-1}} \subseteq \mathcal{W}_{P,K_t}$, and thus $|\mathcal{G}_P(v) \setminus K_t| \leq t - 1$ for all $v \in B_0(t) \cap \mathcal{W}_{P,K_{t-1}}$. Moreover, since $K_{t-1} \supseteq K_t$, it obviously holds that $P_{ii} \geq 1 - 1/(9t^2)$ for all $i \notin K_{t-1}$. This means we can apply Lemma B.4 with $\tau := t - 1$ and $K := K_{t-1}$ and so iteratively define sets $K_{t-2} \subseteq \cdots \subseteq K_1 \subseteq K_0 \subseteq [n]$ in the same way. In the end, we obtain the set $K_0 \subseteq [n]$ with $|K_0| \leq 9t^2 d(1 + 36t^2)^t$ and $\operatorname{supp}(v) \subseteq K_0$ for all $v \in B_0(t) \cap \mathcal{W}_{P,K_0}$. The latter guarantee means that in fact $B_0(t) \cap \mathcal{W}_{P,K_0} = \emptyset$. So for any $t$-sparse $v \in \mathbb{R}^n$ it holds that
$$\left\| v_{K_0^c} \right\|_2 \leq 3\sqrt{v^\top P v} \leq 3\lambda_{d+1}^{-1/2} \sqrt{v^\top \Sigma v}$$
where the last inequality holds since $\lambda_{d+1} P \preceq \Sigma$. ∎

## 2.3 Beyond weak learning

So far, we have sketched a proof that if $\Sigma$ has few outlier eigenvalues, then there is an efficient algorithm to compute a good dictionary (as in Theorem 2.3). This gives an efficient $\alpha$-weak learning algorithm (via Proposition A.5). However, our ultimate goal is to find a regressor $\hat{v}$ with prediction error going to 0 as the number of samples increases. Definition 2.1 is not strong enough to ensure this.[8] However, it turns out that the dictionary constructed in Theorem 2.3 in fact satisfies a stronger guarantee[9] that *is* sufficient to achieve vanishing prediction error:

---

[8]Moreover, standard notions of boosting weak learners (e.g. in distribution-free classification) do not apply in this setting.

[9]See Lemma A.3 for a proof that the $\ell_1$-representation property implies the $(t, \alpha)$-dictionary property.

**Definition 2.8.** Let $\Sigma \in \mathbb{R}^{n \times n}$ be a positive semi-definite matrix and let $t, B > 0$. A set $\{D_1, \ldots, D_N\} \subseteq \mathbb{R}^n$ is a $(t, B)$-$\ell_1$-representation for $\Sigma$ if for any $t$-sparse $v \in \mathbb{R}^n$ there is some $\alpha \in \mathbb{R}^N$ with $v = \sum_{i=1}^N \alpha_i D_i$ and $\sum_{i=1}^N |\alpha_i| \cdot \|D_i\|_\Sigma \leq B \cdot \|v\|_\Sigma$.

With this definition in hand, we can actually prove the following strengthening of Theorem 2.3:

**Lemma 2.9.** *Let* $n, t, d \in \mathbb{N}$. *Let* $\Sigma \in \mathbb{R}^{n \times n}$ *be a positive semi-definite matrix with eigenvalues* $\lambda_1 \leq \cdots \leq \lambda_n$. *Then* $\Sigma$ *has a* $(t, 7\sqrt{t}\sqrt{\lambda_n/\lambda_{d+1}})$-$\ell_1$-*representation* $\mathcal{D}$ *of size at most* $n + t(7t)^{2t^2+t}d^t$. *Moreover,* $\mathcal{D}$ *can be computed in time* $t^{O(t^2)}d^t \operatorname{poly}(n)$.

**Proof sketch.** Let $S$ be the output of `IterativePeeling`$(\Sigma, d, t)$. The dictionary $\mathcal{D}$ consists of the standard basis, together with a $\Sigma$-orthogonal basis for each set of $t$ coordinates from $S$. The bound on $|\mathcal{D}|$ comes from the guarantee $|S| \leq (7t)^{2t+1}d$. For any $t$-sparse vector $v \in \mathbb{R}^n$, we know that $v_{S^c}$ is efficiently represented by the standard basis (because Theorem B.1 guarantees that $\|v_{S^c}\|_2 \leq O(\lambda_{d+1}^{-1/2}\|v\|_\Sigma)$), and $v_S$ is efficiently represented by one of the $\Sigma$-orthonormal bases. See Appendix B for the full proof. ∎

Why is the above guarantee useful? If each $D_i$ is normalized to unit $\Sigma$-norm, then the condition of $(t, B)$-$\ell_1$-representability is equivalent to $\|\alpha\|_1 \leq B \cdot \|v\|_\Sigma$. That is, with respect to the new set of features, the regressor $\alpha$ has bounded $\ell_1$ norm. Thus, if we apply the Lasso with a set of features that is a $(t, B)$-$\ell_1$-representation for $\Sigma$, then standard "slow rate" guarantees hold (proof in Section A):

**Proposition 2.10.** *Let* $n, m, N, t \in \mathbb{N}$ *and* $B > 0$. *Let* $\Sigma \in \mathbb{R}^{n \times n}$ *be a positive semi-definite matrix and let* $\mathcal{D}$ *be a* $(t, B)$-$\ell_1$-*representation of size* $N$ *for* $\Sigma$, *normalized so that* $\|v\|_\Sigma = 1$ *for all* $v \in \mathcal{D}$. *Fix a* $t$-sparse vector $v^* \in \mathbb{R}^n$, *let* $X_1, \ldots, X_m \sim N(0, \Sigma)$ *be independent and let* $y_i = \langle X_i, v^* \rangle + \xi_i$ *where* $\xi_i \sim N(0, \sigma^2)$. *For any* $R > 0$, *define*

$$\hat{w} \in \operatorname*{argmin}_{w \in \mathbb{R}^N : \|w\|_1 \leq BR} \|\mathbb{X}Dw - y\|_2^2$$

*where* $D \in \mathbb{R}^{n \times N}$ *is the matrix with columns comprising the elements of* $\mathcal{D}$, *and* $\mathbb{X} \in \mathbb{R}^{m \times n}$ *is the matrix with rows* $X_1, \ldots, X_m$. *So long as* $m = \Omega(\log(n/\delta))$ *and* $\|w^*\|_\Sigma \in [R/2, R]$, *it holds with probability at least* $1 - \delta$ *that*

$$\|D\hat{w} - w^*\|_\Sigma^2 = O\left(B\|w^*\|_\Sigma \sigma\sqrt{\frac{\log(2n/\delta)}{m}} + \frac{\sigma^2 \log(4/\delta)}{m} + \frac{B^2\|w^*\|_\Sigma^2 \log(n)}{m}\right).$$

Combining Proposition 2.10 with Lemma 2.9 shows that there is an algorithm with time complexity $t^{O(t^2)}d^t \operatorname{poly}(n)$ and sample complexity $O(\operatorname{poly}(t)(\lambda_n/\lambda_{d+1})\log(n)\log(d))$ for finding a regressor with squared prediction error $o(\sigma^2 + \|v^*\|_\Sigma^2)$. This is a simplified version of Theorem 1.2. The full proof involves additional technical details (e.g. more careful analysis to take care of large eigenvalues, and to avoid needing an estimate $R$ for $\|w^*\|_\Sigma$) but the above exposition contains the central ideas. Theorem 1.1 similarly computes the set $S$ from Lemma 2.4 but uses it to construct a different dictionary: the standard basis, plus a $\Sigma$-orthonormal basis for $S$.[10] See Appendix C for the full proofs and pseudocode.

## 3    Additional Results

We now return to Question 2.2 and ask whether there are other families of ill-conditioned $\Sigma$ for which we can prove non-trivial bounds on $\mathcal{N}_{t,\alpha}(\Sigma)$.

First, we ask what can be shown for *arbitrary* covariance matrices. We prove that *every* covariance matrix $\Sigma$ satisfies a non-trivial bound $\mathcal{N}_{t,1/O(t^{3/2}\log n)}(\Sigma) \leq O(n^{t-1/2})$. In fact, building on tools from computational geometry, we show the stronger result that $\Sigma$ has a $(t, O(t^{3/2}\log n))$-$\ell_1$-representation that of size $O(n^{t-1/2})$, that is computable from samples in time $\tilde{O}(n^{t-\Omega(1/t)})$ for

---

[10]More precisely, the algorithm just skips regularizing $S$, which is morally equivalent. As it is simpler to implement, that is shown in Algorithm 1, and analyzed for the proofs.

any constant $t > 1$ (Theorem D.5). As a corollary, we provide the first sparse linear regression algorithm with time complexity that is a polynomial-factor better than brute force, and with near-optimal sample complexity, for any constant $t$ and arbitrary $\Sigma$ (proof in Section D):

**Theorem 3.1.** *Let $n, m, t, B \in \mathbb{N}$ and $\sigma > 0$, and let $\Sigma \in \mathbb{R}^{n \times n}$ be a positive-definite matrix. Let $w^* \in \mathbb{R}^n$ be $t$-sparse, and suppose $\|w^*\|_\Sigma \in [B/2, B]$. Suppose $m \geq \Omega(t \log n)$. Let $(X_i, y_i)_{i=1}^m$ be independent samples where $X_i \sim N(0, \Sigma)$ and $y_i = \langle X_i, w^* \rangle + N(0, \sigma^2)$. Then there is an $O(m^2 n^{t-1/2} + n^{t-\Omega(1/t)} \log^{O(t)} n)$-time algorithm that, given $(X_i, y_i)_{i=1}^m$, $B$, and $\sigma^2$, produces an estimate $\hat{w} \in \mathbb{R}^n$ satisfying, with probability $1 - o(1)$,*

$$\|\hat{w} - w^*\|_\Sigma^2 \leq \tilde{O}\left( \frac{\sigma^2}{\sqrt{m}} + \frac{\sigma \|w^*\|_\Sigma t^{3/2}}{\sqrt{m}} + \frac{\|w^*\|_\Sigma^2 t^3}{m} \right).$$

Second, one goal is to improve "sample complexity" (i.e. obtain $\alpha$ without dependence on condition number) without paying too much in "time complexity" (i.e. retain bounds on $\mathcal{N}_{t,\alpha}$ that are better than $n^t$). To this end, we prove that the dependence on $\kappa$ in the correlation level (see Fact A.4) can actually be replaced by dependence on $\kappa$ in the dictionary size (proof in Appendix E):

**Theorem 3.2.** *Let $n, t \in \mathbb{N}$. Let $\Sigma \in \mathbb{R}^{n \times n}$ be a positive-definite matrix with condition number $\kappa$. Then $\mathcal{N}_{t,1/3^{t+1}}(\Sigma) \leq 2^{O(t^2)} \kappa^{2t+1} \cdot n$.*

In particular, for any constant $t = 1/\epsilon$, our result shows that there is a nearly-linear size dictionary with *constant* correlations even for covariance matrices with *polynomially-large* condition number $\kappa \leq n^{\epsilon/100}$. While we are not currently aware of an efficient algorithm for computing the dictionary, the above bound nonetheless raises the interesting possibility that there may be a sample-efficient and computationally-efficient weak learning algorithm under a super-constant bound on $\kappa$.

# 4 Related work

**Dealing with correlated covariates.**    There is considerable work on improving the performance of Lasso in situations where some clusters of covariates are highly correlated [47, 19, 2, 42, 21, 12, 27]. These methods can work well for two-sparse dependencies, but generally do not work as well for higher-order dependencies — hence they cannot be used to prove our main result. The approach of [2] is perhaps the closest in spirit to ours. They perform agglomerative clustering of correlated covariates, orthonormalize the clusters with respect to $\Sigma$, and apply Lasso (or solve an essentially equivalent group Lasso problem). This method fails, for example, when there is a single three-sparse dependency, and the remaining covariates have some mild correlations. Depending on the correlation threshold, their method will either aggressively merge all covariates into a single cluster, or fail to merge the dependent covariates.

**Feature adaptation and preconditioning.**    Generalizations of Lasso via a preliminary change-of-basis (or explicitly altering the regularization term) have been studied in the past, but largely not to solve sparse linear regression per se; instead the goal has been using $\ell_1$ regularization to encourage other structural properties such as piecewise continuity (e.g. in the "fused lasso", see [35, 36, 20, 8] for some more examples). An exception is recent work showing that a "sparse preconditioning" step can enable Lasso to be statistically efficient for sparse linear regression when the covariates have a certain Markovian structure [23]. Our notion of feature adaptation via dictionaries generalizes sparse preconditioning, which corresponds to choosing a non-standard basis in which $\Sigma$ becomes well-conditioned and the sparsity of the signal is preserved.

**Statistical query (SQ) model; sparse halfspaces.**    From the complexity standpoint, $\mathcal{N}_{t,\alpha}(\Sigma)$ is a covering number and therefore closely corresponds to a packing number $\mathcal{P}_{t,\alpha}(\Sigma)$ (see Section A.1 for the definition). This packing number is essentially the *(correlational) statistical dimension*, which governs the complexity of sparse linear regression with covariates from $N(0, \Sigma)$ in the (correlational) SQ model (see e.g. [14] for exposition of this model). Whereas strong $n^{\Omega(t)}$ SQ lower bounds are known for related problems such as sparse parities with noise [29], no non-trivial (i.e. super-linear) lower bounds are known for sparse linear regression. Relatedly, in a COLT open problem, Feldman asked whether any non-trivial bounds can be shown for the complexity of weak learning sparse halfspaces in the SQ model [11]. Our results also yield improved bounds for weakly SQ-learning sparse halfspaces over certain families of multivariate Gaussian distributions.

**Improving brute-force for arbitrary** $\Sigma$.   Several prior works have suggested improvements on brute-force search for variants of $t$-sparse linear regression [18, 16, 31, 6]. However, all of these have limitations preventing their application to the general setting we address in Theorem 3.1. Specifically, [18] requires $\Omega(n^t)$ preprocessing time on the covariates; [16, 31] require noiseless responses; and [6] has time complexity scaling with $\log^m n$ (since our random-design setting necessitates $m \geq \Omega(t \log n)$, their algorithm has time complexity much larger than $n^t$).

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

# A Preliminaries

Throughout, we use the following standard notation. For positive integers $n, m \in \mathbb{N}$, we write $A : m \times n$ to denote a matrix with $m$ rows, $n$ columns, and real-valued entries. The standard inner product on $\mathbb{R}^n$ is denoted $\langle u, v \rangle := u^\top v$. For a positive semi-definite matrix $\Sigma : n \times n$ we define the $\Sigma$-inner product on $\mathbb{R}^n$ by $\langle u, v \rangle_\Sigma := u^\top \Sigma v$ and the $\Sigma$-norm by $\|u\|_\Sigma = \sqrt{\langle u, u \rangle_\Sigma}$. For $n \in \mathbb{N}$ (made clear by context) we let $e_1, \ldots, e_n \in \mathbb{R}^n$ be the standard basis vectors $e_i(j) := \mathbb{1}[j = i]$. For a vector $v \in \mathbb{R}^n$ and set $S \subseteq [n]$ we write $v_S$ to denote the restriction of $v$ to coordinates in $S$. For symmetric matrices $A, B : n \times n$ we write $A \preceq B$ to denote that $B - A$ is positive semi-definite.

## A.1 Covering, packing, and $\ell_1$-representability

We previously defined the covering number of $t$-sparse vectors with respect to a covariance matrix $\Sigma$. We next define the packing number (i.e. correlational statistical dimension) and $\ell_1$-representability, and discuss the connections between these quantities as well as their algorithmic implications.

**Definition A.1.** Let $\Sigma : n \times n$ be a positive semi-definite matrix and let $t, \alpha > 0$. A set $\{v_1, \ldots, v_N\} \subseteq \mathbb{R}^n$ is a $(t, \alpha)$-packing for $\Sigma$ if every $v_i$ is $t$-sparse, and

$$|\langle v_i, v_j \rangle_\Sigma| < \alpha \|v_i\|_\Sigma \|v_j\|_\Sigma$$

for all $i, j \in [N]$ with $i \neq j$. The (correlational) statistical dimension of $t$-sparse vectors with maximum correlation $\alpha$, under the $\Sigma$-inner product, is denoted $\mathcal{P}_{t,\alpha}(\Sigma)$ and defined as the size of the largest $(t, \alpha)$-packing.

We will make use of the following connections between packing, covering, and $\ell_1$-representability.

**Lemma A.2** (Covering $\Leftrightarrow$ packing). *For any positive semi-definite matrix $\Sigma : n \times n$ and $t, \alpha > 0$, it holds that $(\alpha^2/3)\mathcal{P}_{t,\alpha^2/2}(\Sigma) \leq \mathcal{N}_{t,\alpha}(\Sigma) \leq \mathcal{P}_{t,\alpha}(\Sigma)$.*

*Proof.* **First inequality.** Let $\{D_1, \ldots, D_N\}$ be any maximum-size $(t, \alpha^2/2)$-packing. Since the $D_i$'s are all $t$-sparse, each must be correlated with some element of a $(t, \alpha)$-dictionary. Thus, it suffices to show that for any $v \in \mathbb{R}^n$, the set $S(v) := \{i \in [N] : |\langle D_i, v \rangle_\Sigma| \geq \alpha \|D_i\|_\Sigma \|v\|_\Sigma\}$ has size $|S(v)| \leq 3/\alpha^2$. Indeed, for any $i, j \in S(v)$ with $i \neq j$, we have by the definition of a packing that

$$\left\langle D_i - \frac{\langle D_i, v \rangle_\Sigma}{\|v\|_\Sigma^2} v, D_j - \frac{\langle D_j, v \rangle_\Sigma}{\|v\|_\Sigma^2} v \right\rangle_\Sigma = \langle D_i, D_j \rangle - \frac{\langle D_i, v \rangle_\Sigma \langle D_j, v \rangle_\Sigma}{\|v\|_\Sigma^2}$$

$$\leq -\frac{\alpha^2}{2} \|D_i\|_\Sigma \|D_j\|_\Sigma.$$

For each $i \in S(v)$ define $R_i = D_i - \langle D_i, v \rangle_\Sigma v / \|v\|_\Sigma^2$. Then

$$0 \leq \left\| \sum_{i \in S(v)} \frac{R_i}{\|R_i\|_\Sigma} \right\|_\Sigma^2 = |S(v)| + \sum_{i,j \in S(v): i \neq j} \frac{\langle R_i, R_j \rangle_\Sigma}{\|R_i\|_\Sigma \|R_j\|_\Sigma} \leq |S(v)| - |S(v)|(|S(v)| - 1) \cdot \frac{\alpha^2}{2}$$

where the last inequality uses the bound $\|R_i\|_\Sigma \leq \|D_i\|_\Sigma$. Rearranging gives $|S(v)| \leq 1 + (2/\alpha^2)$.

**Second inequality.** Let $\{D_1, \ldots, D_N\}$ be any maximal $(t, \alpha)$-packing. Then for any $t$-sparse $v \in \mathbb{R}^n$, maximality implies that there must be some $i \in [N]$ with $|\langle D_i, v \rangle_\Sigma| \geq \alpha \|D_i\|_\Sigma \|v\|_\Sigma$. So $\{D_1, \ldots, D_N\}$ is also a $(t, \alpha)$-dictionary. $\square$

**Lemma A.3** ($\ell_1$-representation $\implies$ covering). *Let $\Sigma : n \times n$ be a positive semi-definite matrix and let $t, B > 0$. If $\{D_1, \ldots, D_N\} \subseteq \mathbb{R}^n$ is a $(t, B)$-$\ell_1$-representation for $\Sigma$, then it is also a $(t, 1/B)$-dictionary for $\Sigma$.*

*Proof.* Pick any $t$-sparse $v \in \mathbb{R}^n$. By $\ell_1$-representability, there is some $\alpha \in \mathbb{R}^N$ with $v = \sum_{i=1}^{N} \alpha_i D_i$ and $\sum_{i=1}^{N} |\alpha_i| \cdot \|D_i\|_\Sigma \leq B \cdot \|v\|_\Sigma$. Hence

$$\|v\|_\Sigma^2 = \sum_{i=1}^{N} \alpha_i \langle v, D_i \rangle_\Sigma$$

$$\leq \sum_{i=1}^{N} |\alpha_i| \, \|v\|_\Sigma \, \|D_i\|_\Sigma \cdot \max_{j \in [N]} \frac{|\langle v, D_j \rangle_\Sigma|}{\|v\|_\Sigma \, \|D_j\|_\Sigma}$$

$$\leq B \, \|v\|_\Sigma^2 \cdot \max_{j \in [N]} \frac{|\langle v, D_j \rangle_\Sigma|}{\|v\|_\Sigma \, \|D_j\|_\Sigma}$$

and thus $\max_{j \in [N]} \frac{|\langle v, D_j \rangle_\Sigma|}{\|v\|_\Sigma \|D_j\|_\Sigma} \geq 1/B$. $\qquad \square$

We can now easily prove that the standard basis is a good dictionary for well-conditioned $\Sigma$.

**Fact A.4.** *Let $\Sigma$ be a positive definite matrix with condition number $\frac{\lambda_{max}(\Sigma)}{\lambda_{min}(\Sigma)} \leq \kappa$. Under the $\Sigma$-inner product, every $t$-sparse vector is at least $1/(\sqrt{\kappa t})$-correlated with some standard basis vector.*

*Proof.* By Lemma A.3, it suffices to show that the standard basis $\{e_1, \ldots, e_n\}$ is a $(t, \sqrt{\kappa t})$-$\ell_1$-representation for $\Sigma$. Indeed, for any $t$-sparse $v \in \mathbb{R}^n$,

$$\sum_{i=1}^{n} |v_i| \cdot \|e_i\|_\Sigma \leq \sum_{i=1}^{n} |v_i| \cdot \sqrt{\lambda_{\max}(\Sigma)} \, \|e_i\|_2 = \sqrt{\lambda_{\max}(\Sigma)} \, \|v\|_1$$

$$\leq \sqrt{\lambda_{\max}(\Sigma)} \sqrt{t} \, \|v\|_2 \leq \sqrt{\frac{\lambda_{\max}(\Sigma)}{\lambda_{\min}(\Sigma)}} \sqrt{t} \, \|v\|_\Sigma$$

as desired. $\qquad \square$

## A.2 Algorithmic implications

An existential proof that $\mathcal{N}_{t,\alpha}(\Sigma)$ is small unfortunately does not in general give an efficient algorithm for constructing a concise dictionary. However, with the caveat that the dictionary must be given to the algorithm as advice, bounds on $\mathcal{N}_{t,\alpha}$ do imply weak learning algorithms with sample complexity $O(\alpha^{-2} \log(n))$:

**Proposition A.5.** *Let $\Sigma : n \times n$ be a positive semi-definite matrix and let $\mathcal{D}$ be a $(t, \alpha)$-dictionary for $\Sigma$, for some $t \in \mathbb{N}$ and $\alpha \in (0, 1)$. For $m \in \mathbb{N}$ and $t$-sparse $v^* \in \mathbb{R}^n$, let $X_1, \ldots, X_m \sim N(0, \Sigma)$ be independent and let $y_i = \langle X_i, v^* \rangle + \xi_i$ where $\xi_i \sim N(0, \sigma^2)$. Define the estimator*

$$\hat{v} = \operatorname*{argmin}_{\substack{v \in \mathcal{D} \\ \beta \in \mathbb{R}}} \|\beta \mathbb{X} v - y\|_2^2$$

*where $\mathbb{X} : m \times n$ is the matrix with rows $X_1, \ldots, X_m$. For any $\delta > 0$, if $m \geq C\alpha^{-2} \log(32|\mathcal{D}|/\delta)$ for a sufficiently large absolute constant $C$, then with probability at least $1 - \delta$,*

$$\left\| \hat{\beta}\hat{w} - w^* \right\|_\Sigma^2 \leq (1 - \alpha^2/4) \|w^*\|_\Sigma^2 + \frac{400\sigma^2 \log(4|\mathcal{D}|/\delta)}{\alpha^2 m}.$$

*Proof.* Since $\mathcal{D}$ is a $(t, \alpha)$-dictionary, we know that there is some $\tilde{v} \in \mathcal{D}$ with $|\langle \tilde{v}, v^* \rangle_\Sigma| \geq \alpha \|\tilde{v}\|_\Sigma \|v^*\|_\Sigma$. We then apply Lemma F.4. $\qquad \square$

The above guarantee is essentially of the form "at least 1% of the signal variance can be explained". Under the $\ell_1$-representability condition, something much stronger is true:

**Proposition A.6.** *Let $n, m, N, t \in \mathbb{N}$ and $B > 0$. Let $\Sigma : n \times n$ be a positive semi-definite matrix and let $\mathcal{D}$ be a $(t, B)$-$\ell_1$-representation of size $N$ for $\Sigma$, normalized so that $\|v\|_\Sigma = 1$ for all $v \in \mathcal{D}$.*

*Fix a t-sparse vector $v^* \in \mathbb{R}^n$, let $X_1, \ldots, X_m \sim N(0, \Sigma)$ be independent and let $y_i = \langle X_i, v^* \rangle + \xi_i$ where $\xi_i \sim N(0, \sigma^2)$. For any $R > 0$, define*

$$\hat{w} \in \operatorname*{argmin}_{w \in \mathbb{R}^N : \|w\|_1 \leq BR} \|\mathbb{X}Dw - y\|_2^2$$

*where $D : n \times N$ is the matrix with columns comprising the elements of $\mathcal{D}$, and $\mathbb{X} : m \times n$ is the matrix with rows $X_1, \ldots, X_m$. So long as $m = \Omega(\log(n/\delta))$ and $R \geq \|v^*\|_\Sigma$, it holds with probability at least $1 - \delta$ that*

$$\|D\hat{w} - w^*\|_\Sigma^2 = O\left( BR\sigma\sqrt{\frac{\log(2n/\delta)}{m}} + \frac{\sigma^2 \log(4/\delta)}{m} + \frac{B^2 R^2 \log(n)}{m} \right).$$

*Proof.* By $\ell_1$-representability and normalization of $\mathcal{D}$, there is some $w^* \in \mathbb{R}^N$ such that $v^* = Dw^*$ and $\|w^*\|_1 \leq B \|v^*\|_\Sigma \leq BR$. Let $\Gamma = D^\top \Sigma D$. Also, by normalization, $\max_i \Gamma_{ii} = 1$. Thus, we can apply standard "slow rate" Lasso guarantees to the samples $(D^\top X_i, y_i)_{i=1}^m$ to get the claimed bound (see e.g. Theorem 14 of [22]). $\qquad\square$

### A.3 Optimizing the Lasso in near-linear time

**Theorem A.7** (see e.g. Corollary 4 and Section 5.3 in [34]). *Let $n, m, B, H, T \in \mathbb{N}$ and $\sigma > 0$. Fix $X_1, \ldots, X_m \in \mathbb{R}^n$ with $\|X_i\|_\infty \leq H$ for all $i$, and fix $w^* \in \mathbb{R}^n$ with $\|w^*\|_1 \leq B$. For $i \in [m]$ define $y_i = \langle X_i, w^* \rangle + \xi_i$ where $\xi_i \sim N(0, \sigma^2)$ are independent random variables. Given $(X_i, y_i)_{i=1}^m$ as well as $B$, $T$, and $\sigma^2$, there is an algorithm* $\mathtt{MirrorDescentLasso}((X_i, y_i)_{i=1}^m, B, T, \sigma^2)$, *which optimizes the Lasso objective via $T$ iterations of mirror descent, that produces an estimate $\hat{w} \in \mathbb{R}^n$ satisfying $\|\hat{w}\|_1 \leq B$ and, with probability $1 - o(1)$,*

$$\frac{1}{m} \|X\hat{w} - y\|_2^2 \leq \frac{1}{m} \|Xw^* - y\|_2^2 + \tilde{O}\left( \frac{H^2 B^2}{T} + \sqrt{\frac{H^2 B^2 \sigma^2}{T}} \right).$$

*Moreover, the time complexity of* $\mathtt{MirrorDescentLasso()}$ *is $\tilde{O}(nmT)$.*

**Theorem A.8.** *Let $n, m, B, H \in \mathbb{N}$ and $\sigma > 0$. Let $\Sigma : n \times n$ be positive semi-definite with $\max_{j \in [n]} \Sigma_{jj} \leq H^2$. Fix $w^* \in \mathbb{R}^n$ with $\|w^*\|_1 \leq B$. Let $(X_i, y_i)_{i=1}^m$ be independent draws where $X_i \sim N(0, \Sigma)$ and $y_i = \langle X_i, w^* \rangle + N(0, \sigma^2)$. Then* $\mathtt{MirrorDescentLasso}((X_i, y_i)_{i=1}^m, B, m, \sigma^2)$ *computes, in time $\tilde{O}(nm^2)$, an estimate $\hat{w}$ satisfying, with probability $1 - o(1)$,*

$$\|\hat{w} - w^*\|_\Sigma^2 \leq \tilde{O}\left( \frac{\sigma^2}{\sqrt{m}} + \frac{\sigma HB}{\sqrt{m}} + \frac{H^2 B^2}{m} \right)$$

*Proof.* Since $\max_j \Sigma_{jj} \leq H$ we have that $\max_i \|X_i\|_\infty \leq O(H \log n)$ with probability $1 - o(1)$. Applying Theorem A.7 with this bound and with $T = m$, we obtain some $\hat{w} \in \mathbb{R}^n$ with $\|\hat{w}\|_1 \leq B$ and, with probability $1 - o(1)$,

$$\frac{1}{m} \|X\hat{w} - y\|_2^2 \leq \frac{1}{m} \|Xw^* - y\|_2^2 + \epsilon$$

where $\epsilon = \tilde{O}(H^2 B^2/m) + \sqrt{H^2 B^2 \sigma^2/m}$. By $\chi^2$-concentration, we have $\frac{1}{m} \|Xw^* - y\|_2^2 \leq \sigma^2(1 + O(1/\sqrt{m}))$ with probability $1 - o(1)$. Thus,

$$\|X\hat{w} - y\|_2 \leq \|Xw^* - y\|_2 + \sqrt{\epsilon m} \leq \sigma\sqrt{m} + O(\sigma m^{1/4}) + \sqrt{\epsilon m}$$

and

$$\|X\hat{w} - y\|_2^2 \leq \|Xw^* - y\|_2^2 + m\epsilon \leq \sigma^2 m + O(\sigma^2 \sqrt{m}) + \epsilon m.$$

Next, since $\sup_{w \in \mathbb{R}^n : \|w\|_1 \leq B} \langle w - w^*, x \rangle \leq 2B \|x\|_\infty \leq O(HB \log n)$ with probability $1 - o(1)$ over $x \sim N(0, \Sigma)$, we can apply Theorem C.1 to get that with probability $1 - o(1)$,

$$\|\hat{w} - w^*\|_\Sigma^2 + \sigma^2 \leq \frac{1 + \tilde{O}(1/\sqrt{m})}{m} (\|X\hat{w} - y\|_2 + \tilde{O}(HB))^2.$$

Substituting the bounds on $\|X\hat{w} - y\|_2$ and $\|X\hat{w} - y\|_2^2$ gives

$$\|\hat{w} - w^*\|_\Sigma^2 + \sigma^2 \le \sigma^2 + O(\sigma^2 m^{-1/2} + \epsilon) + \tilde{O}(\sigma HB m^{-1/2} + HB\sqrt{\epsilon/m}) + \tilde{O}(H^2 B^2/m).$$

Substituting in the value of $\epsilon$ and simplifying, we get

$$\|\hat{w} - w^*\|_\Sigma^2 \le \tilde{O}\left(\frac{\sigma^2}{\sqrt{m}} + \frac{\sigma HB}{\sqrt{m}} + \frac{H^2 B^2}{m}\right)$$

as claimed. $\qquad\square$

## B    Iterative Peeling

In this section we give the complete proof of Lemma 2.4, restated below as Theorem B.1, which describes the guarantees of `IterativePeeling()` (see Algorithm 1). This is a key ingredient in the proofs of Theorems 1.1 and 1.2. We also use it to formally prove Theorem 2.3, as well as Lemma 2.9.

**Theorem B.1.** *Let $n, t, d \in \mathbb{N}$. Let $\Sigma : n \times n$ be a positive semi-definite matrix with eigenvalues $\lambda_1 \le \cdots \le \lambda_n$. Given $\Sigma$, $d$, and $t$, there is a polynomial-time algorithm* `IterativePeeling()` *producing a set $S \subseteq [n]$ with the following guarantees:*

- *For every $t$-sparse $v \in \mathbb{R}^n$, it holds that $\|v_{[n]\setminus S}\|_2 \le 3\lambda_{d+1}^{-1/2}\|v\|_\Sigma$.*

- $|S| \le (7t)^{2t+1}d$.

Essentially, the set $S$ contains every coordinate $i \in [n]$ that "participates" in an approximate sparse dependency, in the sense that there is some sparse linear combination of the covariates with small variance compared to the coefficient on $i$. To compute $S$, the algorithm `IterativePeeling()` first computes the orthogonal projection matrix $P$ that projects onto the subspace spanned by the top $n - d$ eigenvectors of $\Sigma$. Starting with the set of coordinates that correlate with $\ker(P)$, the procedure then iteratively grows $S$ in such a way that at each step, a new participant of each approximate sparse dependency is discovered, but $S$ does not become too much larger.

The following lemma will be needed to bound how much $S$ grows at each iteration:

**Lemma B.2.** *Let $V \subseteq \mathbb{R}^n$ be a subspace with $d := \dim V$. For some $\alpha > 0$ define*

$$S = \left\{i \in [n] : \sup_{x \in V\setminus\{0\}} \frac{x_i}{\|x\|_2} \ge \alpha\right\}.$$

*Then $|S| \le d/\alpha^2$. Moreover, given a set of vectors that span $V$, we can compute $S$ in time $\mathrm{poly}(n)$.*

*Proof.* Let $k := |S|$ and without loss of generality suppose $S = \{1, \ldots, k\}$. Define a matrix $A \in \mathbb{R}^{n \times n}$ as follows. For $1 \le i \le k$ let row $A_i \in V$ be some vector such that $\|A_i\|_2 = 1$ and $A_{ii} \ge \alpha$. For $k + 1 \le i \le n$ let $A_i = 0$. Then $\mathrm{tr}(A) \ge k\alpha$ and $\|A\|_F = \sqrt{k}$. However, $\mathrm{rank}(A) \le d$, so the singular values $\sigma_1 \ge \sigma_2 \ge \cdots \ge \sigma_n \ge 0$ of $A$ satisfy $\sigma_{d+1} = 0$. Thus,

$$k\alpha \le \mathrm{tr}(A) \le \sum_{i=1}^n \sigma_i \le \sqrt{d}\sqrt{\sum_{i=1}^n \sigma_i^2} = \sqrt{d}\|A\|_F = \sqrt{dk}$$

where the second inequality is by e.g. Von Neumann's trace inequality, and the third inequality is by $d$-sparsity of the vector $\sigma$. It follows that $k \le d/\alpha^2$ as claimed.

Let $A$ be the matrix with columns consisting of the given spanning set for $V$. By Gram-Schmidt, we may transform the spanning set into an orthonormal basis for $V$, so that $A$ has $d$ columns, and $A^\top A = I_d$. Fix $i \in [n]$. Then $\sup_{x \in V\setminus\{0\}} x_i/\|x\|_2 \ge \alpha$ if and only if $(Av)_i^2 - \alpha^2\|Av\|_2^2 \ge 0$ for some nonzero $v \in \mathbb{R}^d$. Equivalently, $(Av)_i^2 \ge \alpha^2$ for some unit vector $v$. This is possible if and only if $\|A_i\|_2 \ge \alpha$ (where $A_i$ is the $i$-th row of $A$), which can be checked in polynomial time. $\qquad\square$

For notational convenience, we also define the set $\mathcal{W}_{P,S}$ of vectors $v$ with unusually large norm outside the set $S$.

**Definition B.3.** For any matrix $P : n \times n$ and subset $S \subseteq [n]$, define $\mathcal{W}_{P,S} := \{v \in \mathbb{R}^n : \|v_{S^c}\|_2 > 3\sqrt{v^\top P v}\}$.

We then formalize the guarantee of each iteration of `IterativePeeling()` as follows:

**Lemma B.4.** *Let $n, t \in \mathbb{N}$ and let $P : n \times n$ be an orthogonal projection matrix. Suppose $\tau \geq 1$ and $K \subseteq [n]$ satisfy*

- *(a) $P_{ii} \geq 1 - 1/(9t^2)$ for all $i \notin K$,*

- *(b) $|\operatorname{supp}(v) \setminus K| \leq \tau$ for every $v \in B_0(t) \cap \mathcal{W}_{P,K}$.*

*Then there exists a set $\mathcal{I}_P(K)$ with $|\mathcal{I}_P(K)| \leq 36t^2|K|$ such that*

$$|\operatorname{supp}(v) \setminus (\mathcal{I}_P(K) \cup K)| \leq \tau - 1$$

*for all $v \in B_0(t) \cap \mathcal{W}_{P,K}$. Moreover, given $P$, $K$, and $t$, we can compute $\mathcal{I}_P(K)$ in time $\operatorname{poly}(n)$.*

*Proof.* We define the set

$$\mathcal{I}_P(K) := \left\{ a \in [n] \setminus K : \sup_{x \in \operatorname{span}\{Pe_i : i \in K\} \setminus \{0\}} \frac{|x_a|}{\|x\|_2} \geq 1/(6t) \right\}.$$

It is clear from Lemma B.2 (applied with parameters $V := \operatorname{span}\{Pe_i : i \in K\}$ and $\alpha := 1/(6t)$) that $|\mathcal{I}_P(K)| \leq 36t^2|K|$, and that $\mathcal{I}_P(K)$ can be computed in time $\operatorname{poly}(n)$. It remains to show that $|\operatorname{supp}(v) \setminus (\mathcal{I}_P(K) \cup K)| \leq \tau - 1$ for all $v \in B_0(t) \cap \mathcal{W}_{P,K}$.

Consider any $v \in B_0(t) \cap \mathcal{W}_{P,K}$. Then $\|v_{K^c}\|_2 > 3\sqrt{v^\top P v}$. We have

$$\frac{\|v_{K^c}\|_2^2}{9} > v^\top P v = \|Pv\|_2^2 = \left\| \sum_{i=1}^n v_i P_i \right\|_2^2 \tag{1}$$

where the first equality uses the fact that $P$ is a projection matrix. We also know that

$$\left\| \sum_{i \in [n] \setminus K} v_i (P_i - e_i) \right\|_2 \leq \sum_{i \in [n] \setminus K} |v_i| \, \|P_i - e_i\|_2 \leq \frac{1}{3\sqrt{t}} \|v_{K^c}\|_1 \leq \frac{1}{3} \|v_{K^c}\|_2 \tag{2}$$

by the triangle inequality, the bound $\|P_i - e_i\|_2^2 = (I - P)_{ii} = 1 - P_{ii} \leq 1/(9t)$ (since $i \notin K$), and $t$-sparsity of $v$. Moreover, (2) implies that

$$\left\| \sum_{i \in [n] \setminus K} v_i P_i \right\|_2 \leq \left\| \sum_{i \in [n] \setminus K} v_i (P_i - e_i) \right\|_2 + \|v_{K^c}\|_2 \leq \frac{4}{3} \|v_{K^c}\|_2 . \tag{3}$$

Combining (1) and (3), the triangle inequality gives

$$\left\| \sum_{i \in K} v_i P_i \right\|_2 \leq \left\| \sum_{i \in [n] \setminus K} v_i P_i \right\|_2 + \left\| \sum_{i=1}^n v_i P_i \right\|_2 \leq \frac{5}{3} \|v_{K^c}\|_2 . \tag{4}$$

Next, observe that

$$\frac{\|v_{K^c}\|_2^2}{3} > \left\| \sum_{i=1}^n v_i P_i \right\|_2 \|v_{K^c}\|_2 \qquad \text{(by (1))}$$

$$\geq \left| \left\langle \sum_{i=1}^n v_i P_i, v_{K^c} \right\rangle \right| \qquad \text{(by Cauchy-Schwarz)}$$

$$\geq \left| \left\langle \sum_{i \in [n] \setminus K} v_i P_i, v_{K^c} \right\rangle \right| - \left| \left\langle \sum_{i \in K} v_i P_i, v_{K^c} \right\rangle \right| \qquad \text{(by triangle inequality)}$$

$$\geq \left| \left\langle \sum_{i \in [n] \setminus K} v_i e_i, v_{K^c} \right\rangle \right| - \left| \left\langle \sum_{i \in [n] \setminus K} v_i (P_i - e_i), v_{K^c} \right\rangle \right| - \left| \left\langle \sum_{i \in K} v_i P_i, v_{K^c} \right\rangle \right|$$

(by triangle inequality)

$$\geq \|v_{K^c}\|_2^2 - \left\| \sum_{i \in [n] \setminus K} v_i (P_i - e_i) \right\|_2 \|v_{K^c}\|_2 - \left| \left\langle \sum_{i \in K} v_i P_i, v_{K^c} \right\rangle \right|$$

(by Cauchy-Schwarz)

$$\geq \|v_{K^c}\|_2^2 - \frac{1}{3} \|v_{K^c}\|_2^2 - \left| \left\langle \sum_{i \in K} v_i P_i, v_{K^c} \right\rangle \right|$$

(by (2))

and hence

$$\left| \left\langle \sum_{i \in K} v_i P_i, v_{K^c} \right\rangle \right| > \frac{1}{3} \|v_{K^c}\|_2^2 \geq \frac{1}{5} \|v_{K^c}\|_2 \left\| \sum_{i \in K} v_i P_i \right\|_2$$

where the last inequality is by (4). On the other hand, observe that

$$\left| \left\langle \sum_{i \in K} v_i P_i, v_{K^c} \right\rangle \right| \leq \sum_{j \in [n] \setminus K} |v_j| \cdot \left| \left\langle \sum_{i \in K} v_i P_i, e_j \right\rangle \right| \leq \sqrt{t} \|v_{K^c}\|_2 \max_{j \in \mathrm{supp}(v) \setminus K} \left| \left\langle \sum_{i \in K} v_i P_i, e_j \right\rangle \right|.$$

Hence, there is some $j \in \mathrm{supp}(v) \setminus K$ such that

$$\left| \left\langle \sum_{i \in K} v_i P_i, e_j \right\rangle \right| > \frac{1}{5\sqrt{t}} \left\| \sum_{i \in K} v_i P_i \right\|_2.$$

So the vector $x(v) := \sum_{i \in K} v_i P_i \in \mathrm{span}\{P_i : i \in K\}$ satisfies $|x(v)_j| > \|x(v)\|_2 / (5\sqrt{t})$. Moreover, $x(v)$ is nonzero since $|x(v)_j| > 0$. Thus, $j \in \mathcal{I}_P(K)$. Since we chose $j$ to be in $\mathrm{supp}(v) \setminus K$, it follows that

$$|\mathrm{supp}(v) \setminus (\mathcal{I}_P(K) \cup K)| \leq |\mathrm{supp}(v) \setminus K| - 1 \leq \tau - 1$$

where the last inequality is by assumption (b) in the lemma statement. $\qquad\square$

We can now complete the proof of Theorem B.1 by repeatedly invoking Lemma B.4 (this proof was given in Section 2.2 and is duplicated here for completeness).

**Proof of Theorem B.1.** Let $\Sigma = \sum_{i=1}^n \lambda_i u_i u_i^\top$ be the eigendecomposition of $\Sigma$, and let $P := \sum_{i=d+1}^n u_i u_i^\top$ be the projection onto the top $n - d$ eigenspaces of $\Sigma$. Set $K_t = \{i \in [n] : P_{ii} < 1 - 1/(9t^2)\}$. Because $\mathrm{tr}(P) = n - d$ and $P_{ii} \leq 1$ for all $i \in [n]$, it must be that $|K_t| \leq 9t^2 d$. Also, for any $v \in B_0(t) \cap \mathcal{W}_{P,K_t}$ we have trivially by $t$-sparsity that $|\mathrm{supp}(v) \setminus K_t| \leq t$.

Define $K_{t-1}$ to be $K_t \cup \mathcal{I}_P(K_t)$ where $\mathcal{I}_P(K_t)$ is as defined in Lemma B.4; we have the guarantees that $|K_{t-1}| \leq (1 + 36t^2)|K_t|$ and $|\mathcal{G}_P(v) \setminus K_t| \leq t - 1$ for all $v \in B_0(t) \cap \mathcal{W}_{P,K_t}$. Since $K_{t-1} \supseteq K_t$, it holds that $\mathcal{W}_{P,K_{t-1}} \subseteq \mathcal{W}_{P,K_t}$, and thus $|\mathcal{G}_P(v) \setminus K_t| \leq t - 1$ for all $v \in B_0(t) \cap \mathcal{W}_{P,K_{t-1}}$. Moreover, since $K_{t-1} \supseteq K_t$, it obviously holds that $P_{ii} \geq 1 - 1/(9t^2)$ for all $i \notin K_{t-1}$. This means we can apply Lemma B.4 with $\tau := t - 1$ and $K := K_{t-1}$ and so iteratively define sets $K_{t-2} \subseteq \cdots \subseteq K_1 \subseteq K_0 \subseteq [n]$ in the same way. In the end, we obtain the set $K_0 \subseteq [n]$ with $|K_0| \leq 9t^2 d(1 + 36t^2)^t$ and $\mathrm{supp}(v) \subseteq K_0$ for all $v \in B_0(t) \cap \mathcal{W}_{P,K_0}$. The latter guarantee means that in fact $B_0(t) \cap \mathcal{W}_{P,K_0} = \emptyset$. So for any $t$-sparse $v \in \mathbb{R}^n$ it holds that

$$\left\| v_{K_0^c} \right\|_2 \leq 3\sqrt{v^\top P v} \leq 3\lambda_{d+1}^{-1/2} \sqrt{v^\top \Sigma v}$$

where the last inequality holds since $\lambda_{d+1} P \preceq \Sigma$. $\qquad\blacksquare$

**Proof of Lemma 2.9.** By Theorem B.1, there is a polynomial-time computable set $S \subseteq [n]$ such that $\|v_{S^c}\|_2 \le 3\sqrt{t}\lambda_{d+1}^{-1/2}\|v\|_\Sigma$ for all $v \in B_0(t)$, and $|S| \le (7t)^{2t+1}d$. Let the dictionary $\mathcal{D}$ consist of the standard basis $\{e_1, \ldots, e_n\}$ together with a $\Sigma$-orthogonal basis for each subspace spanned by $t$ vectors in $\{e_i : i \in S\}$. Let $v \in \mathbb{R}^n$ be $t$-sparse. Let $v_S$ denote the restriction of $v$ to $S$, i.e. $v_S := v - \sum_{i\in[n]\setminus S} v_i e_i$. By construction of the dictionary, there is a $\Sigma$-orthogonal basis for $\{e_i : i \in S \cap \operatorname{supp}(v)\}$, so there are $d_1, \ldots, d_t \in \mathcal{D}$ and coefficients $b_{d_1}, \ldots, b_{d_t} \in \mathbb{R}$ with $v_S = \sum_{i=1}^t b_{d_i} d_i$ and $\langle d_i, d_j\rangle_\Sigma = 0$ for all $i, j \in [t]$ with $i \ne j$. Note that $\|v_S\|_\Sigma^2 = \sum_{i=1}^t b_{d_i}^2 \|d_i\|_\Sigma^2$, so

$$\sum_{i=1}^t |b_{d_i}|\, \|d_i\|_\Sigma \le \sqrt{t}\sqrt{\sum_{i=1}^t b_{d_i}^2 \|d_i\|_\Sigma^2} = \sqrt{t}\,\|v_S\|_\Sigma.$$

Now, we claim that the desired coefficient vector $\{\alpha_d : d \in \mathcal{D}\}$ for $v$ is defined by $\alpha_d = b_d + \sum_{i\in[n]\setminus S} v_i \mathbb{1}[d = e_i]$. We can check that $\sum_{d\in\mathcal{D}} \alpha_d d = \sum_{i=1}^t b_{d_i} + \sum_{i\in[n]\setminus S} v_i e_i = v$. Also,

$$\begin{aligned}
\|v_S\|_\Sigma &\le \|v\|_\Sigma + \|v_{S^c}\|_\Sigma \\
&\le \|v\|_\Sigma + \sqrt{\lambda_n}\,\|v_{S^c}\|_2 \\
&\le (1 + 3\sqrt{\lambda_n/\lambda_{d+1}})\,\|v\|_\Sigma
\end{aligned}$$

by the guarantee of set $S$.

It follows that

$$\sum_{i=1}^t |b_{d_i}|\, \|d_i\|_\Sigma \le (1 + 3\sqrt{\lambda_n/\lambda_{d+1}})\sqrt{t}\,\|v\|_\Sigma\,\sqrt{\lambda_n/\lambda_{d+1}}.$$

Thus,

$$\begin{aligned}
\sum_{d\in\mathcal{D}} |c_d|\, \|d\|_\Sigma &\le (1 + 3\sqrt{\lambda_n/\lambda_{d+1}})\sqrt{t}\,\|v\|_\Sigma + \sum_{i\in[n]\setminus S} |v_i|\, \|e_i\|_\Sigma \\
&\le (1 + 3\sqrt{\lambda_n/\lambda_{d+1}})\sqrt{t}\,\|v\|_\Sigma + \sqrt{t}\,\|v_{S^c}\|_2\,\sqrt{\lambda_n} \\
&\le (1 + 3\sqrt{\lambda_n/\lambda_{d+1}})\sqrt{t}\,\|v\|_\Sigma + 3\sqrt{t}\,\|v\|_\Sigma\,\sqrt{\lambda_n/\lambda_{d+1}} \\
&\le 7\sqrt{t}\sqrt{\lambda_n/\lambda_{d+1}}\,\|v\|_\Sigma
\end{aligned}$$

which completes the proof. ∎

**Proof of Theorem 2.3.** Immediate from Lemma 2.9 and Lemma A.3. ∎

## C  An efficient algorithm for handling outlier eigenvalues

In this section we describe and provide error guarantees for a novel sparse linear regression algorithm `BOAR-Lasso()` (see Algorithm 2 for pseudocode), completing the proof of Theorem 1.1; in Section C.1 we then analyze a modified algorithm to prove Theorem 1.2.

The key subroutine of `BOAR-Lasso()` is the procedure `AdaptivelyRegularizedLasso()`, which (like the simplified procedure `AdaptedBP()` from Section 3) first invokes procedure `IterativePeeling()` to compute the set of coordinates that participate in sparse approximate dependencies, and second computes a modified Lasso estimate where those coordinates are not regularized.

We start with Theorem C.2, which shows that, in the setting where $\Sigma$ has few outlier eigenvalues, the procedure `AdaptivelyRegularizedLasso()` estimates the sparse ground truth regressor at the "slow rate" (e.g. in the noiseless setting, the excess risk is at most $O(\|v^*\|_\Sigma^2 \, r_{\text{eff}}/m)$). Typical excess risk analyses for Lasso proceed by applying some general-purpose machinery for generalization bounds, such as the following result which only requires understanding $\langle w - w^*, X\rangle$ for $X \sim N(0, \Sigma)$.

---

**Algorithm 2:** Solve sparse linear regression when covariate eigenspectrum has few outliers

---

**Procedure** AdaptivelyRegularizedLasso($\Sigma$, $(X_i, y_i)_{i=1}^m$, $t$, $d_l$, $\delta$)

   **Data:** Covariance matrix $\Sigma : n \times n$, samples $(X_i, y_i)_{i=1}^m$, sparsity $t$, small eigenvalue count
      $d_l$, failure probability $\delta$

   **Result:** Estimate $\hat{v}$ of unknown sparse regressor, satisfying Theorem C.2

   $\sum_{i=1}^n \lambda_i u_i u_i^\top \leftarrow$ eigendecomposition of $\Sigma$

   $S \leftarrow$ IterativePeeling($\Sigma$, $d_l$, $t$)                                     /* See Algorithm 1 */

   Return

$$\hat{v} \leftarrow \underset{v \in \mathbb{R}^n}{\arg\min} \sum_{i=1}^m (\langle X_i, v \rangle - y)^2 + 8\lambda_{n-d_h} \log(12n/\delta) \|v_{S^c}\|_1^2 + 2\sqrt{2\lambda_{n-d_h} \log(12n/\delta)} \|v_{S^c}\|_1 .$$

**Procedure** BOAR-Lasso($\Sigma$, $(Y_i, y_i)_{i=1}^m$, $t$, $d_l$, $L$, $\delta$)

   **Data:** Covariance matrix $\Sigma : n \times n$, samples $(X_i, y_i)_{i=1}^m$, sparsity $t$, small eigenvalue count
      $d_l$, repetition count $L$, failure probability $\delta$

   **Result:** Estimate $\hat{v}$ of unknown sparse regressor, satisfying Theorem C.3

   $\hat{s}^{(0)} \leftarrow 0 \in \mathbb{R}^n$

   **for** $0 \le j < L$ **do**

      Set

$$\Sigma^{(j)} \leftarrow \begin{bmatrix} \Sigma & (\hat{s}^{(j)})^\top \Sigma \\ \Sigma \hat{s}^{(j)} & (\hat{s}^{(j)})^\top \Sigma \hat{s}^{(j)} \end{bmatrix} .$$

      Set $A^{(j)} := \{mj+1, \ldots, m(j+1)\}$

      $\hat{w}^{(j+1)} \leftarrow$ AdaptivelyRegularizedLasso($\Sigma^{(j)}$,

      $((X_i, \langle X_i, \hat{s}^{(j)} \rangle), y_i - \langle X_i, \hat{s}^{(j)} \rangle)_{i \in A^{(j)}}$, $t+1$, $d_l + 1$, $\delta/L$)

      $\hat{v}^{(j+1)} \leftarrow \hat{w}^{(j+1)}_{[n]} + \hat{w}^{(j+1)}_{n+1} \hat{s}^{(j)}$

      $\hat{s}^{(j+1)} \leftarrow \hat{s}^{(j)} + \hat{v}^{(j+1)}$

   **return** $\hat{s}^{(L)}$

---

**Theorem C.1** (Theorem 1 in [45]). *Let $n, m \in \mathbb{N}$ and $\epsilon, \delta, \sigma > 0$. Let $\Sigma : n \times n$ be a positive semi-definite matrix and fix $w^* \in \mathbb{R}^n$. Let $X : m \times n$ have i.i.d. rows $X_1, \ldots, X_m \sim N(0, \Sigma)$, and let $y = Xw^* + \xi$ where $\xi \sim N(0, \sigma^2 I_m)$. Let $F : \mathbb{R}^d \to [0, \infty]$ be a continuous function such that*

$$\Pr_{x \sim N(0, \Sigma)} [\sup_{w \in \mathbb{R}^n} \langle w - w^*, x \rangle - F(w) > 0] \le \delta.$$

*If $m \ge 196\epsilon^{-2} \log(12/\delta)$, then with probability at least $1 - 4\delta$ it holds that for all $w \in \mathbb{R}^d$,*

$$\|w - w^*\|_\Sigma^2 + \sigma^2 \le \frac{1+\epsilon}{m} (\|Xw - y\|_2 + F(w))^2 .$$

In classical settings, e.g. (a) where $\|v^*\|_1$ is bounded and $\max_i \Sigma_{ii} \le 1$ (see Proposition A.6) or (b) where $\Sigma$ satisfies the compatibility condition (see Definition G.1), the above result can be applied together with the straightforward bound $\langle v - v^*, X \rangle \le \|v - v^*\|_1 \|X\|_\infty$. To prove Theorem C.2 we follow the same general recipe as (a), with several modifications.

First, since $\max_i \Sigma_{ii}$ could be arbitrarily large, we need to treat the (few) large eigenspaces of $\Sigma$ separately when bounding $\langle v - v^*, X \rangle$. Similarly, since Theorem B.1 only gives bounds on $v^*$ for coordinates outside $S$, we separately bound $\langle (v - v^*)_S, X \rangle$ using that $|S|$ is small. Second, to achieve the optimal rate of $\sigma^2 n_{\text{eff}}/m$ rather than $\sigma^2 \sqrt{n_{\text{eff}}/m}$, we do not directly apply Theorem C.1 to the noisy samples $(X_i, y_i)$; instead, we derive a modification of that result (Lemma F.7) that only invokes Theorem C.1 on the noiseless samples $(X_i, \langle X_i, v^* \rangle)$, and separately bounds the in-sample prediction error $\|\mathbb{X}(\hat{v} - v^*)\|_2$. A similar technique is used in [45] for constrained least-squares programs (see their Lemma 15); our Lemma F.7 applies to a broad family of additively regularized programs, which obviates the need to independently estimate $\|v^*\|_\Sigma$ but otherwise achieves comparable bounds.

**Theorem C.2.** *Let $n, t, d_l, d_h, m \in \mathbb{N}$ and $\sigma, \delta > 0$. Let $\Sigma : n \times n$ be a positive semi-definite matrix with eigenvalues $\lambda_1 \le \cdots \le \lambda_n$. Let $(X_i, y_i)_{i=1}^m$ be independent samples where $X_i \sim N(0, \Sigma)$ and*

$y_i = \langle X_i, v^* \rangle + \xi_i$, for $\xi_i \sim N(0, \sigma^2)$ and a fixed $t$-sparse vector $v^* \in \mathbb{R}^n$. Let $\hat{v}$ be the output of `AdaptivelyRegularizedLasso`$(\Sigma, (X_i, y_i)_{i=1}^m, t, d_l, \delta)$. Let $n_{eff} := (7t)^{2t+1}d_l + d_h + \log(48/\delta)$ and let $r_{eff} := t(\lambda_{n-d_h}/\lambda_{d_l+1})\log(12n/\delta)$. There are absolute constants $c, C > 0$ so that the following holds. If $m \geq Cn_{eff}$, then with probability at least $1 - \delta$,

$$\|\hat{v} - v^*\|_\Sigma^2 \leq c\left(\frac{\sigma^2 n_{eff}}{m} + \frac{(\sigma + \|v^*\|_\Sigma)\|v^*\|_\Sigma \sqrt{r_{eff}}}{\sqrt{m}} + \frac{\|v^*\|_\Sigma^2 r_{eff}}{m}\right).$$

*Proof.* Define projection matrix $P := \sum_{i=1}^{n-d_h} u_i u_i^\top$, so that $\text{rank}(P^\perp) = d_h$ and $\lambda_{\max}(P\Sigma P) \leq \lambda_{n-d_h}$. For any $v \in \mathbb{R}^n$ and $X \sim N(0, \Sigma)$, we can bound

$$\langle v - v^*, X \rangle = \langle (v - v^*)_{S^c}, PX \rangle + \langle (v - v^*)_{S^c}, P^\perp X \rangle + \langle (v - v^*)_S, X \rangle$$
$$= \langle (v - v^*)_{S^c}, PX \rangle + \langle \Sigma^{1/2}(v - v^*), \Sigma^{-1/2}(P^\perp X)_{S^c} \rangle + \langle \Sigma^{1/2}(v - v^*), \Sigma^{-1/2}X_S \rangle$$
$$\leq \|(v - v^*)_{S^c}\|_1 \|PX\|_\infty + \left\|\Sigma^{1/2}(v - v^*)\right\|_2 (\|Z\|_2 + \|W\|_2)$$

where $PX \sim N(0, P\Sigma P)$, $Z \sim N(0, \Sigma^{-1/2}(P^\perp\Sigma P^\perp)_{S^c S^c}\Sigma^{-1/2})$, and $W \sim N(0, \Sigma^{-1/2}\Sigma_{SS}\Sigma^{-1/2})$. First, since $\max_i(P\Sigma P)_{ii} \leq \lambda_{\max}(P\Sigma P) \leq \lambda_{n-d_h}$, we have the Gaussian tail bound

$$\Pr\left[\|PX\|_\infty > \sqrt{\lambda_{n-d_h} \cdot 2\log(12n/\delta)}\right] \leq \delta/12.$$

Second, since

$$\Sigma^{-1/2}(P^\perp\Sigma P^\perp)_{S^c S^c}\Sigma^{-1/2} \preceq \Sigma^{-1/2}(P^\perp\Sigma P^\perp)\Sigma^{-1/2} \quad \text{(by Cauchy Interlacing Theorem)}$$
$$= P^\perp \quad \text{(since } P^\perp \text{ commutes with } \Sigma)$$

we have that $\|Z\|_2^2$ is stochastically dominated by $\chi_{d_h}^2$, and thus

$$\Pr\left[\|Z\|_2 > \sqrt{2d_h}\right] \leq e^{-m/4} \leq \delta/12.$$

Third, similarly, since $\Sigma^{-1/2}\Sigma_{SS}\Sigma^{-1/2} \preceq I$ (again by Cauchy Interlacing Theorem) and also $\text{rank}(\Sigma^{-1/2}\Sigma_{SS}\Sigma^{-1/2}) \leq |S|$, we have that $\|W\|_2^2$ is stochastically dominated by $\chi_{|S|}^2$, and thus

$$\Pr\left[\|W\|_2^2 > \sqrt{2|S|}\right] \leq e^{-m/4} \leq \delta/12.$$

Combining the above bounds, we have that with probability at least $1 - \delta/4$ over $X \sim N(0, \Sigma)$, for all $v \in \mathbb{R}^n$,

$$\langle v - v^*, X \rangle \leq \|(v - v^*)_{S^c}\|_1 \sqrt{\lambda_{n-d_h} \cdot 2\log(12n/\delta)} + \left\|\Sigma^{1/2}(v - v^*)\right\|_2 (\sqrt{2d_h} + \sqrt{2|S|}).$$

We can therefore apply Lemma F.7 with covariance $\Sigma$, seminorm $\Phi(v) := 2\sqrt{2\lambda_{n-d_h}\log(12n/\delta)}\|v_{S^c}\|_1$, $p := 4(d_h + |S|)$, ground truth $v^*$, samples $(X_i, y_i)_{i=1}^m$, and failure probability $\delta/4$. By the bound on $|S|$ (Theorem B.1) we have $|S| + d_h \leq (7t)^{2t+1}d_l + d_h \leq n_{eff}$, so it holds that $m \geq 16p + 196\log(48/\delta)$. Thus, with probability at least $1 - 2\delta$, we have

$$\|\hat{v} - v^*\|_\Sigma^2 \leq O\left(\frac{\sigma^2 n_{eff}}{m} + \frac{(\sigma + \|v^*\|_\Sigma)\|v_{S^c}^*\|_1 \sqrt{\lambda_{n-d_h}\log(12n/\delta)}}{\sqrt{m}} + \frac{\|v_{S^c}^*\|_1^2 \lambda_{n-d_h}\log(12n/\delta)}{m}\right).$$

By the guarantee of $S$ (Theorem B.1) and $t$-sparsity of $v^*$, we have $\|v_{S^c}^*\|_2 \leq 3\lambda_{d_l+1}^{-1/2}\|v^*\|_\Sigma$, and thus $\|v_{S^c}^*\|_1 \leq 3\sqrt{t}\lambda_{d_l+1}^{-1/2}\|v^*\|_\Sigma$. Substituting into the previous bound, we get

$$\|\hat{v} - v^*\|_\Sigma^2 \leq O\left(\frac{\sigma^2 n_{eff}}{m} + \frac{(\sigma + \|v^*\|_\Sigma)\|v^*\|_\Sigma \sqrt{r_{eff}}}{\sqrt{m}} + \frac{\|v^*\|_\Sigma^2 r_{eff}}{m}\right)$$

as claimed. $\square$

The limitation of `AdaptivelyRegularizedLasso()` is that the excess risk bound depends on $\|v^*\|_\Sigma^2$ rather than just $\sigma^2$. We next show that by a boosting approach, we can exponentially attenuate that dependence, essentially achieving the near-optimal rate of $\sigma^2 n_{\text{eff}}/m$. The key insight is that after producing an estimate $\hat{v}$ of $v^*$, we can augment the set of covariates with the feature $\langle \mathbb{X}, \hat{v} \rangle$, and try to predict the response $y - \langle \mathbb{X}, \hat{v} \rangle$, which is now a $(t+1)$-sparse combination of the features. In standard settings, this is typically a bad idea because it introduces a sparse linear dependence. However, by the Cauchy Interlacing Theorem it increases the number of outlier eigenvalues by at most one – so our algorithms still apply. Thus, if we have enough samples that the excess risk bound in Theorem C.2 is non-trivially smaller than $\|v^*\|_\Sigma^2$, then we can iteratively achieve better and better estimates up to the noise limit. This is precisely what `BOAR-Lasso()` does; the precise guarantees are stated in the following theorem, which completes the proof of Theorem 1.1.

**Theorem C.3.** *Let $n, t, d_l, d_h, m, L \in \mathbb{N}$ and $\sigma, \delta > 0$. Let $\Sigma : n \times n$ be a positive semi-definite matrix with eigenvalues $\lambda_1 \leq \cdots \leq \lambda_n$. Let $(X_i, y_i)_{i=1}^m$ be independent samples where $X_i \sim N(0, \Sigma)$ and $y_i = \langle X_i, v^* \rangle + \xi_i$, for $\xi_i \sim N(0, \sigma^2)$ and a fixed $t$-sparse vector $v^* \in \mathbb{R}^n$.*

*Then, given $\Sigma$, $(X_i, y_i)_{i=1}^m$, $t$, $d_l$, and $\delta$, the algorithm `BOAR-Lasso()` outputs an estimator $\hat{v}$ with the following properties.*

*Let $n_{\text{eff}} := (7t)^{2t+1} d_l + d_h + \log(48/\delta)$ and let $r_{\text{eff}} := t(\lambda_{n-d_h}/\lambda_{d_l+1}) \log(12n/\delta)$. There are absolute constants $c_0, C_0 > 0$ such that the following holds. If $m \geq C_0 L(n_{\text{eff}} + r_{\text{eff}})$, then with probability at least $1 - \delta$, it holds that*

$$\|\hat{v} - v^*\|_\Sigma^2 \leq c_0 \frac{\sigma^2(n_{\text{eff}} + r_{\text{eff}})}{m/L} + 2^{-L} \cdot \|v^*\|_\Sigma^2.$$

*Moreover, `BOAR-Lasso()` has time complexity $\text{poly}(n, m, t)$.*

*Proof.* Let $(A_0, \ldots, A_{L-1})$ be an partition of $[m]$ into $L$ sets of size $m/L$. The idea of the algorithm is to compute vectors $\hat{v}^{(1)}, \ldots, \hat{v}^{(L)}$ where each $v^{(i)}$ is an estimate of $v^* - \sum_{j=1}^{i-1} \hat{v}^{(j)}$. Concretely, fix some $0 \leq j \leq L - 1$ and suppose that we have computed some vectors $\hat{v}^{(1)}, \ldots, \hat{v}^{(j)}$. Set $\hat{s}^{(j)} := \hat{v}^{(1)} + \cdots + \hat{v}^{(j)}$. Define a matrix $\Sigma^{(j)} : (n+1) \times (n+1)$ by

$$\Sigma^{(j)} := \begin{bmatrix} \Sigma & (\hat{s}^{(j)})^\top \Sigma \\ \Sigma \hat{s}^{(j)} & (\hat{s}^{(j)})^\top \Sigma (\hat{s}^{(j)}) \end{bmatrix}.$$

Thus, for example, $\Sigma^{(0)}$ has zeroes in the last row and last column. Now for each $i \in A_j$, define $(X_i^{(j)}, y_i^{(j)})$ by

$$X_i^{(j)} := (X_i, \langle X_i, \hat{s}^{(j)} \rangle)$$

$$y_i^{(j)} := y_i - \langle X_i, \hat{s}^{(j)} \rangle.$$

By construction, the $m/L$ samples $(X_i^{(j)}, y_i^{(j)})_{i \in A_j}$ are independent and distributed as $X_i^{(j)} \sim N(0, \Sigma^{(j)})$ and $y_i^{(j)} = \langle X_i^{(j)}, (v^*, -1) \rangle + \xi_i$. Let $\lambda_1^{(j)} \leq \cdots \leq \lambda_{n+1}^{(j)}$ be the eigenvalues of $\Sigma^{(j)}$.

Now we apply Theorem C.2 with covariance $\Sigma^{(j)}$, samples $(X_i^{(j)}, y_i^{(j)})_{i \in A_j}$, sparsity $t+1$, outlier counts $d_l + 1$ and $d_h + 1$, and failure probability $\delta/L$; let $n_{\text{eff}}^{(j)}$ and $r_{\text{eff}}^{(j)}$ be the induced parameters defined in that theorem statement, and let $c, C$ be the constants. By the Cauchy Interlacing Theorem, we have $\lambda_{d_l+2}^{(j)} \geq \lambda_{d_l+1}$ and similarly $\lambda_{n+1-(d_h+1)}^{(j)} \leq \lambda_{n-d_h}$. Thus $r_{\text{eff}}^{(j)} \leq 2r_{\text{eff}}$. Also $n_{\text{eff}}^{(j)} \leq n_{\text{eff}}$. Thus, if the constant $C_0$ is chosen appropriately large, then $m/L \geq 16cr_{\text{eff}}^{(j)}$ and also $m/L \geq Cn_{\text{eff}}^{(j)}$. Hence (by the error guarantee of Theorem C.2) with probability at least $1 - \delta/L$ we obtain a vector $\hat{w}^{(j+1)}$ such that

$$\left\| \hat{w}^{(j+1)} - (v^*, -1) \right\|_{\Sigma^{(j)}}^2 \leq \frac{c\sigma^2 n_{\text{eff}}^{(j)}}{m/L} + c\|(v^*, -1)\|_{\Sigma^{(j)}}^2 \sqrt{\frac{r_{\text{eff}}^{(j)}}{m/L}} + c\sigma \|(v^*, -1)\|_{\Sigma^{(j)}} \sqrt{\frac{r_{\text{eff}}^{(j)}}{m/L}}$$

$$\leq \frac{2c\sigma^2 n_{\text{eff}}}{m/L} + \frac{\|(v^*, -1)\|_{\Sigma^{(j)}}^2}{4} + \left( \frac{\|(v^*, -1)\|_{\Sigma^{(j)}}^2}{4} + \frac{4c^2\sigma^2 r_{\text{eff}}}{m/L} \right)$$

$$\leq \frac{c_0}{2} \frac{\sigma^2(n_{\text{eff}} + r_{\text{eff}})}{m/L} + \frac{\|(v^*, -1)\|_{\Sigma^{(j)}}^2}{2} \tag{5}$$

where the second inequality uses AM-GM to bound the third term, and the third inequality is by choosing $c_0 \geq 4c + 8c^2$.

But now define $\hat{v}^{(j+1)} := \hat{w}_{[n]}^{(j+1)} + \hat{w}_{n+1}^{(j+1)} \hat{s}^{(j)}$. Then we observe that $\|(v^*, -1)\|_{\Sigma^{(j)}}^2 = \|v^* - \hat{s}^{(j)}\|_\Sigma^2$ and $\|\hat{w}^{(j+1)} - (v^*, -1)\|_{\Sigma^{(j)}}^2 = \|\hat{v}^{(j+1)} - (v^* - \hat{s}^{(j)})\|_\Sigma^2 = \|v^* - \hat{s}^{(j+1)}\|_\Sigma^2$ where $\hat{s}^{(j+1)} = \hat{v}^{(1)} + \cdots + \hat{v}^{(j+1)}$. So (5) is equivalent to

$$\left\|v^* - \hat{s}^{(j+1)}\right\|_\Sigma^2 \leq \frac{c_0}{2} \frac{\sigma^2(n_{\text{eff}} + r_{\text{eff}})}{m/L} + \frac{1}{2} \left\|v^* - \hat{s}^{(j)}\right\|_\Sigma^2.$$

Inductively, we conclude that

$$\left\|v^* - \hat{s}^{(L)}\right\|_\Sigma^2 \leq c_0 \frac{\sigma^2(n_{\text{eff}} + r_{\text{eff}})}{m/L} + 2^{-L} \|v^*\|_\Sigma^2$$

as desired. The time complexity (see Algorithm 2 for full pseudocode) is dominated by $L$ eigen-decompositions of $n \times n$ Hermitian matrices (each of which takes time $O(n^3)$ by e.g. the QR algorithm), as well as $L$ convex optimizations (each of which takes time $\tilde{O}(n^3)$ to solve to inverse-polynomial accuracy [26], which is sufficient for the correctness proof). $\qquad\square$

## C.1 An alternative algorithm (proof of Theorem 1.2)

In this section we prove Theorem 1.2, which essentially states that the sample complexity dependence on $d_l$ in `BOAR-Lasso()` can be removed at the cost of a time complexity depending on $d_l^t$. See Algorithm 3 for the pseudocode of how we modify `AdaptivelyRegularizedLasso()`: essentially, we brute force search over all size-$t$ subsets of the set $S$ produced by `IterativePeeling()`, construct an appropriate dictionary for each of these $\binom{|S|}{t}$ subsets, and then perform a final model selection step (with fresh samples) to pick the best dictionary/estimator. The boosting step is exactly identical to that in `BOAR-Lasso()`.

**Lemma C.4.** *Let $n, t, d \in \mathbb{N}$. Let $\Sigma : n \times n$ be a positive semi-definite matrix with eigenvalues $\lambda_1 \leq \cdots \leq \lambda_n$. Then there is a family $\mathcal{D} \subseteq \mathbb{R}^{n \times (n+t)}$ of size $|\mathcal{D}| \leq (7t)^{2t^2 + t}(2d)^t$, consisting entirely of $n \times (n + t)$ matrices with the form*

$$D := \begin{bmatrix} I_n & d_1 & \ldots & d_t \end{bmatrix},$$

*with the following property. For any $t$-sparse $v \in \mathbb{R}^n$, there is some $D \in \mathcal{D}$ and $w \in \mathbb{R}^{n+k}$ with $v = Dw$ and*

$$\|w\|_1 \leq \frac{7t^{1/2}}{\sqrt{\lambda_{d+1}}} \sqrt{v^\top \Sigma v}.$$

*Proof.* Let $u_1, \ldots, u_n \in \mathbb{R}^n$ be the eigenvectors of $\Sigma$ corresponding to eigenvalues $\lambda_1, \ldots, \lambda_n$ respectively, so that $\Sigma = \sum_{i=1}^n \lambda_i u_i u_i^\top$. Define $\overline{\Sigma} := \lambda_{d+1}^{-1} \sum_{i=1}^n \min(\lambda_i, \lambda_{d+1}) u_i u_i^\top$. Let $S$ be the output of `IterativePeeling`$(\Sigma, d_l, t)$, and let $\mathcal{D} := \{D(T) : T \in \binom{S}{t}\}$, where for any $T \in \binom{S}{t}$, we let $\{d_1, \ldots, d_t\}$ be a $\overline{\Sigma}$-orthonormal basis for $\text{span}\{e_i : i \in T\}$, and let $D(T)$ be the $n \times (n + t)$ matrix with columns $e_1, \ldots, e_n, d_1, \ldots, d_t$. The bound on $|\mathcal{D}|$ follows from Theorem B.1.

For any $t$-sparse $v \in \mathbb{R}^n$, pick the matrix $D \in \mathcal{D}$ indexed by any $T \in \binom{S}{t}$ with $S \cap \text{supp}(v) \subseteq T$. Let $d_1, \ldots, d_t \in \mathbb{R}^n$ be the last $t$ columns of $D$. Then there are coefficients $b_1, \ldots, b_t$ so that we can write $v_S = \sum_{i=1}^t b_i d_i$. Since $d_i^\top \overline{\Sigma} d_{i'} = \mathbb{1}[i = i']$ for all $i, i' \in [t]$, we have $v_S^\top \overline{\Sigma} v_S = \sum_{i=1}^t b_i^2$. Hence, $\|b\|_1 \leq \sqrt{t} \sqrt{v_S^\top \overline{\Sigma} v_S}$. But we can bound

$$\begin{aligned} \sqrt{v_S^\top \overline{\Sigma} v_S} &= \left\|\overline{\Sigma}^{1/2} v_S\right\|_2 \\ &\leq \left\|\overline{\Sigma}^{1/2} v\right\|_2 + \left\|\overline{\Sigma}^{1/2} v_{S^c}\right\|_2 \qquad\qquad \text{(by triangle inequality)} \end{aligned}$$

**Algorithm 3:** Alternative algorithm to solve sparse linear regression when covariate eigenspectrum has few outliers

---

**Procedure** AugmentedDictionaryLasso($\Sigma$, $(X_i, y_i)_{i=1}^m$, $t$, $d_l$, $\delta$)

    **Data:** Covariance matrix $\Sigma : n \times n$, samples $(X_i, y_i)_{i=1}^m$, sparsity $t$, small eigenvalue count $d_l$, failure probability $\delta$

    **Result:** Estimate $\hat{v}$ of unknown sparse regressor, satisfying Theorem C.5

    $\sum_{i=1}^n \lambda_i u_i u_i^\top \leftarrow$ eigendecomposition of $\Sigma$

    $S \leftarrow$ IterativePeeling($\Sigma, d_l, t$)                                  /* See Algorithm 1 */

    $\overline{\Sigma} \leftarrow \lambda_{d_l+1}^{-1} \sum_{i=1}^n \min(\lambda_i, \lambda_{d_l+1}) u_i u_i^\top$

    **for** $T \in \binom{S}{[t]}$ **do**

        $d_1^{(T)}, \ldots, d_t^{(T)} \leftarrow \overline{\Sigma}$-orthogonal basis for $\text{span}\{e_i : i \in T\}$

        $D(T) \leftarrow \begin{bmatrix} I_n & d_1^{(T)} & \ldots & d_t^{(T)} \end{bmatrix}$

        Compute

$$\hat{w}(T) \leftarrow \underset{w \in \mathbb{R}^{n+t}}{\operatorname{argmin}} \left[ \sum_{i=1}^{m/2} \left( \langle X_i, D(T)w \rangle - y_{1:m/2} \right)^2 \right.$$

$$\left. + 8\lambda_{n-d} \log(8n/\delta) \|w\|_1^2 + 2\sqrt{2\lambda_{n-d} \log(8n/\delta)} \|y_{1:m/2}\|_2 \|w\|_1 \right]$$

    Select best hypothesis

$$\hat{T} \leftarrow \underset{T \in \binom{S}{[t]}}{\operatorname{argmin}} \sum_{i=m/2+1}^{m} \left( \langle X_i, D(T)\hat{w}(T) \rangle - y_i \right)^2$$

    **return** $D(\hat{T})\hat{w}(\hat{T})$

---

$$\leq \lambda_{d+1}^{-1/2} \left\| \Sigma^{1/2} v \right\|_2 + \|v_{S^c}\|_2 \qquad\qquad \text{(by } \overline{\Sigma} \preceq \lambda_{d+1}^{-1}\Sigma \text{ and } \overline{\Sigma} \preceq I_n\text{)}$$

$$\leq \lambda_{d+1}^{-1/2} \left\| \Sigma^{1/2} v \right\|_2 + 3\lambda_{d+1}^{-1/2}\sqrt{v^\top \Sigma v} \qquad \text{(by Theorem B.1 and } t\text{-sparsity of } v\text{)}$$

$$\leq 4\lambda_{d+1}^{-1/2}\sqrt{v^\top \Sigma v}.$$

We conclude that $\|b\|_1 \leq 4\sqrt{t}\lambda_{d+1}^{-1/2}\sqrt{v^\top \Sigma v}$. Thus, if we define

$$w := \sum_{i \in [n] \setminus S} v_i e_i + \sum_{i=1}^t b_i e_{n+i},$$

where here $e_1, \ldots, e_{n+k}$ refer to the standard basis vectors in $\mathbb{R}^{n+t}$, then we have $Dw = v_{[n] \setminus S} + \sum_{i=1}^t b_i d_i = v$, and also

$$\|w\|_1 \leq \|b\|_1 + \sum_{i \in [n] \setminus S} |v_i| \leq 4\sqrt{t}\lambda_{d+1}^{-1/2}\sqrt{v^\top \Sigma v} + \sqrt{t}\|v_{S^c}\|_2 \leq \frac{7\sqrt{t}}{\lambda_{d+1}^{1/2}}\sqrt{v^\top \Sigma v}$$

as desired. $\qquad\qquad\qquad\qquad\qquad\qquad\qquad\qquad\qquad\qquad\qquad\qquad\qquad\qquad\qquad\qquad\quad \square$

**Theorem C.5.** *Let $n, t, d_l, d_h, m \in \mathbb{N}$ and let $\Sigma : n \times n$ be a positive semi-definite matrix with eigenvalues $\lambda_1 \leq \cdots \leq \lambda_n$. Let $(X_i, y_i)_{i=1}^m$ be independent samples where $X_i \sim N(0, \Sigma)$ and $y_i = \langle X_i, v^* \rangle + \xi_i$, for $\xi_i \sim N(0, \sigma^2)$ and a fixed $t$-sparse vector $v^* \in \mathbb{R}^n$. Set $k := t(7t)^{2t^2+t}d_l^t$ and let $\mathcal{D}$ be the family of matrices (of size at most $k$) guaranteed by Lemma C.4.*

*Let $\delta > 0$. For every $D \in \mathcal{D}$, define*

$$\hat{w}(D) \in \underset{w \in \mathbb{R}^{n+t}}{\operatorname{argmin}} \left\| \mathbb{X}^{(1)} Dw - y_{1:m/2} \right\|_2^2 + 8\lambda_{n-d}\log(8n/\delta)\|w\|_1^2 + 2\sqrt{2\lambda_{n-d}\log(8n/\delta)}\|y_{1:m/2}\|_2 \|w\|_1$$

$$\tag{6}$$

*where $\mathbb{X}^{(1)} : (m/2) \times n$ is the matrix with rows $X_1, \ldots, X_{m/2}$, and define $\hat{v} = \hat{D}\hat{w}(\hat{D})$ where*

$$\hat{D} \in \underset{D \in \mathcal{D}}{\operatorname{argmin}} \left\| \mathbb{X}^{(2)} D\hat{w}(D) - y_{m/2+1:m} \right\|_2^2$$

*where $\mathbb{X}^{(2)} : (m/2) \times n$ is the matrix with rows $X_{m/2+1}, \ldots, X_m$.*

*Let $n_{\mathit{eff}} := t^2 \log(t) + t \log(d_l) + d_h + \log(48/\delta)$ and let $r_{\mathit{eff}} := t(\lambda_{n-d_h}/\lambda_{d_l+1}) \log(8n/\delta)$. There are absolute constants $c, C > 0$ so that the following holds. If $m \geq C n_{\mathit{eff}}$, then with probability at least $1 - 3\delta$ it holds that*

$$\|\hat{v} - v^*\|_{\Sigma}^2 \leq c \left( \frac{\sigma^2 n_{\mathit{eff}}}{m} + \|v^*\|_{\Sigma}^2 \left( \frac{r_{\mathit{eff}}}{m} + \sqrt{\frac{r_{\mathit{eff}}}{m}} \right) + \sigma \|v^*\|_{\Sigma} \sqrt{\frac{r_{\mathit{eff}}}{m}} \right).$$

Let $D^* \in \mathcal{D}$ and $w^* \in \mathbb{R}^{n+t}$ be the matrix and vector guaranteed by Lemma C.4 for the $t$-sparse vector $v^*$. Let $\Gamma = (D^*)^\top \Sigma D^*$ with eigenvalues $\gamma_1 \leq \cdots \leq \gamma_{n+t}$. We make the following claim:

**Claim C.6.** *With probability at least $1 - \delta/4$ over $G \sim N(0, \Gamma)$, it holds uniformly in $w \in \mathbb{R}^{n+t}$ that*

$$\langle w - w^*, G \rangle \leq \|w - w^*\|_1 \sqrt{\lambda_{n-d_h} \cdot 2 \log(8n/\delta)} + \|w - w^*\|_{\Gamma} \sqrt{2(d_h + t)}.$$

*Proof.* Since $\Sigma$ is a principal submatrix of $\Gamma$, we have $\gamma_{n-d_h} \leq \lambda_{n-d_h}$ (by the Cauchy Interlacing Theorem). Suppose that $\Gamma$ has eigendecomposition $\Gamma = \sum_{i=1}^{n+t} \gamma_i g_i g_i^\top$, and define projection matrix $P : (n+t) \times (n+t)$ by $P := \sum_{i=1}^{n-d_h} g_i g_i^\top$, so that $\operatorname{rank}(P^\perp) = d_h + t$ and $\lambda_{\max}(P\Gamma P) \leq \gamma_{n-d_h} \leq \lambda_{n-d_h}$. Then for any $w \in \mathbb{R}^{n+t}$ and $G \sim N(0, \Gamma)$, we can bound

$$\langle w - w^*, G \rangle = \langle w - w^*, PG \rangle + \langle w - w^*, P^\perp G \rangle$$
$$\leq \|w - w^*\|_1 \|PG\|_\infty + \langle \Gamma^{1/2}(w - w^*), \Gamma^{-1/2} P^\perp G \rangle$$
$$= \|w - w^*\|_1 \|PG\|_\infty + \langle \Gamma^{1/2}(w - w^*), P^\perp \Gamma^{-1/2} G \rangle$$
$$\leq \|w - w^*\|_1 \|PG\|_\infty + \left\| \Gamma^{1/2}(w - w^*) \right\|_2 \|Z\|_2$$

where $Z \sim N(0, P^\perp)$. The second equality above uses that $\Gamma^{-1/2}$ and $P^\perp$ are simultaneously diagonalizable (and therefore commute). But now for any $\delta > 0$, we have the Gaussian tail bounds

$$\Pr\left[ \|PG\|_\infty > \sqrt{\max_i (P\Gamma P)_{ii} \cdot 2 \log(8n/\delta)} \right] \leq \delta/8$$

and

$$\Pr\left[ \|Z\|_2 > \sqrt{2 \operatorname{rank}(P^\perp)} \right] \leq e^{-m/8} \leq \delta/8.$$

Thus, with probability at least $1 - \delta/4$ over $G \sim N(0, \Gamma)$, for any $w \in \mathbb{R}^{n+t}$, we have

$$\langle w - w^*, G \rangle \leq \|w - w^*\|_1 \sqrt{\max_i (P\Gamma P)_{ii} \cdot 2 \log(8n/\delta)} + \left\| \Gamma^{1/2}(w - w^*) \right\|_2 \sqrt{2 \operatorname{rank}(P^\perp)}$$
$$\leq \|w - w^*\|_1 \sqrt{\lambda_{n-d_h} \cdot 2 \log(8n/\delta)} + \left\| \Gamma^{1/2}(w - w^*) \right\|_2 \sqrt{2(d_h + t)} \qquad = F(w)$$

which proves the claim. $\qquad \square$

We now proceed with proving the theorem.

**Proof of Theorem C.5.** Applying Claim C.6, we can now invoke Lemma F.7 with covariance matrix $\Gamma$, seminorm $\Phi(v) := 2\sqrt{2\lambda_{n-d_h} \cdot \log(8n/\delta)} \|v\|_1$, $p := 2(d_h + t)$, ground truth $w^*$, samples $((D^*)^\top X_i, y_i)_{i=1}^{m/2}$, and failure probability $\delta/4$. Since we chose $m$ sufficiently large that $m/2 \geq 16p + 196 \log(12/\delta)$, we conclude that with probability at least $1 - 2\delta$ over the randomness of $(X_i, y_i)_{i=1}^{m/2}$, it holds that

$$\|\hat{w}(D^*) - w^*\|_{\Gamma}^2 \leq O\left( \frac{\sigma^2(d_h + t)}{m} + \frac{(\sigma + \|w^*\|_{\Gamma}) \|w^*\|_1 \sqrt{\lambda_{n-d_h} \cdot \log(8n/\delta)}}{\sqrt{m}} + \frac{\|w^*\|_1^2 \lambda_{n-d_h} \log(8n/\delta)}{m} \right).$$

Since $v^* = D^* w^*$ and $\|w^*\|_1 \leq 7t^{1/2}\lambda_{d_l+1}^{-1/2}\|v^*\|_\Sigma$ (the guarantees of Lemma C.4), it follows that

$$\|D^*\hat{w}(D^*) - v^*\|_\Sigma^2 \leq O\left(\frac{\sigma^2(d_h + t)}{m} + \frac{(\sigma + \|v^*\|_\Sigma)\|v^*\|_\Sigma\sqrt{r_{\text{eff}}}}{\sqrt{m}} + \frac{\|v^*\|_\Sigma^2 r_{\text{eff}}}{m}\right).$$

To complete the proof of the theorem, condition on any values of $(X_i, y_i)_{i=1}^{m/2}$ for which the above bound holds. By applying Lemma F.2 with covariance matrix $\Sigma$, hypothesis set $\mathcal{W} := \{D\hat{w}(D) : D \in \mathcal{D}\}$, and samples $(X_i, y_i)_{i=m/2+1}^{m}$ (which are independent of $\mathcal{W}$), since $m/2 \geq 32\log(2|\mathcal{D}|/\delta)$, we have with probability at least $1 - 2\delta$ over the samples $(X_i, y_i)_{i=m/2+1}^{m}$ that

$$\left\|\hat{D}\hat{w}(\hat{D}) - v^*\right\|_\Sigma^2 \leq 6\min_{D \in \mathcal{D}}\|D\hat{w}(D) - v^*\|_\Sigma^2 + \frac{32\sigma^2\log(2|\mathcal{D}|/\delta)}{m}.$$

Hence, with probability at least $1 - 5\delta$ we have

$$\left\|\hat{D}\hat{w}(\hat{D}) - v^*\right\|_\Sigma^2 \leq O\left(\frac{\sigma^2(d_h + t + \log(2|\mathcal{D}|/\delta))}{m} + \frac{(\sigma + \|v^*\|_\Sigma)\|v^*\|_\Sigma\sqrt{r_{\text{eff}}}}{\sqrt{m}} + \frac{\|v^*\|_\Sigma^2 r_{\text{eff}}}{m}\right)$$

which proves the theorem. ∎

We can use the above theorem (together with the previously discussed boosting approach) to get the following result, which proves Theorem 1.2.

**Theorem C.7.** *Let $n, t, d_l, d_h, m, L \in \mathbb{N}$ and $\sigma, \delta > 0$. Let $\Sigma : n \times n$ be a positive semi-definite matrix with eigenvalues $\lambda_1 \leq \cdots \leq \lambda_n$. Let $(X_i, y_i)_{i=1}^{m}$ be independent samples where $X_i \sim N(0, \Sigma)$ and $y_i = \langle X_i, v^*\rangle + \xi_i$, for $\xi_i \sim N(0, \sigma^2)$ and a fixed $t$-sparse vector $v^* \in \mathbb{R}^n$.*

*Then, given $\Sigma$, $(X_i, y_i)_{i=1}^{m}$, $t$, $d_l$, and $\delta$, there is an estimator $\hat{v}$ with the following properties.*

*Let $n'_{\text{eff}} := t^2\log(t) + t\log(d_l) + d_h + \log(48L/\delta)$ and let $r'_{\text{eff}} := t(\lambda_{n-d_h}/\lambda_{d_l+1})\log(8nL/\delta)$. There are absolute constants $c_0, C_0 > 0$ such that the following holds. If $m \geq C_0 L(n'_{\text{eff}} + r'_{\text{eff}})$, then with probability at least $1 - \delta$, it holds that*

$$\|\hat{v} - v^*\|_\Sigma^2 \leq c_0\frac{\sigma^2(n'_{\text{eff}} + r'_{\text{eff}})}{m/L} + 2^{-L}\cdot\|v^*\|_\Sigma^2.$$

*Moreover, $\hat{v}$ is computable in time $(t+1)^{O(t^2)}(d_l+1)^{t+1}\cdot\text{poly}(n)$.*

*Proof.* Identical to that of Theorem C.3, except using Theorem C.5 instead of Theorem C.2. □

## D  Faster sparse linear regression for arbitrary $\Sigma$

In this section we prove Theorem 3.1. The approach is via feature adaptation: in Theorem D.5, we show that any covariance matrix $\Sigma$ has a $(t, O(t^{3/2}\log n)$-$\ell_1$-representation of size $O(n^{t-1/2})$ that is computable in time $n^{t-\Omega(1/t)}\log^{O(t)} n$, using $O(t\log n)$ samples from $N(0, \Sigma)$. The algorithm for computing this representation is described in Algorithm 4. One of the key tools is the following result from computational geometry:

**Theorem D.1** ([30]). *Let $n, d, k \in \mathbb{N}$ and $\delta > 0$. Given points $p_1, \ldots, p_n \in \mathbb{R}^d$, query dimension $k$, and failure probability $\delta$, there an algorithm $\text{DS}((p_1, \ldots, p_n), k, \delta)$ with time complexity $n^{k+1}(\log n)^{O(k)}\text{poly}(d)\log(1/\delta)$, that constructs a data structure $\mathcal{N}$ that answers queries of the following form. Given a $k$-dimensional subspace $F \subseteq \mathbb{R}^d$, the output $\mathcal{N}(F)$ is some $i^* \in [n]$. With probability at least $1 - \delta$, the query time complexity is $n^{1-1/(2k)}\text{poly}(d)\log(1/\delta)$, and it holds that*

$$\min_{q \in F}\|p_{i^*} - q\|_2 \leq O(\log n)\cdot\min_{i \in [n]}\min_{q \in F}\|p_i - q\|_2.$$

How do we use the above theorem to efficiently construct the $\ell_1$-representation? The intuition is as follows. Let $\mathbb{X}$ be the $m \times n$ matrix where each row is a sample from $N(0, \Sigma)$. Then each column is a vector $p_i$ representing a particular covariate. To find the $\ell_1$-representation, it essentially suffices

---

**Algorithm 4:** $\ell_1$-representation for arbitrary $\Sigma$

---

**1** **Procedure** FindOrthonormalization($p_1, \ldots, p_t$)

    **Data:** Nonzero vectors $p_1, \ldots, p_t \in \mathbb{R}^m$

    **Result:** $\alpha^{(1)}, \ldots, \alpha^{(t)} \in \mathbb{R}^t$ such that $\text{span}\{\alpha^{(1)}, \ldots, \alpha^{(t)}\} = \text{span}\{e_1, \ldots, e_t\}$ and

        $\langle \sum_\ell \alpha^{(i)}_\ell p_\ell, \sum_\ell \alpha^{(j)}_\ell p_\ell \rangle = 0$ for all $i \neq j$

**2**

**3**     **for** $i = 1, \ldots, t$ **do**

**4**         $\alpha^{(i)} \leftarrow e_i / \|p_i\|_2 \in \mathbb{R}^k$

**5**         **for** $j = 1, \ldots, i-1$ **do**

**6**             **if** $\sum_\ell \alpha^{(j)}_\ell p_\ell \neq 0$ **then**

**7**                 $\alpha^{(i)} \leftarrow \alpha^{(i)} - \frac{\langle \sum_\ell \alpha^{(i)}_\ell p_\ell, \sum_\ell \alpha^{(j)}_\ell p_\ell \rangle}{\left\| \sum_\ell \alpha^{(j)}_\ell p_\ell \right\|_2} \alpha^{(j)}$

**8**     **return** $\alpha^{(1)}, \ldots, \alpha^{(k)}$

**9** **Procedure** RepresentVectors($\{p_1, \ldots, p_n\}, t, \delta$)

    **Data:** Unit vectors $p_1, \ldots, p_n \in \mathbb{R}^m$, sparsity parameter $t$, failure probability $\delta$

    **Result:** Set $\mathcal{D} \subseteq \mathbb{R}^n$ of size $O(n^{t-1/2})$, where all elements $d \in \mathcal{D}$ are $t$-sparse (and

        represented succinctly)

**10**

**11**     Compute partition $I_1 \sqcup \cdots \sqcup I_{\sqrt{n}} = [n]$ where $|I_i| \leq \lceil \sqrt{n} \rceil$ for all $i$

**12**     Initialize $\mathcal{D} \leftarrow \emptyset$

**13**     **for** $j = 1, \ldots, \sqrt{n}$ **do**

**14**         Construct data structure $\mathcal{N}^j \leftarrow \text{DS}((p_i : i \in I_j), t-1, \delta/n^t)$    /* Theorem D.1 */

**15**         **for** $T \subseteq \binom{[n]}{t-1}$ **do**

**16**             $h(T, j) \leftarrow \mathcal{N}^j(\text{span}\{p_i : i \in T\})$                      /* Theorem D.1 */

**17**             Find $\gamma \in \mathbb{R}^T$ such that $\sum_{i \in T} \gamma_i p_i = \text{Proj}_{\text{span}\{p_i : i \in T\}} p_{h(T,j)}$

**18**             Write $\gamma$ as a sparse vector in $\mathbb{R}^n$ (supported on $T$)

**19**             Add $\gamma - e_{h(T,j)}$ to $\mathcal{D}$

**20**         **for** $T \subseteq \binom{[n]}{t-2}$ **do**

**21**             **for** $a, b \in I_j$ **do**

**22**                 $\gamma^{(1)}, \ldots, \gamma^{(t)} \leftarrow$ FindOrthonormalization($(p_i : i \in T \cup \{a, b\})$)

**23**                 Write $\gamma^{(1)}, \ldots, \gamma^{(t)}$ as sparse vectors in $\mathbb{R}^n$ (supported on $T \cup \{a, b\}$)

**24**                 Add $\gamma^{(1)}, \ldots, \gamma^{(t)}$ to $\mathcal{D}$

**25**     **return** $\mathcal{D}$

**26** **Procedure** ComputeL1Representation($\{X_1, \ldots, X_m\}, t$)

**27**     Let $\mathbb{X} : m \times n$ be the matrix with rows $X_1, \ldots, X_m$

**28**     Let $q_1, \ldots, q_n$ be the columns of $\mathbb{X}$, and let $p_i := q_i / \|q_i\|_2$ for $i \in [n]$

**29**     $\tilde{\mathcal{D}} \leftarrow$ RepresentVectors($\{p_1, \ldots, p_n\}, t, e^{-m}$)

**30**     $\hat{D} \leftarrow \text{diag}(\|q_1\|_2, \ldots, \|q_n\|_2)$

**31**     $\mathcal{D} \leftarrow \{\hat{D}d : d \in \tilde{\mathcal{D}}\}$

**32**     **return** $\mathcal{D}$

---

to find a dictionary $\mathcal{D}$ of $O(n^{t-1/2})$ sparse combinations of $\{p_1, \ldots, p_n\}$ so that *every* $t$-sparse combination of $\{p_1, \ldots, p_n\}$ can be written in terms of the chosen combinations, with a coefficient vector that has bounded $\ell_1$ norm.

For notational ease, we define $C(x)$ to be the "cost" of a particular linear combination $x \in \mathbb{R}^n$ with respect to the set $\mathcal{D}$ of chosen combinations:

**Definition D.2.** For a subset $\mathcal{D} \subseteq \mathbb{R}^n$, define $C_{\mathcal{D}} : \mathbb{R}^n \to [0, \infty]$ by

$$C_{\mathcal{D}}(x) := \min_{\alpha \in \mathbb{R}^{\mathcal{D}} : \sum_{d \in \mathcal{D}} \alpha_d d = x} \sum_{d \in \mathcal{D}} |\alpha_d| \cdot \left\| \sum_{i=1}^{n} d_i p_i \right\|_2 .$$

With this notation, we want to construct a set $\mathcal{D}$ of size $O(n^{t-1/2})$, consisting of $t$-sparse vectors, such that

$$C_{\mathcal{D}}(x) \leq \text{poly}(t, \log n) \cdot \left\| \sum x_i p_i \right\|_2$$

for all $t$-sparse $x \in \mathbb{R}^n$.

The construction is quite simple: divide the set $\{p_1, \ldots, p_n\}$ into $\sqrt{n}$ equal-sized groups. For each set $T$ of $t-1$ vectors and each of the $\sqrt{n}$ groups, find the closest vector in the group to the subspace spanned by $T$ (using Theorem D.1 to achieve sublinear time complexity). Then add the difference between the vector and its projection (onto the subspace) to the dictionary. Finally, for each set of $t$ vectors where two of the vectors lie in the same group, add an orthonormal basis for those vectors to the dictionary. See the procedure `RepresentVectors()` in Algorithm 5 for pseudocode.

By construction, the dictionary clearly has size $O(n^{t-1/2})$. At a high level, the reason it satisfies the representational property is the following. Consider some $t$-sparse combination, such as $p_1 + \cdots + p_t$. If $-p_t$ is not very close to $p_1 + \cdots + p_{t-1}$, then we can bound $C(p_1 + \cdots + p_t)$ by $C(p_1 + \cdots + p_{t-1})$ and $C(p_t)$, which are $O(\sqrt{t} \| p_1 + \cdots + p_{t-1} \|_2)$ and $O(\sqrt{t} \| p_t \|_2)$ respectively, since the dictionary contains an orthonormal basis for both terms. The only case where these bounds are not good enough is when $\| p_1 + \cdots + p_t \|_2$ is much smaller than $\| p_1 + \cdots + p_{t-1} \|_2$ and $\| p_t \|_2$. In this case, $p_t$ is very close to $\text{span}\{p_1, \ldots, p_{t-1}\}$. However, in the construction we found some (potentially different) $p_j$ which is just as close to $\text{span}\{p_1, \ldots, p_{t-1}\}$, and moreover is in the same group as $p_t$. Letting $q$ be the projection of $p_j$ onto $\text{span}\{p_1, \ldots, p_{t-1}\}$, we have the crucial fact that $\| p_j - q \|_2$ is as small as $\| p_1 + \cdots + p_t \|_2$.

Now, bounding $C(p_1 + \cdots + p_t)$ proceeds as follows. We can subtract some appropriate (bounded) multiple of $p_j - q$ from $p_1 + \cdots + p_t$ to zero out at least one of the coefficients. This residual then is a $t$-sparse combination of $\{p_1, \ldots, p_t, p_j\}$ where two of the vectors $\{p_t, p_j\}$ are in the same group; thus it has small cost with respect to $\mathcal{D}$. Moreover, $p_j - q$ is contained in $\mathcal{D}$ and thus has small cost (specifically, not much more than $\| p_j - q \|_2$, which crucially is not much more than $\| p_1 + \cdots + p_t \|_2$). It follows that $p_1 + \cdots + p_t$ has small cost.

Formalizing this argument, we start by proving one of the facts that we freely used above: that the cost function $C$ satisfies the triangle inequality.

**Fact D.3.** *For any $\mathcal{D} \subseteq \mathbb{R}^n$ and $x, y \in \mathbb{R}^n$, it holds that $C(x+y) \leq C(x) + C(y)$.*

*Proof.* For any $\alpha, \beta \in \mathbb{R}^{\mathcal{D}}$ with $\sum_d \alpha_d d = x$ and $\sum_d \beta_d d = y$, the vector $\alpha + \beta$ satisfies $\sum_d (\alpha + \beta)_d d = x + y$. Applying the triangle inequality to $\sum_d |(\alpha + \beta)_d| \cdot \| \sum_i d_i p_i \|_2$ completes the proof. $\qquad\square$

We now prove the key lemma, formalizing the above intuition.

**Lemma D.4.** *Let $n, m, t \in \mathbb{N}$, with $t \geq 2$, and $\delta > 0$. Fix $p_1, \ldots, p_n \in \mathbb{R}^m$ with $\| p_i \|_2 = 1$ for all $i \in [n]$. Let $\mathcal{D}$ be the output of `RepresentVectors`$(\{p_1, \ldots, p_n\}, t, \delta)$. Then $|\mathcal{D}| = O(n^{t-1/2})$, and every element of $\mathcal{D}$ is $t$-sparse. Also, with probability at least $1 - \delta$, the following guarantees hold. The time complexity of computing $\mathcal{D}$ is $O(n^{t-\Omega(1/t)} (\log n)^{O(t)} m^{O(1)} \log(1/\delta))$. Moreover, for every $t$-sparse $x \in \mathbb{R}^n$ it holds that*

$$C_{\mathcal{D}}(x) \leq O(t^{3/2} \log n) \cdot \left\| \sum x_i p_i \right\|_2.$$

*Proof.* Since the algorithm `RepresentVectors()` makes less than $n^t$ queries to the data structures $\mathcal{N}^j$, and each query has failure probability at most $\delta' = \delta/n^t$, the probability that any query fails is at most $1 - \delta$. We henceforth assume that all queries succeed, i.e. satisfy the correctness guarantee and time complexity bound stated in Theorem D.1.

**Time complexity.** We start by analyzing the time complexity of `RepresentVectors`$(\{p_1, \ldots, p_n\}, t, \delta)$. For any fixed $j \in [\sqrt{n}]$, the construction time of $\mathcal{N}^j$ (with $|I_j| = O(\sqrt{n})$ points in $\mathbb{R}^m$, query dimension $t-1$, and failure probability $\delta/n^t$) is $O(n^{t/2} (\log n)^{O(t)} m^{O(1)} \log(1/\delta))$. We make $\binom{n}{t-1} + |I_j|^2 \binom{n}{t-2} = O(n^{t-1})$ queries to $\mathcal{N}^j$, each with time complexity $n^{1/2 - 1/(4(t-1))} m^{O(1)} \log(1/\delta)$. Each projection step and each orthonormalization step has time complexity $\text{poly}(t, m)$. Thus, since $t \geq 2$, the time complexity for any

fixed $j$ is bounded by $n^{t-1/2-1/(8t)}(\log n)^{O(t)}m^{O(1)}\log(1/\delta)$. Summing over $j$, the overall time complexity to compute $\mathcal{D}$ is at most $n^{t-1/(8t)}(\log n)^{O(t)}m^{O(1)}\log(1/\delta)$ as claimed.

**Correctness.** The bound on $|\mathcal{D}|$ and the fact that all elements of $\mathcal{D}$ are $t$-sparse are immediate from the algorithm definition. It remains to bound $C_{\mathcal{D}}(x)$ for $t$-sparse vectors $x$. First, note that for any $(t-1)$-sparse $y \in \mathbb{R}^n$, because of step (4), the dictionary contains vectors $\gamma^1, \ldots, \gamma^{t-1}$ that span $\mathrm{supp}(y)$ and satisfy $\langle \sum_{i=1}^n \gamma_i^k p_i, \sum_{i=1}^n \gamma_i^\ell p_i \rangle = 0$ for all $k \neq \ell$. Thus, letting $\alpha_1, \ldots, \alpha_{t-1} \in \mathbb{R}$ be such that $y = \alpha_1 \gamma^1 + \cdots + \alpha_{t-1}\gamma^{t-1}$, we get

$$C_{\mathcal{D}}(y) \leq \sum_{j=1}^{t-1} |\alpha_j| \cdot \left\| \sum_{i=1}^n \gamma_i^j p_i \right\|_2 \leq \sqrt{t}\sqrt{\sum_{j=1}^{t-1} \alpha_j^2 \left\| \sum_{i=1}^n \gamma_i^j p_i \right\|_2^2} = \sqrt{t}\left\| \sum_{i=1}^n y_i p_i \right\|_2. \tag{7}$$

Now fix any nonzero $t$-sparse $x \in \mathbb{R}^n$. Fix any $a \in \arg\max_{i \in [n]} |x_i|$, and let $j \in [\sqrt{n}]$ be such that $a \in I_j$. Let $T = \mathrm{supp}(x) \setminus \{a\}$. Let $q := \mathrm{Proj}_{\mathrm{span}\{p_i : i \in T\}} p_{h(T,j)}$. Then by the correctness guarantee of $\mathcal{N}^j$ on query $\mathrm{span}\{p_i : i \in T\}$,

$$\left\| p_{h(T,j)} - q \right\|_2 \leq O(\log n) \cdot \left\| p_a + \sum_{i \neq a} \frac{x_i}{x_a} p_i \right\|_2 = O(\log n) \cdot \frac{\|\sum_i x_i p_i\|_2}{|x_a|}. \tag{8}$$

**Case I.** Suppose that $\left\| p_{h(T,j)} - q \right\|_2 \geq 1/2$. Then by (8), we have $|x_a| \leq O(\log n) \cdot \|\sum_i x_i p_i\|_2$. Thus, by the triangle inequality,

$$\left\| \sum_{i \neq a} x_i p_i \right\|_2 \leq |x_a| + \left\| \sum_i x_i p_i \right\|_2 \leq O(\log n) \cdot \left\| \sum_i x_i p_i \right\|_2.$$

It follows from Fact D.3 and (7) that

$$C_{\mathcal{D}}(x) \leq C_{\mathcal{D}}(x_a e_a) + C_{\mathcal{D}}(x - x_a e_a) \leq \sqrt{t}|x_a| + \sqrt{t}\left\| \sum_{i \neq a} x_i p_i \right\|_2 \leq O(\sqrt{t}\log n) \cdot \left\| \sum_i x_i p_i \right\|_2$$

as desired.

**Case II.** It remains to consider the case that $\left\| p_{h(T,j)} - q \right\|_2 \leq 1/2$. In this case we have $\|q\|_2 \geq \|p_{h(T,j)}\|_2 - 1/2 \geq 1/2$. By step (3) of the algorithm, the dictionary contains some vector $\gamma - e_{h(T,j)}$ such that $\mathrm{supp}(\gamma) \subseteq T$ and $q = \sum_{i \in T} \gamma_i p_i$. Fix any $b \in \arg\max_i |\gamma_i|$. Since $q = \sum \gamma_i p_i$ we get $|\gamma_b| \geq \frac{\|q\|_2}{t} \geq 1/(2t)$. Now, by Fact D.3,

$$C_{\mathcal{D}}(x) \leq C_{\mathcal{D}}\left( -\frac{x_b}{\gamma_b}(e_{h(T,j)} - \gamma) \right) + C_{\mathcal{D}}\left( x + \frac{x_b}{\gamma_b}(e_{h(T,j)} - \gamma) \right).$$

By construction, $e_{h(T,j)} - \gamma$ is an element of the dictionary, so we can bound the first term as

$$C_{\mathcal{D}}\left( -\frac{x_b}{\gamma_b}(e_{h(T,j)} - \gamma) \right) \leq \frac{|x_b|}{|\gamma_b|}\left\| \sum_{i=1}^n (e_{h(T,j)} - \gamma)_i p_i \right\|_2$$

$$= \frac{|x_b|}{|\gamma_b|}\left\| p_{h(T,j)} - q \right\|_2$$

$$\leq 2t|x_a|\left\| p_{h(T,j)} - q \right\|_2$$

$$\leq O(t\log n)\left\| \sum_{i=1}^n x_i p_i \right\|_2$$

where the equality uses that $q = \sum_{i=1}^n \gamma_i p_i$, the second inequality uses that $|x_b| \leq |x_a|$ and $|\gamma_b| \geq 1/(2t)$, and the final inequality uses (8).

Finally, observe that

$$z := x + \frac{x_b}{\gamma_b}(e_{h(T,j)} - \gamma) = x_a e_a + \frac{x_b}{\gamma_b} e_{h(T,j)} + \sum_{i \in T \setminus \{a,b\}} \left( x_i - \frac{x_b \gamma_i}{\gamma_b} \right) e_i$$

since the coefficients on $e_b$ cancel out. Thus, $z$ is a linear combination of two elements of $\{p_i : i \in I_j\}$ together with $t - 2$ elements of $\{p_i : i \in [n]\}$. Because of step (4) of the algorithm, the dictionary contains vectors $\gamma^1, \ldots, \gamma^t$ that span $\operatorname{supp}(z)$ and satisfy $\langle \sum_{i=1}^{n} \gamma_i^k p_i, \sum_{i=1}^{n} \gamma_i^\ell p_i \rangle = 0$ for all $k \neq \ell$. The same argument as for (7) gives that

$$
\begin{aligned}
C_{\mathcal{D}}\left( x + \frac{x_b}{\gamma_b}(e_{h(T,j)} - \gamma) \right) &\leq \sqrt{t} \left\| \sum_{i=1}^{n} x_i p_i + \frac{x_b}{\gamma_b}(p_{h(T,j)} - q) \right\|_2 \\
&\leq \sqrt{t} \left\| \sum_{i=1}^{n} x_i p_i \right\|_2 + O(\sqrt{t} \log n) \frac{|x_b|}{|\gamma_b||x_a|} \left\| \sum_{i=1}^{n} x_i p_i \right\|_2 \\
&\leq O(t^{3/2} \log n) \left\| \sum_{i=1}^{n} x_i p_i \right\|_2
\end{aligned}
$$

where the second inequality uses the triangle inequality and (8), and the final inequality uses that $|x_b| \leq |x_a|$ and $|\gamma_b| \geq 1/(2t)$. Putting everything together, we conclude that

$$C_{\mathcal{D}}(x) \leq O(t^{3/2} \log n) \left\| \sum_{i=1}^{n} x_i p_i \right\|_2$$

as claimed. □

We now show that $\texttt{RepresentVectors()}$ can be applied to the columns of the sample matrix to obtain a $\ell_1$-representation for $\Sigma$ (procedure $\texttt{ComputeL1Representation()}$ in Algorithm 5). Up to an appropriate rescaling of the covariates, Lemma D.4 immediately implies that $\mathcal{D}$ gives a $\ell_1$-representation for the empirical covariance $\hat{\Sigma}$. The main result then follows from concentration of $\hat{\Sigma}$ and sparsity of the elements of the dictionary.

**Theorem D.5.** *Let $n, m, t \in \mathbb{N}$ and let $\Sigma : n \times n$ be a positive-definite matrix. Suppose $m \geq Ct \log n$ for a sufficiently large constant $C$. Let $X_1, \ldots, X_m \sim N(0, \Sigma)$ be independent samples, and let $\mathcal{D}$ be the output of $\texttt{ComputeL1Representation}(\{X_1, \ldots, X_m\}, t)$. Then $|\mathcal{D}| \leq O(n^{t-1/2})$, and every element of $\mathcal{D}$ is $t$-sparse. Also, with probability at least $1 - e^{-\Omega(m)}$, the time complexity of the algorithm is $O(n^{t-\Omega(1/t)}(\log n)^{O(t)} m^{O(1)})$, and $\mathcal{D}$ is a $(t, C_{l1rep} t^{3/2} \log n)$-$\ell_1$-representation for $\Sigma$, for some universal constant $C_{l1rep}$.*

*Proof.* Let $\hat{\Sigma} = \frac{1}{m} \mathbb{X}^\top \mathbb{X}$. Let $\tilde{\mathcal{D}}$ denote the intermediary dictionary constructed by the algorithm using $\texttt{RepresentVectors()}$. With probability at least $1 - e^{-m}$, the successful event of Lemma D.4 holds. By standard concentration bounds (see e.g. Exercise 4.7.3 in [41]), it holds that $\frac{1}{2} \|x\|_\Sigma \leq \|x\|_{\hat{\Sigma}} \leq 2\|x\|_\Sigma$ for all $t$-sparse $x \in \mathbb{R}^n$, with probability at least $1 - e^{-\Omega(m)}$. Henceforth assume that both of these events hold.

**Time complexity.** The time complexity of the algorithm is dominated by the call to $\texttt{RepresentVectors()}$. By the guarantee of Lemma D.4, this takes time $O(n^{t-\Omega(1/t)}(\log n)^{O(t)} m^{O(1)})$.

**Correctness.** The bounds on $|\mathcal{D}|$ and sparsity of elements of $\mathcal{D}$ follow from identical bounds for $\tilde{\mathcal{D}}$ (see Lemma D.4), and the fact that every element of $\mathcal{D}$ is obtained by rescaling the coordinates of some element of $\tilde{\mathcal{D}}$. It remains to show that $\mathcal{D}$ is a $(t, O(t^{3/2} \log n))$-$\ell_1$ representation for $\Sigma$.

Fix any $t$-sparse $v \in \mathbb{R}^n$, and define $\tilde{v} = \hat{D}v$. By the guarantee of Lemma D.4, since $\tilde{v}$ is also $t$-sparse, there is some $\alpha \in \mathbb{R}^{\hat{\mathcal{D}}}$ such that $\tilde{v} = \sum_{\tilde{d} \in \tilde{\mathcal{D}}} \alpha_{\tilde{d}} \tilde{d}$ and

$$\sum_{\tilde{d} \in \tilde{\mathcal{D}}} |\alpha_{\tilde{d}}| \cdot \left\| \sum_{i=1}^{n} \tilde{d}_i \frac{q_i}{\|q_i\|_2} \right\|_2 \leq O(t^{3/2} \log n) \cdot \left\| \sum_{i=1}^{n} \tilde{v}_i \frac{q_i}{\|q_i\|_2} \right\|_2 .$$

---

**Algorithm 5:** Sparse linear regression for arbitrary $\Sigma$

---

1 **Procedure** SparseLinearRegression($(X_i, y_i)_{i=1}^m$, $t$, $B$, $\sigma^2$)

2    $\mathcal{D} \leftarrow$ ComputeL1Representation($\{X_1, \ldots, X_{100t \log n}\}, t$)

3    **for** $j = m/2 + 1, \ldots, m$ **do**

4       **for** $d \in \mathcal{D}$ **do**

5          $\tilde{X}_{j,d} \leftarrow \left\langle X_j, d / \sqrt{(2/m) \sum_{i=1}^{m/2} \langle X_i, d \rangle^2} \right\rangle$

      /* See Theorem A.7 for definition of MirrorDescentLasso(), and
      Theorem D.5 for definition of $C_{\mathsf{l1rep}}$                 */

6    $\hat{\beta} \leftarrow$ MirrorDescentLasso($(\tilde{X}_i, y_i)_{i=m/2+1}^m$, $2C_{\mathsf{l1rep}} t^{3/2} B \log(n)$, $m/2$, $\sigma^2$)

7    $\hat{w} \leftarrow \sum_{d \in \mathcal{D}} \hat{\beta}_d d / \sqrt{(2/m) \sum_{i=1}^{m/2} \langle X_i, d \rangle^2}$

8    **return** $\hat{w}$

---

But note that $\tilde{v}_i = \hat{D}_{ii} v_i = \|q_i\|_2 v_i$ for all $i$. Similarly, every $\tilde{d} \in \tilde{\mathcal{D}}$ corresponds to some $d \in \mathcal{D}$ with $\tilde{d}_i = \|q_i\|_2 d_i$ for all $i$. Thus, reindexing $\alpha$ according to $\mathcal{D}$ in the natural way, we have that $v = \sum_{d \in \mathcal{D}} \alpha_d d$ and

$$\sum_{d \in \mathcal{D}} |\alpha_d| \cdot \left\| \sum_{i=1}^n d_i q_i \right\|_2 \leq O(t^{3/2} \log n) \cdot \left\| \sum_{i=1}^n v_i q_i \right\|_2 .$$

But now let $\hat{\Sigma} = \frac{1}{m} \mathbb{X}^\top \mathbb{X}$. For any $i, j \in [n]$ we have $\langle q_i, q_j \rangle = m \hat{\Sigma}_{ii}$. Hence,

$$\left\| \sum_{i=1}^n v_i q_i \right\|_2^2 = \sum_{i,j \in [n]} v_i v_j \hat{\Sigma}_{ij} = v^\top \hat{\Sigma} v$$

and similarly for $\| \sum_{i=1}^n d_i q_i \|_2^2$. Thus, we get

$$\sum_{d \in \mathcal{D}} |\alpha_d| \cdot \|d\|_{\hat{\Sigma}} \leq O(t^{3/2} \log n) \cdot \|v\|_{\hat{\Sigma}} .$$

But as shown above, we know that $\frac{1}{2} \|x\|_\Sigma \leq \|x\|_{\hat{\Sigma}} \leq 2 \|x\|_\Sigma$ for all $t$-sparse $x \in \mathbb{R}^n$. Since $v$ and all $d \in \mathcal{D}$ are $t$-sparse, we conclude that

$$\sum_{d \in \mathcal{D}} |\alpha_d| \cdot \|d\|_\Sigma \leq O(t^{3/2} \log n) \cdot \|v\|_\Sigma$$

as desired. $\qquad\qquad\qquad\qquad\qquad\qquad\qquad\qquad\qquad\qquad\qquad\qquad\qquad\qquad\qquad\qquad$ $\square$

We finally restate and prove Theorem 3.1, as a corollary of Theorem D.5 and the well-known fact that standard "slow rate" guarantees for Lasso (i.e. based on the $\ell_1$ norm of the regressor) can be achieved in near-linear time (Theorem A.8). The pseudocode for the main algorithm is given in Algorithm 5.

**Corollary D.6.** *Let $n, m, t, B \in \mathbb{N}$ and $\sigma > 0$, and let $\Sigma : n \times n$ be a positive-definite matrix. Let $w^* \in \mathbb{R}^n$ be $t$-sparse with $\|w^*\|_\Sigma \leq B$. Suppose $m \geq Ct \log n$ for a sufficiently large constant $C$. Let $(X_i, y_i)_{i=1}^m$ be independent samples where $X_i \sim N(0, \Sigma)$ and $y_i = \langle X_i, w^* \rangle + N(0, \sigma^2)$. Then there is an $O(m^2 n t^{-1/2} + n^{t - \Omega(1/t)} \log^{O(t)} n)$-time algorithm (Algorithm 5) that, given $(X_i, y_i)_{i=1}^m$, $t$, $B$, $\sigma^2$, produces an estimate $\hat{w} \in \mathbb{R}^n$ satisfying, with probability $1 - o(1)$,*

$$\|\hat{w} - w^*\|_\Sigma^2 \leq \tilde{O} \left( \frac{\sigma^2}{\sqrt{m}} + \frac{\sigma B t^{3/2}}{\sqrt{m}} + \frac{B^2 t^3}{m} \right) .$$

*Proof.* By Theorem D.5 it holds with probability $1 - n^{-100t}$ that $\mathcal{D}$ is a $(t, C_{\mathsf{l1rep}} t^{3/2} \log n)$-$\ell_1$-representation for $\Sigma$. Also, by standard concentration bounds (e.g. Exercise 4.7.3 in [41]), we have

$\frac{1}{2} \|x\|_\Sigma \le \|x\|_{\hat\Sigma} \le 2 \|x\|_\Sigma$ for all $t$-sparse $x \in \mathbb{R}^n$ (where $\hat\Sigma = \frac{2}{m} \sum_{i=1}^{m/2} X_i X_i^\top$) with probability at least $1 - \exp(-\Omega(m))$. Suppose that both of these events occur.

For each of the remaining $m/2$ samples $X_j$, compute $\tilde X_j \in \mathbb{R}^\mathcal{D}$ where the entry $\tilde X_{j,d}$ corresponding to $d \in \mathcal{D}$ is $\langle X_j, d/\|d\|_{\hat\Sigma}\rangle$ (where $\hat\Sigma = \frac{2}{m} \sum_{i=1}^{m/2} X_i X_i^\top$ is not explicitly computed; since $d$ is sparse, both $\langle X_j, d\rangle$ and $\|d\|_{\hat\Sigma}$ can be computed in $\mathrm{poly}(t,m)$ time). Let $N(0,\Gamma)$ denote the distribution of each $\tilde X_j$. For each $d \in \mathcal{D}$, since $d$ is $t$-sparse, we have that $\mathbb{E}_{x \sim N(0,\Sigma)}\langle x, d/\|d\|_{\hat\Sigma}\rangle^2 = \|d\|_\Sigma^2 / \|d\|_{\hat\Sigma}^2 \le 4$. Thus, $\Gamma_{dd} \le 4$ for all $d$.

Moreover, since $w^*$ is $t$-sparse, there is some $\alpha \in \mathbb{R}^\mathcal{D}$ with $w^* = \sum_d \alpha_d d$ and $\sum_d |\alpha_d| \|d\|_\Sigma \le C_{\mathsf{l1rep}} t^{3/2} \log(n) \cdot \|w^*\|_\Sigma$. Define $\beta \in \mathbb{R}^d$ by $\beta_d = \alpha_d \|d\|_{\hat\Sigma}$. Then $w^* = \sum_d \beta_d d/\|d\|_{\hat\Sigma}$ and

$$\sum_d |\beta_d| \le 2 \cdot \sum_d |\alpha_d| \|d\|_\Sigma \le 2C_{\mathsf{l1rep}} t^{3/2} \log(n) \cdot \|w^*\|_\Sigma.$$

But now for any of the remaining $m/2$ samples, we have that

$$\langle \tilde X_j, \beta\rangle = \sum_d \langle X_j, d/\|d\|_{\hat\Sigma}\rangle \alpha_d \|d\|_{\hat\Sigma} = \langle X_j, \sum_d \alpha_d d\rangle = \langle X_j, w^*\rangle,$$

and thus $y - \langle \tilde X_j, \beta\rangle \sim N(0,\sigma^2)$. So we can apply Theorem A.8 to samples $(\tilde X_j, y_j)_{j=m/2+1}^m$ to compute an estimator $\hat\beta$ satisfying

$$\left\|\hat\beta - \beta\right\|_\Gamma^2 \le \tilde O\left(\frac{\sigma^2}{m} + \frac{\sigma B t^{3/2}}{\sqrt{m}} + \frac{B^2 t^3}{m}\right)$$

using that $\|\beta\|_1 \le 2C_{\mathsf{l1rep}} t^{3/2} \log(n) \cdot \|w^*\|_\Sigma \le 2C_{\mathsf{l1rep}} t^{3/2} B \log(n)$, and using the bound $\max_d \Gamma_{dd} \le 4$. The time complexity of this step is $\tilde O(|\mathcal{D}|m^2) = \tilde O(m^2 n^{t-1/2})$. Finally, compute $\hat w := \sum_d \hat\beta_d d/\|d\|_{\hat\Sigma}$. We have that $\|\hat w - w^*\|_\Sigma = \left\|\hat\beta - \beta\right\|_\Gamma$, which completes the proof. $\square$

## E   Fixed-parameter tractability in $\kappa$ and $t$

In this section we prove Theorem 3.2, which shows we can achieve upper bounds on $\mathcal{N}_{t,\alpha}(\Sigma)$ for $\alpha$ independent of $\kappa$ and $n$, if we are willing to incur dependence on $\kappa$ in the resulting bound. In fact, we actually prove an upper bound on the packing number $\mathcal{P}_{t,\alpha}(\Sigma)$.

To achieve this, the first key idea is to consider the dual certificates for a packing. Suppose that $v_1, \ldots, v_N$ are unit vectors (in the $\Sigma$-norm) with $|\langle v_i, v_j\rangle_\Sigma| \le \alpha$ for all $i \ne j$. Then $|\langle v_i, \Sigma v_i\rangle| \ge \alpha^{-1} \max_{j \ne i} |\langle v_j, \Sigma v_i\rangle|$, so $\Sigma v_i$ certifies that any linear combination $v_i = \sum_{j \ne i} x_j v_j$ must have the property that $\|x\|_1 \ge \alpha^{-1}$. Thus, to show that there cannot be a large packing of sparse vectors in the $\Sigma$-norm, it would suffice to prove that any large set of sparse vectors must have one vector that can be written as a linear combination of the remaining vectors, where the coefficient vector has small $\ell_1$ norm. In fact, this would give an upper bound on $\mathcal{N}_{t,\alpha}(\Sigma)$ for all $\Sigma$.

We do not know if such a statement is true. However, we can prove an *approximate* analogue. The following lemma shows that under a condition number bound on $\Sigma$, the dual certificate argument can be generalized to require only a weaker property: that any large set of sparse vectors must have one vector that can be *approximately* written as a linear combination of the remaining vectors, with low $\ell_1$ cost. The approximation error determines how small the condition number must be:

**Lemma E.1.** *Let $n, N, t, T \in \mathbb{N}$ and let $\delta > 0$. Suppose that for all $t$-sparse vectors $v_1, \ldots, v_N \in \mathbb{R}^n$, there exists some $i \in [N]$ and $x \in \mathbb{R}^N$ such that $\|x\|_1 \le T$ and*

$$\left\|v_i - \sum_{j \ne i} x_j v_j\right\|_2 \le \delta \cdot \max_{j \in [N]} \|v_j\|_2.$$

*Then for every positive-definite matrix $\Sigma : n \times n$ with $\kappa(\Sigma) < 1/(4\delta^2)$ it holds that $\mathcal{P}_{t,1/(3T)}(\Sigma) \le N \log_2 \kappa(\Sigma)$.*

*Proof.* Fix a positive-definite matrix $\Sigma : n \times n$ and suppose that $K := \mathcal{P}_{t,1/(3T)}(\Sigma) > N \log_2 \kappa(\Sigma)$. By definition, there are nonzero $t$-sparse vectors $v_1, \ldots, v_K \in \mathbb{R}^N$ such that

$$|\langle v_i, v_j \rangle_\Sigma| \leq \frac{1}{3T} \|v_i\|_\Sigma \|v_j\|_\Sigma$$

for all $i \neq j$. Without loss of generality, assume that $\|v_i\|_2 = 1$ for all $i \in [K]$, so that

$$\lambda_{\min}(\Sigma) \leq \|v_i\|_\Sigma^2 \leq \lambda_{\max}(\Sigma).$$

So we can partition $[K]$ into $\log_2 \kappa(\Sigma)$ buckets such that $\max_{i \in B} \|v_i\|_\Sigma^2 / \min_{i \in B} \|v_i\|_\Sigma^2 \leq 2$ for each bucket $B \subseteq [K]$. There must be some bucket $B$ with $|B| \geq N$. By assumption, there is some $i \in B$ and $x \in \mathbb{R}^N$ such that $\|x\|_1 \leq T$ and

$$\left\| v_i - \sum_{j \in B : j \neq i} x_j v_j \right\|_2 \leq \delta.$$

Now

$$
\begin{aligned}
\langle v_i, v_i \rangle_\Sigma &= \left\langle \Sigma v_i, \sum_{j \in B : j \neq i} x_j v_j \right\rangle + \left\langle \Sigma v_i, v_i - \sum_{j \in B : j \neq i} x_j v_j \right\rangle \\
&= \sum_{j \in B : j \neq i} x_j \langle v_i, v_j \rangle_\Sigma + \left\langle \Sigma v_i, v_i - \sum_{j \in B : j \neq i} x_j v_j \right\rangle \\
&\leq \|x\|_1 \max_{j \in B : j \neq i} |\langle v_i, v_j \rangle_\Sigma| + \left\| v_i^\top \Sigma \right\|_2 \cdot \delta \\
&\leq \frac{\|x\|_1}{3T} \max_{j \in B : j \neq i} \|v_i\|_\Sigma \|v_j\|_\Sigma + \delta \sqrt{\lambda_{\max}(\Sigma) \cdot v_i^\top \Sigma v_i} \\
&\leq \frac{\sqrt{2} \|v_i\|_\Sigma^2}{3} + \|v_i\|_\Sigma \delta \sqrt{\lambda_{\max}(\Sigma)}.
\end{aligned}
$$

Simplifying, we get $\|v_i\|_\Sigma \leq 2\delta \sqrt{\lambda_{\max}(\Sigma)}$. Since also $\|v_i\|_\Sigma \geq \sqrt{\lambda_{\min}(\Sigma)}$, it follows that $\kappa(\Sigma) = \lambda_{\max}(\Sigma)/\lambda_{\min}(\Sigma) \geq 1/(4\delta^2)$. $\qquad\square$

It remains to show that the precondition of Lemma E.1 can be satisfied for sub-constant $\delta$ without requiring $N$ to scale with $n^t$. We start by proving the desired property when the vectors are all $t$-sparse and *binary*, i.e. $v_1, \ldots, v_N \in \{0, 1\}^n$, and afterwards we will black-box extend the result to the real-valued setting. Concretely, given sparse binary vectors $v_1, \ldots, v_N \in \{0, 1\}^n$ (with $N \gg n$), we want to find one that can be "efficiently" approximated (in $\ell_2$ norm) by the rest, where "efficient" means that the coefficients have small absolute sum. Thinking of each vector as the indicator vector of a subset of $[n]$, a first step towards an efficient approximation for $v_i = \mathbb{1}[\cdot \in S_i]$ may be constructing an efficient approximation for a standard basis vector $e_j$ for some $j \in S_i$.

Indeed, there is some $j \in [n]$ such that $\mathcal{S}^j := \{i : v_{ij} = 1\}$ is large, i.e. $|\mathcal{S}^j| \geq N/n$. If the vectors $(v_i)_{i \in \mathcal{S}^j}$ were in some sense random, then the average $\frac{1}{|\mathcal{S}^j|} \sum_{i \in \mathcal{S}^j} v_i$ would be a good approximation for $e_j$. It is also efficient, in that the absolute sum of coefficients is 1. But of course the vectors are not random; it could be that many vectors in $\mathcal{S}^j$ also contain some other coordinate $j'$. In this case we restrict to the set of vectors containing both $j$ and $j'$. Now we may hope to approximate the vector $\mathbb{1}[\cdot \in \{j, j'\}]$. Completing this argument, we get the following lemma which states that there exists a subset of $[n]$ that is contained in many of the vectors, and that is well-approximated by the average of those vectors.

For notational convenience, for vectors $x, y \in \{0, 1\}^n$ we say that $x \preceq y$ if $x_i \leq y_i$ for all $i \in [n]$.

**Lemma E.2.** *Let* $n, N, t, s \in \mathbb{N}$ *with* $sn \leq N$, *and let* $v_1, \ldots, v_N \in \{0, 1\}^n$ *be nonzero $t$-sparse binary vectors. Then there is some set* $S \subseteq [N]$ *of size* $|S| \geq s$ *and some nonzero vector* $u \in \{0, 1\}^n$ *such that* $u \preceq v_i$ *for all* $i \in S$, *and*

$$\left\| u - \frac{1}{|S|} \sum_{i \in S} v_i \right\|_2 \leq \sqrt{t(sn/N)^{1/t}}.$$

*Proof.* For each $J \subseteq [n]$, define $\mathcal{S}^J := \{i \in [N] : v_{ij} = 1 \quad \forall j \in J\}$. Since all $v_i$ are nonzero, there is some $j^* \in [n]$ with $|\mathcal{S}^{\{j^*\}}| \geq N/n$. We iteratively construct a set $J \subseteq [n]$ as follows. Initially, set $J = \{j^*\}$. While there exists some $a \in [n] \setminus J$ such that $|\mathcal{S}^{J \cup \{a\}}| > (sn/N)^{1/t}|\mathcal{S}^J|$, update $J$ to $J \cup \{a\}$ (if there are multiple such $a$, pick any one of them arbitrarily). At termination of this process, we have $|\mathcal{S}^J| > 0$. Since every $v_i$ is $t$-sparse, it must be that $|J| \leq t$. Thus, $|\mathcal{S}^J| \geq (N/n) \cdot (sn/N)^{(t-1)/t} \geq s$. Set $S := \mathcal{S}^J$ and $u := \mathbb{1}_J \in \{0,1\}^n$. By definition of $\mathcal{S}^J$, we have that $u \preceq v_i$ for all $i \in S$.

For any $j \in J$, we have $u_j = 1 = \frac{1}{|S|}\sum_{i \in S} v_{ij}$. For any $j \notin J$, we have $u_j = 0$ and

$$\left| \frac{1}{|S|}\sum_{i \in S} v_{ij} \right| = \frac{|\{i \in S : v_{ij} = 1\}|}{|S|} = \frac{|\mathcal{S}^{J \cup \{j\}}|}{|\mathcal{S}^J|} \leq (sn/N)^{1/t}$$

by construction of $J$. Thus,

$$\left\| u - \frac{1}{|S|}\sum_{i \in S} v_i \right\|_\infty \leq (sn/N)^{1/t}.$$

Additionally,

$$\left\| u - \frac{1}{|S|}\sum_{i \in S} v_i \right\|_1 \leq \left\| \frac{1}{|S|}\sum_{i \in S} v_i \right\|_1 \leq \frac{1}{|S|}\sum_{i \in S} \|v_i\|_1 \leq t.$$

By the inequality $\|x\|_2^2 \leq \|x\|_1 \|x\|_\infty$, we conclude that

$$\left\| u - \frac{1}{|S|}\sum_{i \in S} v_i \right\|_2 \leq \sqrt{t(sn/N)^{1/t}}$$

as claimed. $\qquad\square$

We now use Lemma E.2 to show that if $N$ is sufficiently large, then at least one of the vectors $v_i$ can be efficiently approximated by the rest. The proof is by induction on $t$. As a first attempt, one might use Lemma E.2 to find some $u \in \{0,1\}^n$ and some large set $S \subseteq [N]$ such that $u \preceq v_i$ for all $i \in S$, and the average of the $v_i$'s approximates $u$. Then, restrict to the vectors in $S$, and induct on the $(t-1)$-sparse residual vectors $\{v_i - u : i \in S\}$. If one of the $v_i - u$'s can be efficiently approximated by the other residuals, then since $u$ can also be efficiently approximated, we can derive an efficient approximation of $v_i$ by the remaining $v_j$'s.

This doesn't quite work, since at each step of the induction the set of vectors will become smaller by a factor of roughly $n$. However, instead of throwing away the vectors outside $S =: S^{(1)}$ we can iteratively re-apply Lemma E.2 to get disjoint sets $S^{(1)}, S^{(2)}, \ldots, S^{(m)}$, where each $S^{(a)}$ has the same property as $S$ (for some potentially different vector $u^{(a)}$). We can then induct on the residual vectors $\cup_a \{v_i - u^{(a)} : i \in S^{(a)}\}$. This suffices to efficiently approximate some $v_i$. Since we throw away fewer vectors at each step of the induction, we do not need the initial number of vectors $N$ to be as large.

We formalize the above ideas in the following theorem.

**Theorem E.3.** *Let $n, N, t \in \mathbb{N}$ and let $v_1, \ldots, v_N \in \{0,1\}^n$ be $t$-sparse binary vectors. Then there is some $i \in [N]$ and $x \in \mathbb{R}^N$ such that $\|x\|_1 \leq 3^t$ and*

$$\left\| v_i - \sum_{j \neq i} x_j v_j \right\|_2 \leq 4^t \sqrt{9t(tn/N)^{1/t}}.$$

*Proof.* We induct on $t$, observing that the case $t = 0$ is immediate. Fix $t > 0$ and $t$-sparse vectors $\{v_1, \ldots, v_N\} \in \{0,1\}^n$, and suppose that the theorem statement holds for $t - 1$. If any $v_i$ is identically zero, then the claim is trivially true with $x = 0$. If $N \leq t3^{t+1}n$ then the RHS of the desired norm bound exceeds $4^t \sqrt{t}$, so the claim is trivially true with $x = 0$ and any $i \in [N]$. Thus, we may assume that all $v_i$ are nonzero, and $N \geq t3^{t+1}n$. Applying the previous lemma with

$s := 3^{t+1} \leq N/n$ gives some $S^{(1)} \subseteq [N]$ and nonzero $u^{(1)} \in \{0,1\}^n$ such that $|S^{(1)}| \geq 3^{t+1}$ and $u^{(1)} \preceq v_i$ for all $i \in S^{(1)}$, and

$$\left\| u^{(1)} - \frac{1}{|S^{(1)}|} \sum_{i \in S^{(1)}} v_i \right\|_2 \leq \sqrt{9t(n/N)^{1/t}}.$$

If $|N| - |S^{(1)}| \geq N/t \geq 3^{t+1}n$ then we can reapply the lemma with vectors $(v_i)_{i \in [N] \setminus S^{(1)}}$ and $s := 3^{t+1}$ to get some $S^{(2)} \subseteq [N] \setminus S^{(1)}$ and $u^{(2)} \in \{0,1\}^n$. Continuing this process so long as there are at least $N/t \geq 3^{t+1}n$ remaining vectors, we can generate disjoint sets $S^{(1)}, \ldots, S^{(m)} \subseteq [N]$ and vectors $u^{(1)}, \ldots, u^{(m)} \in \{0,1\}^n$ with the following properties:

**(i)** $|S^{(1)} \cup \cdots \cup S^{(m)}| > N - N/t$

**(ii)** $|S^{(a)}| \geq 3^{t+1}$ for every $a \in [m]$

**(iii)** For every $a \in [m]$, it holds that $u^{(a)}$ is nonzero and $u^{(a)} \preceq v_i$ for all $i \in S^{(a)}$

**(iv)** For every $a \in [m]$,

$$\left\| u^{(a)} - \frac{1}{|S^{(a)}|} \sum_{i \in S^{(a)}} v_i \right\|_2 \leq \sqrt{9t(tn/N)^{1/t}}.$$

For each $a \in [m]$ and $i \in S^{(a)}$, define $v_i' := v_i - u^{(a)}$. By Property **(iii)** we have that $v_i' \in \{0,1\}^N$ and $v_i'$ is $(t-1)$-sparse. By the inductive hypothesis applied to vectors $(v_i')_{i \in S^{(1)} \cup \cdots \cup S^{(m)}}$, there is some $i \in S^{(1)} \cup \cdots \cup S^{(m)}$ and $x' \in \mathbb{R}^N$ (supported on $S^{(1)} \cup \cdots \cup S^{(m)}$) such that $\|x'\|_1 \leq 3^{t-1}$ and

$$\left\| v_i' - \sum_{j \neq i} x_j' v_j' \right\|_2 \leq 4^{t-1} \sqrt{9(t-1)((t-1)n/|S^{(1)} \cup \cdots \cup S^{(m)}|)^{1/(t-1)}}$$

$$\leq 4^{t-1} \sqrt{9t(tn/N)^{1/t}} \tag{9}$$

where the last inequality uses Property **(i)** and the bound $N \geq tn$. Of course, without loss of generality $x_i' = 0$. Let $a \in [m]$ be the unique index such that $i \in S^{(a)}$. We define $x \in \{0,1\}^N$ (supported on $S^{(1)} \cup \cdots \cup S^{(m)}$) as follows. For each $b \in [m]$ and each $r \in S^{(b)}$, set

$$x_r = x_r' - \frac{1}{|S^{(b)}|} \sum_{j \in S^{(b)}} x_j' + \frac{\mathbb{1}[b = a]}{|S^{(b)}|}.$$

Since $\|x'\|_1 \leq 3^{t-1}$, we can see that

$$\|x\|_1 \leq \|x'\|_1 + \sum_{b \in [m]} \sum_{r \in S^{(b)}} \frac{1}{|S^{(b)}|} \sum_{j \in S^{(b)}} |x_j'| + \sum_{r \in S^{(a)}} \frac{1}{|S^{(a)}|}$$

$$\leq 2\|x'\|_1 + 1$$

$$\leq 2 \cdot 3^{t-1} + 1.$$

Next, we use $x$ to approximate $v_i$. The following bound is almost what we want:

**Claim E.4.** $\left\| v_i - \sum_{j \in [N]} x_j v_j \right\|_2 \leq 3 \cdot 4^{t-1} \sqrt{9t(tn/N)^{1/t}}$

*Proof of claim.* We have

$$\left\| v_i - \sum_{r \in [N]} x_r v_r \right\|_2 \leq \left\| u^{(a)} - \frac{1}{|S^{(a)}|} \sum_{r \in S^{(a)}} v_r \right\|_2 + \left\| v_i' + \frac{1}{|S^{(a)}|} \sum_{r \in S^{(a)}} v_r - \sum_{r \in [N]} x_r v_r \right\|_2$$

$$= \left\| u^{(a)} - \frac{1}{|S^{(a)}|} \sum_{r \in S^{(a)}} v_r \right\|_2 + \left\| v_i' - \sum_{r \in [N]} x_r' v_r + \sum_{b \in [m]} \sum_{r \in S^{(b)}} \frac{1}{|S^{(b)}|} \sum_{j \in S^{(b)}} x_j' v_r \right\|_2$$

$$\leq \left\| u^{(a)} - \frac{1}{|S^{(a)}|} \sum_{r \in S^{(a)}} v_r \right\|_2 + \left\| v_i' - \sum_{r \in [N]} x_r' v_r' \right\|_2$$

$$+ \left\| - \sum_{b \in [m]} \sum_{r \in S^{(b)}} x_r' u^{(b)} + \sum_{b \in [m]} \sum_{r \in S^{(b)}} \frac{1}{|S^{(b)}|} \sum_{j \in S^{(b)}} x_j' v_r \right\|_2$$

$$= \left\| u^{(a)} - \frac{1}{|S^{(a)}|} \sum_{r \in S^{(a)}} v_r \right\|_2 + \left\| v_i' - \sum_{r \in [N]} x_r' v_r' \right\|_2$$

$$+ \left\| \sum_{b \in [m]} \sum_{j \in S^{(b)}} x_j' \left( u^{(b)} - \frac{1}{|S^{(b)}|} \sum_{r \in S^{(b)}} v_r \right) \right\|_2$$

where the first and third inequalities use that $v_r = v_r' + u^{(b)}$ for all $r \in S^{(b)}$, and throughout we use that $x_r = x_r' = 0$ for $r \notin S^{(1)} \cup \cdots \cup S^{(m)}$. Applying Property **(iv)**, equation (9), and the bound $\|x'\|_1 \leq 3^{t-1}$, we get

$$\left\| v_i - \sum_{r \in [N]} x_r v_r \right\|_2 \leq \sqrt{9t(tn/N)^{1/t}} + 4^{t-1}\sqrt{9t(tn/N)^{1/t}} + 3^{t-1}\sqrt{9t(tn/N)^{1/t}}$$

$$\leq 3 \cdot 4^{t-1}\sqrt{9t(tn/N)^{1/t}}$$

as claimed. $\qquad\square$

However, we wanted a bound on $v_i - \sum_{j \neq i} x_j v_j$, and unfortunately $x_i \neq 0$. Fortunately, it is enough that $x_i$ is bounded away from 1. Since $x_i' = 0$, we have

$$|x_i| \leq \frac{1}{|S^{(a)}|} \sum_{j \in S^{(a)}} |x_j'| + \frac{1}{|S^{(a)}|} \leq \frac{\|x'\|_1 + 1}{|S^{(a)}|} \leq \frac{3^{t-1}}{3^{t+1}} = \frac{1}{9}.$$

Thus, by Claim E.4,

$$\left\| v_i - \frac{1}{1 - x_i} \sum_{j \neq i} x_j v_j \right\|_2 \leq \frac{1}{1 - x_i} \cdot 3 \cdot 4^{t-1}\sqrt{9t(tn/N)^{1/t}} \leq 4^t\sqrt{9t(tn/N)^{1/t}}.$$

Finally, we have $\|x/(1 - x_i)\|_1 \leq (9/8)(2 \cdot 3^{t-1} + 1) \leq 3^t$, so $x/(1 - x_i)$ satisfies all the desired conditions. This completes the induction. $\qquad\square$

Finally, we extend Theorem E.3 to real-valued sparse vectors via a discretization argument.

**Lemma E.5.** *Let $n, N, t \in \mathbb{N}$ and let $v_1, \ldots, v_N \in \mathbb{R}^n$ be $t$-sparse vectors. Then there is some $i \in [N]$ and $x \in \mathbb{R}^n$ such that $\|x\|_1 \leq 3^t$ and*

$$\left\| v_i - \sum_{j \neq i} x_j v_j \right\|_2 \leq 4^{t+2}\sqrt{t}(n/N)^{1/(4t)} \cdot \max_{j \in [N]} \|v_j\|_\infty.$$

*Proof.* Without loss of generality assume that $\max_{j \in [N]} \|v_j\|_\infty = 1$. Let $k \in \mathbb{N}$ be fixed later. Define a map $\varphi : [-1, 1] \to \{0, 1\}^{2k+1}$ by

$$\varphi(c) = \begin{cases} e_{k+1+\lfloor ck \rfloor} & \text{if } c < 0 \\ e_{k+1} & \text{if } c = 0 \\ e_{k+1+\lceil ck \rceil} & \text{if } c > 0 \end{cases}.$$

Also let $\Phi : \mathbb{R}^{2k+1} \to \mathbb{R}$ be the linear map that sends $\Phi e_i \mapsto (i - k - 1)/k$ for each $i \in [2k + 1]$. Note that $|\Phi\varphi(c) - c| \leq 1/k$ for all $c \in [-1, 1]$ and $\Phi\varphi(0) = 0$. Define $\varphi^{\oplus n} : [-1, 1]^n \to \{0, 1\}^{(2k+1)n}$ by $\varphi(c_1, \ldots, c_n) = (\varphi(c_1), \ldots, \varphi(c_n))$, and define $\Phi^{\oplus n} : \{0, 1\}^{(2k+1)n} \to \mathbb{R}^n$ by $\Phi^{\oplus n}(x_1, \ldots, x_n) = (\Phi(x_1), \ldots, \Phi(x_n))$. For any $i \in [N]$, the vector $\varphi^{\oplus n}(v_i)$ is $t$-sparse and lies in $\{0, 1\}^{(2k+1)n}$. Thus, applying Theorem E.3 gives some $i \in [N]$ and $x \in \mathbb{R}^N$ with $\|x\|_1 \leq 3^t$ and

$$\left\| \varphi^{\oplus n}(v_i) - \sum_{j \neq i} x_j \varphi^{\oplus n}(v_j) \right\|_2 \leq 4^t \sqrt{9t(tn(2k+1)/N)^{1/t}}.$$

Since $\Phi^{\oplus n}$ is a linear map and $\|\Phi^{\oplus n}\|_2 = \|\Phi\|_2 \leq \sqrt{2k+1}$, we then get

$$\left\| \Phi^{\oplus n}\varphi^{\oplus n}(v_i) - \sum_{j \neq i} x_j \Phi^{\oplus n}\varphi^{\oplus n}(v_j) \right\|_2 \leq 4^t \sqrt{9t(2k+1)(tn(2k+1)/N)^{1/t}}.$$

But now for every $j \in [N]$, we know that

$$\left\| v_j - \Phi^{\oplus n}\varphi^{\oplus n}(v_j) \right\|_2^2 = \sum_{a \in \mathrm{supp}(v_j)} (v_{ja} - \Phi\varphi(v_{ja}))^2 \leq \frac{t}{k^2}.$$

We conclude that

$$\begin{aligned}
\left\| v_i - \sum_{j \neq i} x_j v_j \right\|_2 &\leq \left\| \Phi^{\oplus n}\varphi^{\oplus n}(v_i) - \sum_{j \neq i} x_j \Phi^{\oplus n}\varphi^{\oplus n}(v_j) \right\|_2 + \left\| v_i - \Phi^{\oplus n}\varphi^{\oplus n}(v_i) \right\|_2 \\
&\quad + \sum_{j \neq i} |x_j| \cdot \left\| v_j - \Phi^{\oplus n}\varphi^{\oplus n}(v_j) \right\|_2 \\
&\leq 4^t \sqrt{9t(2k+1)(tn(2k+1)/N)^{1/t}} + (1 + 3^t) \cdot \frac{\sqrt{t}}{k} \\
&\leq (2k+1) \cdot 4^{t+1} \sqrt{t}(n/N)^{1/(2t)} + \frac{4^t \sqrt{t}}{k}.
\end{aligned}$$

Taking $k = (N/n)^{1/(4t)}$ gives the claimed bound. $\qquad\square$

Combining Lemma E.5 with Lemma E.1 lets us prove Theorem 3.2.

**Proof of Theorem 3.2.** Set $\delta := \sqrt{1/(4\kappa)}$ and $N = 4^{4t(t+3)}t^{2t}\kappa^{2t}n$. By Lemma E.5, for any $t$-sparse vectors $v_1, \ldots, v_N \in \mathbb{R}^n$ with $\|v_i\|_2 \leq 1$ for all $i \in [N]$, there is some $i \in [N]$ and $x \in \mathbb{R}^n$ such that $\|x\|_1 \leq 3^t$ and

$$\left\| v_i - \sum_{j \neq i} x_j v_j \right\|_2 \leq 4^{t+2} \sqrt{t}(n/N)^{1/(4t)} \leq \frac{1}{4\sqrt{\kappa}} < \delta.$$

It follows from Lemma E.1 that $\mathcal{P}_{t,1/3^{t+1}}(\Sigma) \leq N \log_2 \kappa$. Finally, by Lemma A.2, we conclude that $\mathcal{N}_{t,1/3^{t+1}}(\Sigma) \leq N \log_2 \kappa$. $\qquad\blacksquare$

# F  Generalization bounds

## F.1  Finite-class model selection

**Lemma F.1.** *Let $n, m, n_{eff} \in \mathbb{N}$ and let $\Sigma$ be a positive semi-definite matrix. Fix a vector $w^* \in \mathbb{R}^n$ and a closed set $\mathcal{W} \subseteq \mathbb{R}^n$ and let $(X_i, y_i)_{i=1}^m$ be independent draws $X_i \sim N(0, \Sigma)$ and $y_i = \langle X_i, w^* \rangle + \xi_i$ where $\xi_i \sim N(0, \sigma^2)$. Pick*

$$\hat{w} \in \operatorname*{argmin}_{w \in \mathcal{W}} \|\mathbb{X}w - y\|_2^2$$

*where $\mathbb{X} : m \times n$ is the matrix with rows $X_1, \ldots, X_m$. For any $\epsilon, \delta \in (0, 1)$, suppose that with probability at least $1 - \delta$, the following bounds hold uniformly over $w \in \mathcal{W}$:*

1. $\left| \frac{1}{m} \left\| \mathbb{X}(w - w^*) \right\|_2^2 - \left\| w - w^* \right\|_\Sigma^2 \right| \le \epsilon \left\| w - w^* \right\|_\Sigma^2$

2. $\left| \left\langle \xi, \frac{\mathbb{X}(w-w^*)}{\|\mathbb{X}(w-w^*)\|_2} \right\rangle \right| \le \sigma \sqrt{n_{\textit{eff}}}.$

*Then with probability at least $1 - \delta$ it also holds that*

$$\|\hat{w} - w^*\|_\Sigma \le \sqrt{\frac{1+\epsilon}{1-\epsilon}} \inf_{w \in \mathcal{W}} \|w - w^*\|_\Sigma + 2\sigma \sqrt{\frac{2n_{\textit{eff}}}{m}}.$$

*Proof.* Consider the event in which both bounds hold. Let $w_{\text{opt}} \in \operatorname{argmin}_{w \in \mathcal{W}} \|w - w^*\|_\Sigma^2$. Then

$$\begin{aligned}
\|\mathbb{X}(\hat{w} - w^*)\|_2^2 &= \|\mathbb{X}\hat{w} - y\|_2^2 + 2\langle \xi, \mathbb{X}(\hat{w} - w^*) \rangle - \|\xi\|_2^2 \\
&\le \|\mathbb{X}w_{\text{opt}} - y\|_2^2 + 2\langle \xi, \mathbb{X}(\hat{w} - w^*) \rangle - \|\xi\|_2^2 \\
&= \|\mathbb{X}(w_{\text{opt}} - w^*)\|_2^2 + 2\langle \xi, \mathbb{X}(\hat{w} - w^*) \rangle - 2\langle \xi, \mathbb{X}(w_{\text{opt}} - w^*) \rangle \\
&\le \|\mathbb{X}(w_{\text{opt}} - w^*)\|_2^2 + 2 \left( \|\mathbb{X}(\hat{w} - w^*)\|_2 + \|\mathbb{X}(w_{\text{opt}} - w^*)\|_2 \right) \sigma \sqrt{n_{\text{eff}}}.
\end{aligned}$$

Subtracting $\|\mathbb{X}(w_{\text{opt}} - w^*)\|_2^2$ from both sides and dividing by $\|\mathbb{X}(\hat{w} - w^*)\|_2 + \|\mathbb{X}(w_{\text{opt}} - w^*)\|_2$, we get that

$$\|\mathbb{X}(\hat{w} - w^*)\|_2 - \|\mathbb{X}(w_{\text{opt}} - w^*)\|_2 \le 2\sigma \sqrt{n_{\text{eff}}}.$$

It follows that

$$\begin{aligned}
\|\hat{w} - w^*\|_\Sigma &\le \sqrt{\frac{1}{(1-\epsilon)m}} \|\mathbb{X}(\hat{w} - w^*)\|_2 \\
&\le \sqrt{\frac{1}{(1-\epsilon)m}} \|\mathbb{X}(w_{\text{opt}} - w^*)\|_2 + 2\sigma \sqrt{\frac{(1+\epsilon)n_{\text{eff}}}{m}} \\
&\le \sqrt{\frac{1+\epsilon}{1-\epsilon}} \|w_{\text{opt}} - w^*\|_\Sigma + 2\sigma \sqrt{\frac{2n_{\text{eff}}}{m}}
\end{aligned}$$

as desired. $\qquad \square$

**Lemma F.2.** *Let $n, m \in \mathbb{N}$ and let $\Sigma$ be a positive semi-definite matrix. Fix a vector $w^* \in \mathbb{R}^n$ and a finite set $\mathcal{W} \subseteq \mathbb{R}^n$ and let $(X_i, y_i)_{i=1}^m$ be independent draws $X_i \sim N(0, \Sigma)$ and $y_i = \langle X_i, w^* \rangle + \xi_i$ where $\xi_i \sim N(0, \sigma^2)$. Pick*

$$\hat{w} \in \operatorname*{argmin}_{w \in \mathcal{W}} \|\mathbb{X}w - y\|_2^2.$$

*For any $\epsilon, \delta \in (0, 1)$, if $m \ge 8\epsilon^{-2} \log(2|\mathcal{W}|/\delta)$, then with probability at least $1 - 2\delta$, we have*

$$\|\hat{w} - w^*\|_\Sigma \le \sqrt{\frac{1+\epsilon}{1-\epsilon}} \inf_{w \in \mathcal{W}} \|w - w^*\|_\Sigma + 4\sigma \sqrt{\frac{\log(2|\mathcal{W}|/\delta)}{m}}.$$

*Proof.* For any fixed $w \in \mathcal{W}$, the random variables $\langle X_i, w - w^* \rangle \sim N(0, \|w - w^*\|_\Sigma^2)$ are independent, and therefore $\|\mathbb{X}(w - w^*)\|_2^2 \sim \|w - w^*\|_\Sigma^2 \chi_m^2$. It follows that for any $\epsilon > 0$,

$$\Pr \left[ \left| \frac{1}{m} \|\mathbb{X}(w - w^*)\|_2^2 - \|w - w^*\|_\Sigma^2 \right| > \epsilon \|w - w^*\|_\Sigma^2 \right] \le 2e^{-m\epsilon^2/8}.$$

By the union bound, if $m \ge 8\epsilon^{-2} \log(2|\mathcal{W}|/\delta)$, then with probability at least $1 - \delta$ it holds that for all $w \in \mathcal{W}$,

$$\left| \frac{1}{m} \|\mathbb{X}(w - w^*)\|_2^2 - \|w - w^*\|_\Sigma^2 \right| \le \epsilon \|w - w^*\|_\Sigma^2. \tag{10}$$

Also, for any fixed $w \in \mathcal{W}$, conditioned on $\mathbb{X}$, the random variable $\langle \xi, \frac{\mathbb{X}(w-w^*)}{\|\mathbb{X}(w-w^*)\|_2} \rangle$ has distribution $N(0, \sigma^2)$. Thus, by a Gaussian tail bound and the union bound, we have for any $t > 0$ that

$$\Pr \left[ \max_{w \in \mathcal{W}} \left| \left\langle \xi, \frac{\mathbb{X}(w - w^*)}{\|\mathbb{X}(w - w^*)\|} \right\rangle \right| \ge \sigma t \right] \le 2|\mathcal{W}| \cdot e^{-t^2/2}.$$

In particular, with probability at least $1 - \delta$ it holds that

$$\max_{w \in \mathcal{W}} \left| \left\langle \xi, \frac{\mathbb{X}(w - w^*)}{\|\mathbb{X}(w - w^*)\|} \right\rangle \right| \leq \sigma \sqrt{2 \log(2|\mathcal{W}|/\delta)}. \tag{11}$$

Using (10) and (11) we apply Lemma F.1 which gives the desired bound. $\qquad\square$

### F.2 Weak learning

**Lemma F.3.** *Let $n, m \in \mathbb{N}$ and $\epsilon, \delta > 0$. Let $\Sigma : n \times n$ be a positive semi-definite matrix and let $\mathbb{X} : m \times n$ have independent rows $X_1, \ldots, X_m \sim N(0, \Sigma)$. For any fixed $u, v \in \mathbb{R}^n$, if $m \geq 8\epsilon^{-2} \log(8/\delta)$, then it holds with probability at least $1 - \delta$ that*

$$\left| u^\top \left( \frac{1}{m} \mathbb{X}^\top \mathbb{X} - \Sigma \right) v \right| \leq 2\epsilon \|u\|_\Sigma \|v\|_\Sigma.$$

*Proof.* Decompose $u = av + w$ where $\langle v, w \rangle_\Sigma = 0$, so that $a = \langle u, v \rangle_\Sigma / \|v\|_\Sigma^2$. Since $\|\mathbb{X}v\|_2^2 \sim \|v\|_\Sigma^2 \chi_m^2$ and $m \geq 8\epsilon^{-2} \log(4/\delta)$ it holds with probability at least $1 - \delta/2$ that

$$\left| v^\top \left( \frac{1}{m} \mathbb{X}^\top \mathbb{X} - \Sigma \right) v \right| = \left| \frac{1}{m} \sum_{i=1}^m \langle X_i, v \rangle^2 - \|v\|_\Sigma^2 \right| \leq \epsilon \|v\|_\Sigma^2.$$

Next,

$$\left| w^\top \left( \frac{1}{m} \mathbb{X}^\top \mathbb{X} - \Sigma \right) v \right| = \left| \frac{1}{m} \sum_{i=1}^m \langle X_i, w \rangle \langle X_i, v \rangle \right| = \left| \frac{1}{m} \sum_{i=1}^m \langle Z_i, \Sigma^{1/2} w \rangle \langle Z_i, \Sigma^{1/2} v \rangle \right|$$

where we define independent random vectors $Z_1, \ldots, Z_m \sim N(0, I_n)$ so that $X_i = \Sigma^{1/2} Z_i$. Since $m \geq 8 \log(2/\delta)$, with probability at least $1 - \delta/4$ we have $\sum_{i=1}^m \langle Z_i, \Sigma^{1/2} v \rangle^2 \leq 2m \|v\|_\Sigma^2$. Condition on the value of this sum, and note that since $\Sigma^{1/2} v \perp \Sigma^{1/2} w$, the random variables $\langle Z_i, \Sigma^{1/2} w \rangle$ are still (independent and) distributed as $N(0, \|w\|_\Sigma^2)$. Thus

$$\frac{1}{m} \sum_{i=1}^m \langle Z_i, \Sigma^{1/2} w \rangle \langle Z_i, \Sigma^{1/2} v \rangle \sim N \left( 0, \frac{1}{m^2} \sum_{i=1}^m \|w\|_\Sigma^2 \langle Z_i, \Sigma^{1/2} v \rangle^2 \right).$$

When the variance is at most $2 \|w\|_\Sigma^2 \|v\|_\Sigma^2 / m$, we have with probability at least $1 - \delta/4$ that the sum is at most $2 \|w\|_\Sigma \|v\|_\Sigma \sqrt{2 \log(8/\delta)/m}$ in magnitude. So, using $m \geq 8\epsilon^{-2} \log(8/\delta)$ it holds unconditionally with probability at least $1 - \delta/2$ that

$$\left| \frac{1}{m} \sum_{i=1}^m \langle Z_i, \Sigma^{1/2} w \rangle \langle Z_i, \Sigma^{1/2} v \rangle \right| \leq \epsilon \|w\|_\Sigma \|v\|_\Sigma.$$

In all, we have that

$$\left| u^\top \left( \frac{1}{m} \mathbb{X}^\top \mathbb{X} - \Sigma \right) v \right| \leq |a| \epsilon \|v\|_\Sigma^2 + \epsilon \|w\|_\Sigma \|v\|_\Sigma \leq 2\epsilon \|u\|_\Sigma \|v\|_\Sigma$$

using that $|a| \leq \|u\|_\Sigma / \|v\|_\Sigma$ and $\|w\|_\Sigma \leq \|u\|_\Sigma$. $\qquad\square$

**Lemma F.4.** *Let $n, m \in \mathbb{N}$ and let $\Sigma$ be a positive semi-definite matrix. Fix a vector $w^* \in \mathbb{R}^n$ and a finite set $\mathcal{W} \subseteq \mathbb{R}^n$ and let $(X_i, y_i)_{i=1}^m$ be independent draws $X_i \sim N(0, \Sigma)$ and $y_i = \langle X_i, w^* \rangle + \xi_i$ where $\xi_i \sim N(0, \sigma^2)$. Pick*

$$(\hat{w}, \hat{\beta}) \in \operatorname*{argmin}_{\substack{w \in \mathcal{W} \\ \beta \in \mathbb{R}}} \|\beta \mathbb{X} w - y\|_2^2.$$

*Suppose $\alpha := \max_{w \in \mathcal{W}} \frac{\langle w, w^* \rangle_\Sigma}{\|w\|_\Sigma \|w^*\|_\Sigma} > 0$. For any $\delta > 0$, if $m \geq C\alpha^{-2} \log(32|\mathcal{W}|/\delta)$ for a sufficiently large absolute constant $C$, then with probability at least $1 - \delta$,*

$$\left\| \hat{\beta} \hat{w} - w^* \right\|_\Sigma^2 \leq (1 - \alpha^2/4) \|w^*\|_\Sigma^2 + \frac{400 \sigma^2 \log(4|\mathcal{W}|/\delta)}{\alpha^2 m}.$$

*Proof.* For any vectors $u, v \in \mathbb{R}^n$, define $\Delta(u, v) = u^\top \left( \frac{1}{m} \mathbb{X}^\top \mathbb{X} - \Sigma \right) v$.

**Claim F.5.** *With probability at least $1 - \delta$, the following bounds hold uniformly over $w \in \mathcal{W}$ and $\beta \in \mathbb{R}$:*

1. $\left| \left\langle \xi, \frac{\mathbb{X}(\beta w - w^*)}{\|\mathbb{X}(\beta w - w^*)\|_2} \right\rangle \right| \leq \sigma \sqrt{n_{\text{eff}}}$ *where* $n_{\text{eff}} := 2 \log(32|\mathcal{W}|/\delta)$.

2. $|\Delta(\beta w, w^*)| \leq \frac{\alpha}{100} \|\beta w\|_\Sigma \|w^*\|_\Sigma$

3. $|\Delta(\beta w, \beta w)| \leq \frac{\alpha}{100} \|\beta w\|_\Sigma^2$.

*Proof of claim.* For item (1), fix $w \in \mathcal{W}$. Let $\Phi^{(w)} : 2 \times m$ be a matrix whose rows form an orthonormal basis for $\text{span}\{\mathbb{X}w, \mathbb{X}w^*\} \subseteq \mathbb{R}^m$. Then (denoting the unit Euclidean ball in $\mathbb{R}^2$ by $B_2$) we have for all $\beta \in \mathbb{R}$ that

$$\left| \left\langle \xi, \frac{\mathbb{X}(\beta w - w^*)}{\|\mathbb{X}(\beta w - w^*)\|_2} \right\rangle \right| \leq \sup_{u \in B_2} \left| \left\langle \xi, (\Phi^{(w)})^\top u \right\rangle \right| \leq \left\| \Phi^{(w)} \xi \right\|_2 \leq \sqrt{2} \max_{i \in [2]} |\langle \Phi_i^{(w)}, \xi \rangle|.$$

Since $\langle \Phi_i^{(w)}, \xi \rangle \sim N(0, \sigma^2)$, we have $\Pr[|\langle \Phi_i^{(w)}, \xi \rangle| > \sigma \sqrt{2 \log(4|\mathcal{W}|/\delta)}] \leq \delta/(4|\mathcal{W}|)$. A union bound over $i \in [2]$ and $w \in \mathcal{W}$ gives that condition (2) in Lemma F.1 is satisfied with probability at least $1 - \delta/2$.

For items (2) and (3), note that $\Delta$ is bilinear, so it suffices to take $\beta = 1$. Applying Lemma F.3 and the union bound, so long as $m \geq C\alpha^{-2} \log(32|\mathcal{W}|/\delta)$ for a sufficiently large constant $C$, items (2) and (3) hold simultaneously with probability at least $1 - \delta/2$. $\qquad\square$

Henceforth we assume that all of the events in the above claim hold. Let $w_0 \in \mathcal{W}$ be such that $|\langle w_0, w^* \rangle_\Sigma| = \alpha \|w_0\|_\Sigma \|w^*\|_\Sigma$. Let $\beta_0 = \langle w_0, w^* \rangle_\Sigma / \|w_0\|_\Sigma^2$. Then

$$\|\beta_0 w_0 - w^*\|_\Sigma^2 = (1 - \alpha^2) \|w^*\|_\Sigma^2.$$

**Claim F.6.** *The excess empirical risk can be bounded as*

$$\left\| \mathbb{X}(\hat{\beta}\hat{w} - w^*) \right\|_2 \leq \|\mathbb{X}(w_0 - w^*)\|_2 + 2\sigma\sqrt{n_{\text{eff}}}.$$

*Proof of claim.* We have

$$\left\| \mathbb{X}(\hat{\beta}\hat{w} - w^*) \right\|_2^2 = \left\| \mathbb{X}\hat{\beta}\hat{w} - y \right\|_2^2 + 2\langle \xi, \mathbb{X}(\hat{\beta}\hat{w} - w^*) \rangle - \|\xi\|_2^2$$

$$\leq \|\mathbb{X}\beta_0 w_0 - y\|_2^2 + 2\langle \xi, \mathbb{X}(\hat{\beta}\hat{w} - w^*) \rangle - \|\xi\|_2^2$$

$$= \|\mathbb{X}(\beta_0 w_0 - w^*)\|_2^2 + 2\langle \xi, \mathbb{X}(\hat{\beta}\hat{w} - w^*) \rangle - 2\langle \xi, \mathbb{X}(\beta_0 w_0 - w^*) \rangle$$

$$\leq \|\mathbb{X}(\beta_0 w_0 - w^*)\|_2^2 + 2 \left( \left\| \mathbb{X}(\hat{\beta}\hat{w} - w^*) \right\|_2 + \|\mathbb{X}(\beta_0 w_0 - w^*)\|_2 \right) \sigma\sqrt{n_{\text{eff}}}$$

where the last bound is by item (1) of Claim F.5. Simplifying, we get the claimed bound. $\qquad\square$

Now we have

$$\left\| \hat{\beta}\hat{w} - w^* \right\|_\Sigma^2 = \frac{1}{m} \left\| \mathbb{X}(\hat{\beta}\hat{w} - w^*) \right\|_2^2 - \Delta(\hat{\beta}\hat{w} - w^*, \hat{\beta}\hat{w} - w^*)$$

$$\leq \frac{1}{m} \left( \|\mathbb{X}(\beta_0 w_0 - w^*)\|_2 + 2\sigma\sqrt{n_{\text{eff}}} \right)^2 - \Delta(\hat{\beta}\hat{w} - w^*, \hat{\beta}\hat{w} - w^*)$$

$$\leq \frac{1 + \alpha^2/100}{m} \|\mathbb{X}(\beta_0 w_0 - w^*)\|_2^2 + (1 + 100\alpha^{-2}) \frac{\sigma^2 n_{\text{eff}}}{m} - \Delta(\hat{\beta}\hat{w} - w^*, \hat{\beta}\hat{w} - w^*)$$

$$= (1 + \alpha^2/100) \|\beta_0 w_0 - w^*\|_\Sigma^2 + (1 + 100\alpha^{-2}) \frac{\sigma^2 n_{\text{eff}}}{m}$$

$$- \Delta(\hat{\beta}\hat{w} - w^*, \hat{\beta}\hat{w} - w^*) + \left( 1 + \frac{\alpha^2}{100} \right) \Delta(\beta_0 w_0 - w^*, \beta_0 w_0 - w^*)$$

$$\leq (1 + \alpha^2/100) \left\| \beta_0 w_0 - w^* \right\|_\Sigma^2 + (1 + 100\alpha^{-2}) \frac{\sigma^2 n_{\text{eff}}}{m}$$
$$+ |\Delta(\hat{\beta}\hat{w}, \hat{\beta}\hat{w})| + 2|\Delta(\hat{\beta}\hat{w}, w^*)|$$
$$+ (1 + \alpha^2/100)|\Delta(\beta_0 w_0, \beta_0 w_0)| + 2(1 + \alpha^2/100)|\Delta(\beta_0 w_0, w^*)|$$
$$+ (\alpha^2/100)|\Delta(w^*, w^*)|.$$

where the first inequality is by Claim F.6, the second inequality is by AM-GM, and the final inequality is expanding out the terms $\Delta(\hat{\beta}\hat{w} - w^*, \hat{\beta}\hat{w} - w^*)$ and $\Delta(\beta_0 w_0 - w^*, \beta_0 w_0 - w^*)$ (via bilinearity) and cancelling out the common term $\Delta(w^*, w^*)$. Finally applying items (2) and (3) of Claim F.5, we get

$$\left\| \hat{\beta}\hat{w} - w^* \right\|_\Sigma^2 \leq (1 + \alpha^2/100) \left\| \beta_0 w_0 - w^* \right\|_\Sigma^2 + (1 + 100\alpha^{-2}) \frac{\sigma^2 n_{\text{eff}}}{m}$$
$$+ \frac{\alpha}{100} \left\| \hat{\beta}\hat{w} \right\|_\Sigma^2 + \frac{\alpha}{50} \left\| \hat{\beta}\hat{w} \right\|_\Sigma \|w^*\|_\Sigma$$
$$+ \frac{\alpha}{50} \|\beta_0 w_0\|_\Sigma^2 + \frac{\alpha}{25} \|\beta_0 w_0\|_\Sigma \|w^*\|_\Sigma + \frac{\alpha^3}{100} \|w^*\|_\Sigma^2$$
$$\leq (1 - 9\alpha^2/10) \|w^*\|_\Sigma^2 + \frac{101\sigma^2 n_{\text{eff}}}{\alpha^2 m}$$
$$+ \frac{\alpha}{100} \left\| \hat{\beta}\hat{w} \right\|_\Sigma^2 + \frac{\alpha}{50} \left\| \hat{\beta}\hat{w} \right\|_\Sigma \|w^*\|_\Sigma \tag{12}$$

where the second inequality uses the bounds $\|\beta_0 w_0 - w^*\|_\Sigma^2 = (1 - \alpha^2) \|w^*\|_\Sigma^2$ and

$$\|\beta_0 w_0\|_\Sigma = \frac{|\langle w_0, w^* \rangle_\Sigma|}{\|w_0\|_\Sigma} = \alpha \|w^*\|_\Sigma.$$

But now on the other hand,

$$\left\| \hat{\beta}\hat{w} - w^* \right\|_\Sigma^2 = \left\| \hat{\beta}\hat{w} \right\|_\Sigma^2 + \|w^*\|_\Sigma^2 - 2\langle \hat{\beta}\hat{w}, w^* \rangle_\Sigma \geq \left\| \hat{\beta}\hat{w} \right\|_\Sigma^2 + \|w^*\|_\Sigma^2 + 2\alpha \left\| \hat{\beta}\hat{w} \right\|_\Sigma \|w^*\|_\Sigma.$$

Comparing with (12) gives

$$\left(1 - \frac{\alpha}{100}\right) \left\| \hat{\beta}\hat{w} \right\|_\Sigma^2 \leq \frac{101\sigma^2 n_{\text{eff}}}{\alpha^2 m} + 3\alpha \left\| \hat{\beta}\hat{w} \right\|_\Sigma \|w^*\|_\Sigma$$

and therefore

$$\left\| \hat{\beta}\hat{w} \right\|_\Sigma \leq 4\alpha \|w^*\|_\Sigma + \sigma \sqrt{\frac{101 n_{\text{eff}}}{\alpha^2 m}}.$$

Substituting into (12) we finally get

$$\left\| \hat{\beta}\hat{w} - w^* \right\|_\Sigma^2 \leq (1 - \alpha^2/2) \|w^*\|_\Sigma^2 + \frac{200\sigma^2 n_{\text{eff}}}{\alpha^2 m}$$

as desired. $\qquad \square$

### F.3 Excess risk at optima of additively-regularized programs

**Lemma F.7.** *Let $n \in \mathbb{N}$, and let $\Sigma : n \times n$ be a positive semi-definite matrix. For some seminorm $\Phi : \mathbb{R}^n \to [0, \infty)$ and some $p, \delta > 0$, assume that with probability at least $1 - \delta$ over $G \sim N(0, \Sigma)$ it holds uniformly over $v \in \mathbb{R}^n$ that*

$$\langle v, G \rangle \leq \frac{1}{2}\Phi(v) + \sqrt{p} \|v\|_\Sigma.$$

*Fix a vector $v^* \in \mathbb{R}^n$. For any $m \in \mathbb{N}$ and $\sigma > 0$ let $(X_i, y_i)_{i=1}^m$ be independent samples distributed as $X_i \sim N(0, \Sigma)$ and $y_i = \langle X_i, v^* \rangle + \xi_i$ where $\xi_i \sim N(0, \sigma^2)$. Define*

$$\hat{v} \in \operatorname*{argmin}_{v \in \mathbb{R}^n} \|\mathbb{X}v - y\|_2^2 + \Phi(v)^2 + \|y\|_2 \Phi(v)$$

*where $\mathbb{X} : m \times n$ is the matrix with rows $X_1, \ldots, X_m$. Then with probability at least $1 - 7\delta$ over $(X_i, y_i)_{i=1}^m$, so long as $m \geq 16p + 196 \log(12/\delta))$, it holds that*

$$\|\hat{v} - v^*\|_\Sigma^2 \leq \frac{128\sigma^2 p}{m} + \frac{8(\sigma + \|v^*\|_\Sigma)\Phi(v^*)}{\sqrt{m}} + \frac{8\Phi(w^*)^2}{m}.$$

*Proof.* For notational convenience, define $F(v) := (1/2)\Phi(v - v^*) + \sqrt{p}\,\|v - v^*\|_\Sigma$. We apply the lemma's assumption twice:

- For any fixed $\xi$, the random variable $\mathbb{X}\xi$ has distribution $N(0, \|\xi\|_2^2\,\Sigma)$. By the above claim, with probability at least $1 - \delta$ over $\mathbb{X}$, we have $\langle \xi, \mathbb{X}(v - v^*)\rangle \leq \|\xi\|_2\,F(v)$ uniformly in $v \in \mathbb{R}^n$.

  Since $\|\xi\|_2^2 \sim \sigma^2\chi_m^2$ and $m \geq 8\log(2/\delta)$, it holds with probability at least $1 - \delta$ that $\frac{1}{\sqrt{m}}\|\xi\|_2 \leq \sqrt{2}\sigma$. Thus, with probability at least $1 - 2\delta$, we have

  $$\langle \xi, \mathbb{X}(v - v^*)\rangle \leq \sqrt{2m}\sigma F(v) \tag{13}$$

  uniformly in $v \in \mathbb{R}^n$.

- The assumption means that we can apply Theorem C.1 with (noiseless) samples $(X_i, \langle X_i, v^*\rangle)_{i=1}^m$ to get the following: since $m \geq 196\log(12/\delta)$, it holds with probability at least $1 - 4\delta$ over the randomness of $\mathbb{X}$ that for all $v \in \mathbb{R}^n$,

  $$\|v - v^*\|_\Sigma^2 \leq \frac{2}{m}\|\mathbb{X}(v - v^*)\|_2^2 + \frac{2}{m}F(v)^2. \tag{14}$$

We also observe that the entries of $y$ are independent and identically distributed as $N(0, \|v^*\|_\Sigma^2 + \sigma^2)$, so by a $\chi^2$ tail bound, since $m \geq 32\log(2/\delta)$, it holds with probability at least $1 - \delta$ that

$$\frac{1}{m}\|y\|_2^2 \in \left[\frac{1}{2}(\|v^*\|_\Sigma^2 + \sigma^2), \frac{3}{2}(\|v^*\|_\Sigma^2 + \sigma^2)\right]. \tag{15}$$

We now condition on the event (which occurs with probability at least $1 - 7\delta$) that the bounds (13), (14), and (15) all hold. Specifying (14) to $v := \hat{v}$, we get that

$$\frac{m}{2}\|\hat{v} - v^*\|_\Sigma^2$$
$$\leq \|\mathbb{X}(\hat{v} - v^*)\|_2^2 + F(\hat{v})^2$$
$$\leq \|\mathbb{X}(\hat{v} - v^*)\|_2^2 - \|\mathbb{X}\hat{v} - y\|_2^2 - \Phi(\hat{v})^2 - \|y\|_2\,\Phi(\hat{v})$$
$$\qquad + \|\mathbb{X}v^* - y\|_2^2 + \Phi(v^*)^2 + \|y\|_2\,\Phi(v^*) + F(\hat{v})^2$$
$$= 2\langle \mathbb{X}v^* - y, \mathbb{X}(\hat{v} - v^*)\rangle$$
$$\qquad - \Phi(\hat{v})^2 - \|y\|_2\,\Phi(\hat{v}) + \Phi(v^*)^2 + \|y\|_2\,\Phi(v^*) + F(\hat{v})^2$$
$$\leq \sqrt{2m}\sigma F(\hat{v}) - \Phi(\hat{v})^2 - \|y\|_2\,\Phi(\hat{v}) + \Phi(v^*)^2 + \|y\|_2\,\Phi(v^*) + F(\hat{v})^2$$

where the first inequality is by (14), the second inequality is by optimality of $\hat{v}$, and the third inequality is by (13). We now expand $F(\hat{v})$ in the above expression. If $\sqrt{2mp}\sigma\|\hat{v} - v^*\|_\Sigma$ exceeds $\frac{m}{8}\|\hat{v} - v^*\|_\Sigma^2$ then the lemma immediately holds since

$$\|\hat{v} - v^*\|_\Sigma^2 \leq \frac{128\sigma^2 p}{m}.$$

So we may assume that in fact $\sqrt{2mp}\sigma\|\hat{v} - v^*\|_\Sigma \leq \frac{m}{8}\|\hat{v} - v^*\|_\Sigma^2$. By the lemma assumptions, we also know that $m \geq 16p$. Thus, expanding $F(\hat{v})$ and applying these bounds,

$$\frac{m}{2}\|\hat{v} - v^*\|_\Sigma^2 \leq \sqrt{2m}\sigma\left(\frac{1}{2}\Phi(\hat{v} - v^*) + \sqrt{p}\,\|\hat{v} - v^*\|_\Sigma\right)$$
$$\qquad - \Phi(\hat{v})^2 - \|y\|_2\,\Phi(\hat{v}) + \Phi(v^*)^2 + \|y\|_2\,\Phi(v^*)$$
$$\qquad + \frac{1}{2}\Phi(\hat{v} - v^*)^2 + 2p\,\|\hat{v} - v^*\|_\Sigma^2$$
$$\leq \sqrt{\frac{m}{2}}\sigma\Phi(\hat{v} - v^*) + \frac{m}{8}\|\hat{v} - v^*\|_\Sigma^2$$
$$\qquad - \Phi(\hat{v})^2 - \|y\|_2\,\Phi(\hat{v}) + \Phi(v^*)^2 + \|y\|_2\,\Phi(v^*)$$

$$+ \frac{1}{2}\Phi(\hat{v} - v^*)^2 + \frac{m}{8}\|\hat{v} - v^*\|_\Sigma^2.$$

Simplifying, applying the triangle inequality $\Phi(\hat{v} - v^*) \leq \Phi(\hat{v}) + \Phi(v^*)$, and grouping terms, we get

$$\frac{m}{4}\|\hat{v} - v^*\|_\Gamma^2$$

$$\leq \left(\sqrt{\frac{m}{2}}\sigma - \|y\|_2\right)\Phi(\hat{v}) + \left(\sqrt{\frac{m}{2}}\sigma + \|y\|_2\right)\Phi(v^*) + 2\Phi(v^*)^2$$

$$\leq 2(\sigma + \|v^*\|_\Sigma)\sqrt{m}\Phi(v^*) + 2\Phi(v^*)^2$$

where the last inequality uses both sides of the bound (15). $\qquad\square$

# G   Covering bounds from classical assumptions

In this section, we further motivate the definition of our covering number $\mathcal{N}_{t,\alpha}(\Sigma)$ by showing that in all settings where efficient SLR algorithms are known, there is a straightforward *linear* upper bound on the covering number. This lends weight to the need for stronger upper bounds on $\mathcal{N}_{t,\alpha}$ as a stepping stone towards more efficient algorithms for sparse linear regression.

## G.1   Compatibility condition

**Definition G.1** (Compatibility Condition, see e.g. [40]). For a positive semidefinite matrix $\Sigma : n \times n$, $L \geq 1$, and set $S \subset [n]$, we say $\Sigma$ has *S-restricted $\ell_1$-eigenvalue*

$$\phi^2(\Sigma, S) = \min_{w \in \mathcal{C}(S)} \frac{|S| \cdot \langle w, \Sigma w \rangle}{\|w_S\|_1^2}$$

where the cone $\mathcal{C}(S)$ is defined as

$$\mathcal{C}(S) = \{w \neq 0 : \|w_{S^C}\|_1 \leq L\|w_S\|_1\}.$$

For $t \in \mathbb{N}$, the *t-restricted $\ell_1$-eigenvalue* $\phi^2(\Sigma, t)$ is the minimum over all $S$ of size at most $t$.

It is well-known that an upper bound on $\frac{\max_i \Sigma_{ii}}{\phi^2(\Sigma, t)}$ is sufficient for the success of Lasso (as well as nearly necessary; see e.g. the Weak Compatibility Condition defined in [23]):

**Theorem G.2** (see e.g. Corollary 5 in [45]). *Fix $n, m, t \in \mathbb{N}$, $\sigma, \delta > 0$, and a positive semi-definite matrix $\Sigma : n \times n$ with $\max_i \Sigma_{ii} \leq 1$. Fix a $t$-sparse vector $v^* \in \mathbb{R}^n$ and let $(X_i, y_i)_{i=1}^m$ be independent samples distributed as $X_i \sim N(0, \Sigma)$ and $y_i = \langle X_i, v^* \rangle + \xi_i$ where $\xi_i \sim N(0, \sigma^2)$. Define*

$$\hat{v} \in \operatorname*{argmin}_{v \in \mathbb{R}^n : \|v\|_1 \leq \|v^*\|_1} \|\mathbb{X}v - y\|_2^2$$

*where $\mathbb{X} : m \times n$ is the matrix with rows $X_1, \ldots, X_m$. If $m \geq 4\phi^2(\Sigma, t) \cdot t \log(16n/\delta)$, then with probability at least $1 - \delta$, it holds that*

$$\|\hat{v} - v^*\|_\Sigma^2 \leq O\left(\frac{\sigma^2 t \log(16n/\delta)}{\phi^2(\Sigma, t)m}\right).$$

**Fact G.3.** *Let $n, t \in \mathbb{N}$. For any positive semi-definite $\Sigma : n \times n$ with $\phi^2 := \phi^2(\Sigma, t)$ and $\max_i \Sigma_{ii} \leq 1$, it holds that $\mathcal{N}_{t, \phi/\sqrt{t}}(\Sigma) \leq n$.*

*Proof.* The proof is essentially the same as that of Fact A.4. By Lemma A.3, it suffices to show that the standard basis is a $(t, \sqrt{t}/\phi)$-$\ell_1$-representation for $\Sigma$. Indeed, for any $t$-sparse $v \in \mathbb{R}^n$, we have

$$\sum_{i=1}^n |v_i| \cdot \|e_i\|_\Sigma \leq \|v\|_1 \cdot \max_i \sqrt{\Sigma_{ii}} \leq \frac{\sqrt{t}\|v\|_\Sigma}{\phi}$$

as claimed. $\qquad\square$

## G.2 Submodularity ratio

**Definition G.4** (see e.g. [9]). For a positive semi-definite matrix $\Sigma : n \times n$ and a set $L \subseteq [n]$ define the normalized residual covariance matrx $\Sigma^{(L)} : n \times n$ by

$$\Sigma^{(L)} := (D^{1/2})^{\dagger} \left( \Sigma - \Sigma_L^{\top} \Sigma_{LL}^{\dagger} \Sigma_L \right) (D^{1/2})^{\dagger}$$

where $D := \mathrm{diag} \left( \Sigma - \Sigma_L^{\top} \Sigma_{LL}^{\dagger} \Sigma_L \right)$.

**Definition G.5.** Fix a positive semi-definite matrix $\Sigma : n \times n$, a positive integer $t \in \mathbb{N}$, and any $v^* \in \mathbb{R}^n$. Define the *$t$-submodularity ratio* of $\Sigma$ with respect to $v^*$ by

$$\gamma_t(\Sigma, v^*) := \min_{L, S \subseteq [n] : |L|, |S| \leq t, L \cap S = \emptyset} \frac{(v^*)^{\top} (\Sigma^{(L)})_S^{\top} (\Sigma^{(L)})_S v^*}{(v^*)^{\top} (\Sigma^{(L)})_S^{\top} (\Sigma^{(L)})_{SS}^{\dagger} (\Sigma^{(L)})_S v^*}.$$

In any $t$-sparse linear regression model with true regressor $v^*$, when the above quantity $\gamma := \gamma_t(\Sigma, v^*)$ is bounded away from zero, it can be shown that the standard Forward Regression algorithm finds some $t$-sparse estimate $\hat{v} \in \mathbb{R}^n$ such that $\|\hat{v} - v^*\|_{\Sigma}^2 \leq e^{-\gamma} \|v^*\|_{\Sigma}^2$ (see e.g. Theorem 3.2 in [9]; that result is for the model where the algorithm is given exact access to $\langle v, v^* \rangle_{\Sigma}$ for any $t$-sparse $v \in \mathbb{R}^n$, but analogous finite-sample bounds can be obtained with $O(\gamma^{-O(1)} t \log(n))$ samples by applying the theorem to the empirical covariance matrix and using concentration of $t \times t$ submatrices). A similar guarantee is also known for Orthogonal Matching Pursuit (Theorem 3.7 in [9]).

Once again, it is simple to show that the standard basis is a good dictionary for matrices with a large submodularity ratio.

**Fact G.6.** *Let* $n, t \in \mathbb{N}$. *For any positive semi-definite* $\Sigma : n \times n$ *with* $\gamma := \min_{v^* \in \mathbb{R}^n \cap B_0(t)} \gamma_t(\Sigma, v^*)$, *it holds that* $\mathcal{N}_{t, \sqrt{\gamma/t}}(\Sigma) \leq n$.

*Proof.* We show that the standard basis is a $(t, \gamma/t)$-dictionary for $\Sigma$. Without loss of generality assume that $\Sigma_{ii} = 1$ for all $i \in [n]$. Then $\Sigma^{(\emptyset)} = \Sigma$. Fix any $t$-sparse $v^* \in \mathbb{R}^n$. Setting $S := \mathrm{supp}(v^*)$, we have that

$$\sum_{i \in S} \langle e_i, v^* \rangle_{\Sigma}^2 = (v^*)^{\top} \Sigma_S^{\top} \Sigma_S v^* \geq \gamma (v^*)^{\top} \Sigma_S^{\top} (\Sigma_{SS})^{\dagger} \Sigma_S v^* = \gamma \|v^*\|_{\Sigma}^2$$

where the inequality is by definition of $\gamma$, and the final equality uses that $\Sigma_S v^* = \Sigma_{SS}(v^*)_S$ (since $v^*$ is supported on $S$). It follows that $\max_{i \in S} \langle e_i, v^* \rangle_{\Sigma}^2 \geq (\gamma/t) \|v^*\|_{\Sigma}^2$. Since $\|e_i\|_{\Sigma} = 1$ for all $i$, we conclude that

$$\max_{i \in [n]} \frac{|\langle e_i, v^* \rangle_{\Sigma}|}{\|e_i\|_{\Sigma} \|v^*\|_{\Sigma}} \geq \sqrt{\frac{\gamma}{t}}$$

as claimed. $\qquad\square$

## G.3 Sparse preconditioning

Recent work [23] showed that if $\Sigma : n \times n$ is a positive definite matrix and the support of $\Theta := \Sigma^{-1}$ is the adjacency matrix of a graph with low *treewidth*, then there is a polynomial-time, sample-efficient algorithm for sparse linear regression with covariates drawn from $N(0, \Sigma)$. The key to this result was a proof that such covariance matrices are *sparsely preconditionable*: i.e., there is a matrix $S : n \times n$ such that $\Sigma = SS^{\top}$ and $S$ has sparse rows. We claim that this property also immediately enables succinct dictionaries.

Concretely, suppose that $S$ has $s$-sparse rows. By a change-of-basis argument, any $t$-sparse vector in the standard basis is $st$-sparse in the basis $\{(S^{\top})_1^{-1}, \ldots, (S^{\top})_n^{-1}\}$. Moreover these vectors are orthonormal under $\Sigma$. Thus, by the same argument as for Fact A.4, it's easy to see that $\{(S^{\top})_1^{-1}, \ldots, (S^{\top})_n^{-1}\}$ is a $(t, 1/\sqrt{st})$-dictionary for $\Sigma$.

# H  Supplementary figure

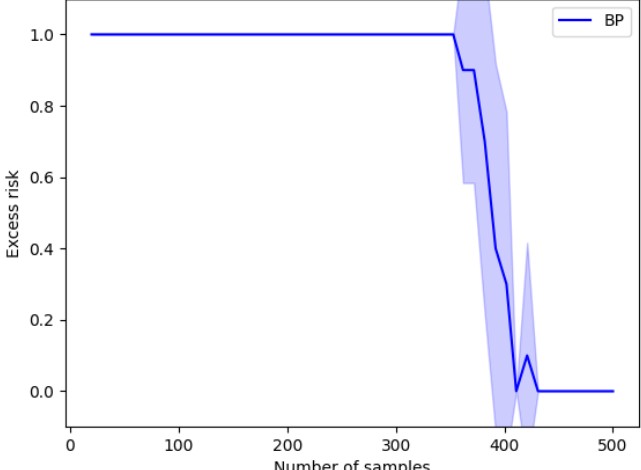

Figure 2: Performance of Basis Pursuit in a synthetic example with $n = 1000$ covariates. The co-variates $X_{1:1000}$ are all independent $N(0,1)$ except for $(X_0, X_1, X_2)$, which have joint distribution $X_0 = Z_0$, $X_1 = Z_0 + 0.4Z_1$, and $X_2 = Z_1 + 0.4Z_2$ where $Z_0, Z_1, Z_2 \sim N(0,1)$ are independent. The noiseless responses are $y = 6.25(X_1 - X_2) + 2.5X_3$, i.e. the ground truth is 3-sparse. The $x$-axis is the number of samples. The $y$-axis is the out-of-sample prediction error (averaged over 10 independent runs, and error bars indicate the standard deviation).

# I  Experimental details

The simulations were done using Python 3.9 and the Gurobi library [17]. Each figure took several minutes to generate using a standard laptop. See the file `auglasso.py` for code and execution instructions.

