# OpenReview forum: "Feature Adaptation for Sparse Linear Regression"
_NeurIPS.cc/2023/Conference — NeurIPS 2023 spotlight_

### Official Review · Reviewer_Ahsd · 2023-06-27

**Soundness:** 4 excellent
**Presentation:** 4 excellent
**Contribution:** 4 excellent
**Rating:** 8
**Confidence:** 4

**Summary:**

This work studies the problem of sparse linear regression under the statistical model where the examples are drawn as zero-mean Gaussians with covariance matrix $\Sigma$ and each response is a t-sparse linear combination of the examples plus i.i.d. Gaussian noise. While classical results establish that the LASSO can computationally efficiently recover a good solution v* with a nearly optimal number of samples when the covariance matrix $\Sigma$ is well-conditioned, guarantees beyond this simple setting are substantially lacking, with a few notable exceptions. In general, not much is known beyond a brute-force approach of trying all $\binom{n}{t}$ sparsity patterns, which requires $n^t$ time. This work shows that even if $\Sigma$ is ill-conditioned, if Sigma is well-conditioned after removing the top or bottom O(1) eigenvalues (i.e. a notion of “robust” well-condtionedness), then this running time can be improved to roughly $f(t) \cdot n^3$.

When there are a small number of tiny eigenvalues, the main problem is the existence of sparse linear dependencies among the variables. The algorithmic approach to address this problem is then to iteratively peel off a small number of these variables at a time (IterativePeeling()). This procedure gives a construction of a small dictionary of vectors such that any t-sparse vector can be written as a linear combination of this dictionary with coefficients bounded in L1 (Lemma 2.9), which in turn implies that the LASSO identifies a good solution v in the sense of bounded excess risk. The authors interpret the technique of applying LASSO to an augmented dictionary as “feature adaptation”.

**Strengths:**

* Sparse linear regression is a widely studied yet notoriously difficult problem, and this work identifies a very natural class of inputs for which positive results can be obtained.
* I find the discussion in Section 2.1 on the proposal of (t, alpha) dictionaries as a canonical abstraction of all existing approaches to sparse linear regression to be very interesting and valuable. It helps concentrate research efforts on this problem to this more structured approach, and I believe it may prove to be influential in following works on sparse linear regression.

**Weaknesses:**

* It seems like like there are no bounds on the sparsity of the approximate solution v that is outputted by this algorithm, probably due to the fact that the feature adaptation step augments the feature set with dense linear combinations of the existing variables (please let me know if I have misunderstood). Thus, the measure of the “sparsity” of this algorithm lies in the small sample complexity, rather than its ability to output a sparse solution. Thus, the guarantees of this algorithm are tightly linked to the statistical setting of sparse linear regression, and it may be difficult to adapt these techniques to the problem of outputting a good sparse solution to linear regression. However, I do find this limitation very interesting, and I wonder if there are gaps between the performance of these algorithms if one is required to output a sparse solution. Such separations are known in certain sparse recovery settings (see, e.g., https://arxiv.org/abs/1110.4414).

**Questions:**

n/a

**Limitations:**

The authors have provided adequate discussion of limitations

---

> ### Author Rebuttal · Authors · 2023-08-09
>
> We thank the reviewer for their time and comments, and for appreciating our techniques! The question about sparsity of the estimate is indeed interesting. In general, if we use a feature adaptation approach, some feature may be a dense combination of the original covariates and so we cannot guarantee sparsity in the original basis.
>
> Nevertheless, for our main result it should be possible to guarantee some level of sparsity in the original basis by using tools from high-dimensional geometry. This is because (ignoring the final boosting step) the adapted features used in Algorithm 1 are rescalings of the coordinate basis, and because the proof of the result establishes an $\ell_1$-norm bound on the predictor in the rescaled space outside of the set $S$ (which has bounded size) \--- see page 21 of the supplementary material. Given this, one can sparsify the predictor using the Approximate Caratheodory Theorem (Theorem 0.0.2 of Vershynin's book [41]) to ensure sparsity $|S|$ plus a polynomial in the $\ell_1$-norm bound and the reciprocal of the desired prediction error. It should also be possible to achieve a similar sparsity guarantee by replacing the LASSO with an orthogonal matching pursuit method; see theorem 15 and remark 5 of [24].
>
> It's true that the boosting step includes (one) denser feature at each iteration, but this feature corresponds to the predictor at the previous iteration, so it may be possible to iteratively bound the sparsities of the predictors at each iteration. Understanding the guarantees achieved here is an interesting open problem. Another is understanding whether a stronger sparsity guarantee (than that obtained by approximate Caratheodory) is possible in our setting.

---

### Official Review · Reviewer_BTho · 2023-06-27

**Soundness:** 3 good
**Presentation:** 4 excellent
**Contribution:** 4 excellent
**Rating:** 8
**Confidence:** 3

**Summary:**

The paper introduces an algorithm to solve sparse regression when the covariates are generated from a normal distribution with ill-conditioned covariance matrix, i.e. outlayer eigenvalues. The algorithm is based on feature augmentation where, meaning that the covariates are completed with well-chosen vectors, and is designed to provably achieve near optimal sample complexity in the studied framework.

**Strengths:**

Regarding presentation, the paper is clearly written and presented. Each theoretical result is explained and justified, which makes the paper easy to follow. Regarding the content, the idea of augmenting the features by taking the data distribution into account before solving the Lasso seems new. The paper provides interesting theoretical insights on this method, while maintaining a computational perspective, for instance when explaining the design of the dictionary.

**Weaknesses:**

While the main contributions of the paper are theoretical, it would have been interesting to get more experimental details on the algorithm. In particular, numerical illustrations of the behavior with respect to the conditioning of the covariance matrix (number of outlayers, gap between largest and smallest eigenvalues, ...), and of the robustness with respect to standard algorithms like Basis Pursuit (for instance in "phase transitions" between well and ill-conditioned scenarios), may broaden the audience and the impact of the contributions.

**Questions:**

The ideas of experiments below may help provide more information on the practical behavior of the algorithm and put it into perspective with the theoretical results.

* Could the authors provide the number of samples necessary to reach low risk with respect to the ratio $\frac{\lambda_{n - d_h}}{\lambda_{d_l+1}}$ and with respect to the number of outlayer eigenvalues when using their algorithm and when using BP ?

* Could the authors provide the computation time with respect to the number of samples in addition to the risk ?

* Could the authors provide the size of the set $S$ obtained in practice with Iterative Peeling, and compare it to the bounds given in Lemma 2.4 and 2.5, with respect to the sparsity $t$ ?

* Could the authors provide a comparison of the performance with and without the knowledge of $\Sigma$ (estimated from samples v. known distribution) ?


**Limitations:**

The limitations are discussed by the authors.

-----------

After reading the other reviews and the rebuttal, I increased my rating.

---

> ### Author Rebuttal · Authors · 2023-08-09
>
> We thank the reviewer for their time and comments. In particular thanks for the good suggestions for experimental directions to consider. It's tricky to understand which instances are the ``worst'' practical instances for the algorithm. However we can give partial answers to some of your questions:
>
> First, regarding the size of $S$. We're actually not aware of any instances where in practice the size of $S$ will exceed roughly $d \cdot t$ (number of outlier eigenvalues times sparsity). It would be quite interesting to understand if there is indeed a ``hard'' example for our algorithm or if there is a tighter analysis than what we were able to show.
>
> Second, regarding estimating $\Sigma$ using samples. Unfortunately our algorithm does not have a hope of succeeding when the estimated covariance is low-rank, because it needs an accurate estimate of the eigenspaces of $\Sigma$ --- thus, we need at least $\Omega(n)$ unlabelled samples (in addition to the $m$ labelled samples) to have any hope of success. Numerically, on our simple synthetic example (Figure 1) we do find that the algorithm works when we estimate $\Sigma$ using $2n$ samples.
>
> Third, regarding numerical runtime. Our algorithm has two parts. The first part is dentifying the set $S$. This has runtime that does not depend on the number of samples, but depends on the dimension due to requiring an eigendecomposition of $\Sigma$; for our example in Figure 1, this step took $0.65$ seconds. The second part of the algorithm is solving the adapted basis pursuit, which had runtime ranging from $0.12$ seconds (on average) when $m = 20$, to $1.8$ seconds when $m = 500$. The scaling was roughly linear in the number of samples. The runtime of the standard basis pursuit was very similar to the runtime of the second part of our algorithm.
>
> Finally, regarding the number of outlier eigenvalues. For our simple synthetic example, we observe that adding a second, independent sparse dependency does not double the sample complexity; in fact, it's essentially unchanged even with $10$ independent dependencies. This tracks with our theoretical understanding: the sample complexity is additive between (a) a component due to size of $S$, and (b) a component on the order of $t\log n$. In our setting when $d \leq 10$, it appears that the second component is dominant.

---

> > ### Comment · Reviewer_BTho · 2023-08-14
> > **Thanks for the answer**
> >
> > Thanks to the authors for their answer and insights on practical details. After reading the other reviews and the rebuttal, I will increase my rating.

---

### Official Review · Reviewer_8yrN · 2023-07-09

**Soundness:** 4 excellent
**Presentation:** 3 good
**Contribution:** 4 excellent
**Rating:** 7
**Confidence:** 3

**Summary:**

This paper presents an innovative polynomial-time algorithm for sparse linear regression in the correlated random design setting. The algorithm adapts the Lasso technique to effectively tolerate a limited number of approximate dependencies, resulting in both computational and statistical efficiency for covariance matrices with a few "outlier" eigenvalues. The proposed method is part of a more extensive framework of feature adaptation for sparse linear regression with ill-conditioned covariates and offers the first polynomial-factor improvement over brute-force search for constant sparsity and arbitrary covariance.

**Strengths:**

1. The paper contributes a novel algorithm for sparse linear regression, adeptly adapting the Lasso to accommodate a small number of approximate dependencies.

2. The proposed algorithm exhibits both computational and statistical efficiency for covariance matrices with a few "outlier" eigenvalues, providing a substantial advancement over existing methods in this context.

**Weaknesses:**

1. The paper assumes constant sparsity for simplicity, but it is essential to explore the impact of this assumption on the main results.

2. To more convincingly demonstrate the algorithm's computational and statistical efficiency, the paper would benefit from the inclusion of supplementary numerical simulations.


**Questions:**

1. The paper assumes constant sparsity $t$ for simplicity. It is unclear whether this assumption is necessary to establish the main results of the paper? In particular, it would be interesting to examine the case where the sparsity $t$ takes the order of $\log n$.

2. It is important to consider the more general case where $t$ is treated as a variable instead of a constant. I wonder whether the main results in the paper remain valid in this case.




**Limitations:**

Yes

---

> ### Author Rebuttal · Authors · 2023-08-09
>
> We thank the reviewer for their time and comments. In case it was a point of confusion, we'd like to emphasize that our results do apply when $t$ is a variable, i.e. in Theorems 1.1 and 1.2, there are no factors of $t$ ``hidden'' in any constants. So for example when $t = \log \log n$ our results still yield state-of-the-art sample efficiency / computational efficiency tradeoffs. It is indeed true that when $t = \Omega(\log n)$ our results become vacuous, and it's a very interesting direction for future research whether this limitation can be alleviated further.
>
> **Q:** *``To more convincingly demonstrate the algorithm's computational and statistical efficiency, the paper would benefit from the inclusion of supplementary numerical simulations.''* Certainly; this paper was primarily theoretical (with some simple numeric validation) but it would indeed be interesting to eventually apply to real data.

---

> > ### Comment · Reviewer_8yrN · 2023-08-14
> >
> > Thank you for your response.

---

### Official Review · Reviewer_EHBK · 2023-07-09

**Soundness:** 3 good
**Presentation:** 1 poor
**Contribution:** 2 fair
**Rating:** 5
**Confidence:** 4

**Summary:**

This paper studies the correlated random design setting, where the covariates are drawn from a multivariate Gaussian, and seeks an estimator with small excess risk. This work provides a polynomial-time algorithm that, given Σ, automatically adapts the Lasso to tolerate a small number of approximate dependencies, and achieves near-optimal sample complexity for constant sparsity and if Σ has few “outlier” eigenvalues.

**Strengths:**

Sparse linear regression is a fundamental problem in high-dimensional statistics.
This paper studies a polynomial-time algorithm that automatically adapts the Lasso to tolerate a small number of approximate dependencies. In theoretical analysis, this work achieves near-optimal sample complexity for constant sparsity. The proposed algorithm fits into a broader framework of feature adaptation for sparse linear regression with ill-conditioned covariates.

**Weaknesses:**

1.If this work can provide experimental verification of the superiority of the proposed algorithm, it will be more convincing. For example, compare with the related work to verify the performance of the proposed algorithm in terms of time complexity and accuracy.

2.In figure 1, when the number of samples is greater than 100, is the standard deviation of adapted BP algorithm zero?

3.The presentation of references is not standardized, such as:

[38] Sara Van De Geer. On tight bounds for the lasso. Journal of Machine Learning Research, 19:46, 2018.

[22] Jonathan Kelner, Frederic Koehler, Raghu Meka, and Ankur Moitra. Learning some popular gaussian graphical models without condition number bounds. In Proceedings of Neural Information Processing Systems (NeurIPS), 2020.

4.The organization and presentation of this paper can be further improved.

**Questions:**

See “Weaknesses”.

---

> ### Author Rebuttal · Authors · 2023-08-09
>
> We thank the reviewer for their time and comments. To address their questions:
>
> **Q:** *``compare with the related work to verify the performance of the proposed algorithm in terms of time complexity and accuracy.''* We would like to emphasize that *no* prior work has addressed the problem of sparse dependencies among covariates, aside from the very special case where the dependencies have sparsity $2$. We did provide experimental evidence (Figure 1) that our algorithm achieves superior accuracy to Lasso/Basis Pursuit, the algorithm that practitioners would typically try. We did not experimentally compare with e.g. the agglomerative clustering approach mentioned in our related work section, since (as discussed) it's clear from first principles that this approach cannot succeed in our general setting.
>
> **Q:** *``In figure 1, when the number of samples is greater than 100, is the standard deviation of adapted BP algorithm zero?''* Yes. For each sample size beyond $100$, in all ten trials our algorithm achieves zero prediction error. Note that this is reasonable since the samples are noiseless.
>
> **Q:** *``The organization and presentation of this paper can be further improved.''* If the reviewer has constructive and concrete suggestions for improvement of the organization/presentation, we would love to hear.

---

> > ### Comment · Reviewer_EHBK · 2023-08-21
> >
> > Thank you for your response.

---

### Decision · Program_Chairs · 2023-09-21

**Decision:**

Accept (spotlight)

**Comment:**

The paper presents an adaptively regularized Lasso algorithm for sparse linear regression with ill-conditioned random covariates. Consistency analysis shows that the proposed method is able to achieve near-optimal sample complexity even if the convariance matrix has a few outlier eigenvalues. After discussions, the reviewers generally agreed that this is a theoretically sound paper that contributes several new insights into the addressed fundamental problem. The meta-reviewer thus would be happy to recommend the paper for acceptance.